# Towards a simple representation of chalk hydrology in land surface modelling

**Mostaquimur Rahman[1], Rafael Rosolem[1,2]**

[1]Department of Civil Engineering, University of Bristol, Bristol, UK

[2]Cabot Institute, University of Bristol, Bristol, UK

## Abstract

Modelling and monitoring of hydrological processes in the unsaturated zone of chalk, a porous medium with fractures, is important to optimize water resources assessment and management practices in the United Kingdom (UK). However, incorporating the processes governing water movement through chalk unsaturated zone in a numerical model is complicated mainly due to the fractured nature of chalk that creates high-velocity preferential flow paths in the subsurface. In general, flow through chalk unsaturated zone is simulated using dual-porosity concept, which often involves calibration of relatively large number of model parameters, potentially undermining applications to large regions. Therefore, this approach may be not be suitable for large-scale land surface modelling applications. In this study, a simplified parameterization, namely the Bulk Conductivity (BC) model is proposed for simulating hydrology in chalk unsaturated zone. This new parameterization is implemented in the Joint UK Land Environment Simulator (JULES) and applied to a study area encompassing the Kennet catchment in the Southern UK. The simulation results are evaluated using field measurements and satellite remote sensing observations of various fluxes and states of the hydrological cycle (e.g., soil moisture, runoff and latent heat flux) at two distinct spatial scales (i.e., point and catchment). The results demonstrate that the inclusion of the BC model in JULES improves simulated land surface mass and energy fluxes over the chalk-dominated Kennet catchment. Therefore, the simple approach described in this

study may be used to incorporate the flow processes through chalk unsaturated zone in large
scale land surface modelling applications.
**Keywords:** Chalk hydrology, macroporosity, land surface model, bulk conductivity model.

## 1. Introduction

Chalk can be described as a fine-grained porous medium traversed by fractures [*Price et al.*,
1993]. Previous studies showed that the unsaturated zone of the chalk aquifers plays an
important role on groundwater recharge in the UK [e.g., *Lee et al.*, 2006; *Ireson et al.*, 2009].
Therefore, both monitoring [e.g., *Bloomfield*, 1997; *Ireson et al.*, 2006] and modelling [e.g.,
*Bakopoulou*, 2015; *Brouyère*, 2006; *Ireson and Butler*, 2011, 2013; *Sorensen et al.*, 2014]
strategies have been adapted previously to understand the governing hydrological processes
in the chalk unsaturated zone.
In chalk, the matrix provides porosity and storage capacity, while the fractures greatly
enhance permeability [*Van den Daele et al.*, 2007]. Water movement through chalk matrix is
slow due to its relatively high porosity (0.3-0.4) and low permeability ($10^{-9}$-$10^{-8}$ ms$^{-1}$). A
fractured chalk system, in contrast, conducts water at a considerably higher velocity because
of relatively high permeability ($10^{-5}$-$10^{-3}$ ms$^{-1}$) and low porosity (of the order $10^{-4}$) of
fractures [*Price et al.*, 1993].
Simulating water flow through the matrix-fracture system of chalk has been the subject of
research for some time. Both conceptual [e.g., *Price et al.*, 2000; *Haria et al.*, 2003] and
physics-based [e.g., *Mathias et al.*, 2006; *Ireson et al.*, 2009] models have been proposed
previously to describe water flow through chalk unsaturated zone. The physics-based models
mentioned above were developed based on dual-continua approach and required relatively
large number of parameters that were calibrated via inverse modelling using observed soil
moisture and matric potential data.
In recent years, representation of chalk has gained attention in land surface modelling.
*Gascoin et al.* [2009] applied the Catchment Land Surface Model (CLSM) over the Somme
River basin in northern France. A linear reservoir was included in the TOPMODEL based
runoff formulation of CLSM to account for the contribution of chalk aquifers to river
discharge. *Le Vine et al.* [2016] applied the Joint UK Land Environment Simulator (JULES
[*Best et al.*, 2011]) over the Kennet catchment in southern England to evaluate the
hydrological limitations of land surface models. In that study, two intersecting Brooks and
Corey curves were proposed, which allowed a dual curve soil moisture retention
representation for the two distinct flow domains of chalk (i.e., matrix and fracture) in the
model. Considering this dual Brooks and Corey curve, a three-dimensional groundwater flow
model (ZOOMQ3D [*Jackson and Spink*, 2004]) was coupled to JULES to demonstrate the
strong influence of representing chalk hydrology and groundwater dynamics on simulated
soil moisture and runoff.
The above mentioned studies illustrate the importance of representing  chalk in land surface
modelling. However, including chalk hydrology in large-scale land surface modelling using
the contemporary dual-porosity concept can be complicated because this approach generally
involves relatively large number of parameters. In this context, we propose a new
parameterization, namely the Bulk Conductivity (BC) model as a first step towards a simple
chalk representation suitable for land surface modelling. The BC model is included in JULES
and evaluated at two distinct spatial scales (i.e., point and catchment). At the point-scale, the
BC model is evaluated using observed soil moisture data. The proposed model is then applied
to the Kennet catchment in the Southern England and the fluxes and states of the hydrological
cycle are simulated for multiple years. The simulation results are evaluated using observed
latent heat flux (LE) and runoff to assess the performance of the BC model in simulating land
surface processes at the catchment scale.

## 2. A model of flow through chalk unsaturated zone

In this study, the *Bulk Conductivity* (BC) model based on the work by *Zehe et al.* [2001] is incorporated in JULES to represent the flow of water through the fractured chalk unsaturated zone. According to this approach, if the relative saturation ($S$) exceeds a certain threshold ($S_0$) at a soil grid, the saturated hydraulic conductivity ($K_s$) is increased to a bulk saturated hydraulic conductivity ($K_{sb}$) as follows

$$K_{sb} = K_s + K_s f_m \frac{S-S_0}{1-S_0} \qquad\qquad \text{if } S > S_0 \qquad\qquad (1)$$

with $\quad S = \frac{\theta - \theta_r}{\theta_s - \theta_r}$

where $f_m$ is a macroporosity factor (-), $\theta$ is soil moisture ($m^3 m^{-3}$), $\theta_s$ is soil moisture at saturation ($m^3 m^{-3}$), and $\theta_r$ is the residual soil moisture ($m^3 m^{-3}$). Note that $S$ ranges from zero in case of completely dry soils to one for fully wet soils.

At the first step of evaluation, the $K_s$, $S_0$ and $f_m$ parameters are estimated based on existing literature to assess the performance of the uncalibrated BC model. In this uncalibrated BC model, $K_s$ for chalk matrix is 16 mmd$^{-1}$ according to *Le Vine et al.* [2016] for the catchment investigated in this study (Figure 1). Equation 1 indicates that the onset of water flow through the fracture system of chalk is controlled by the threshold $S_0$. According to *Wellings and Bell* [1980], water flow through fractures dominates over matrix flow in chalk when the pressure head in soil becomes higher than -0.50 mH$_2$O. We consider a value of $S_0 = 0.80$ for the uncalibrated BC model, which is based on observed soil moisture-matric potential relationship in the study area (Figure S1).

In *Zehe et al.* [2001], $f_m$ was defined as the ratio of the saturated water flow rate in all macropores in a model element to the corresponding value in soil matrix, which can be determined based on the density and length of fractures at small scales. In addition, $f_m$ has

also been considered as a calibration parameter previously [e.g., *Blume*, 2008; *Zehe et al.*,
2013]. In this study, we define $f_m$ as a characteristic soil property reflecting the influence of
fractures on soil water movement [*Zehe and Blöschl*, 2004], and estimate it from the relative
difference of permeability between chalk matrix and fractured chalk system that can be of the
order $10^4$-$10^6$ according to *Price et al.* [1999]. Consequently, we consider a macroporosity
factor of $f_m = 10^5$ for the uncalibrated BC model.
In the following step, the BC model parameters are optimized to minimize the differences
between the variability of observed and simulated soil moisture. Price et al. [1999] argued
that the $K_s$ for chalk matrix is generally around 3-5 mmd$^{-1}$ (3.5-5.8x$10^{-5}$ mms$^{-1}$). In order to
optimize the BC model performance, we consider a range of $K_s = 0.8$-86 mmd$^{-1}$ ($10^{-5}$-$10^{-3}$
mms$^{-1}$) for chalk matrix in this study. As mentioned earlier, *S* is zero for completely dry soils
and one in case of fully wet soils. Therefore, we consider a range of 0-1.0 for $S_0$ to optimize
the BC model. For $f_m$, a range of $10^4$-$10^6$ is considered, which, as discussed earlier, is
consistent with the relative difference between the permeability of fractured chalk and chalk
matrix according to *Price et al* [1999].
We use the Root Mean Squared Error (RMSE) as the objective function to optimize the BC
model parameters [e.g., *Ireson et al.*, 2009]
$$RMSE = \frac{1}{nd}\sum_1^{nd}\sqrt{\left(\frac{1}{nt-1}\sum_2^{nt}\left(\Delta\theta_{d,t}^{obs} - \Delta\theta_{d,t}^{sim}\right)^2\right)} \qquad (2)$$
where *nd* is the number of soil layers, *nt* is the number of soil moisture observations available
for a layer *d*, $\Delta\theta^{obs}$ is the observed variability of soil moisture and $\Delta\theta^{sim}$ is the simulated
variability of soil moisture. Note that we consider $\Delta\theta$ for this optimization because of its
relevance to the water flux and recharge through chalk unsaturated zone [e.g., *Ireson and*
*Butler*, 2011]. Latin hypercube technique [e.g., *McKay et al.*, 2016] is used to generate 2000
random samples for each BC model parameter within the respective range discussed above.
We perform simulations using these random samples and calculate model performance
(Equation 2) to select the optimum parameter values for the BC model (discussed in section

123 4.1).

## 3. Methods

### 3.1. Study area

The study area encompasses the Kennet catchment located in the Southern England with an
area of about 1033 km$^2$ (Figure 1a). Generally, Kennet is rural in nature with scattered
settlements and has a maximum altitude of approximately 297 m (Above Ordnance Level).
The River Kennet discharges into the North Sea through London. The major tributaries of
this river are Lambourn, Dun, Enborne, and Foudry Brook. An average annual rainfall of
approximately 760 mm was recorded in the catchment over a 40 year period from 1961-1990.
Solid geology of the Kennet catchment is dominated by chalk, which is overlain by thin soil
layer. While lower chalk outcrops along the northern catchment boundary, progressively
younger rocks are found in the southern part. In general, surface runoff production is very
limited over the regions of the catchment where chalk outcrops. The flow regime shows a
distinct characteristics of slow response to groundwater held within the chalk aquifer [*Le*
*Vine et al.*, 2016]. According to *Ireson and Butler* [2013], the unsaturated zone of chalk
shows slow drainage over summer and bypass flow during wet periods in this catchment.

### 3.2. Field measurements and remotely sensed data

Table 1 summarizes the field measurements and remote sensing data used in this study. We
use *in-situ* soil moisture and runoff measurements along with remotely sensed LE data to
assess model performance in simulating the mass and energy balance components of the
hydrological cycle. Point scale soil moisture measurements at two adjacent sites (~20 m
apart) at the Warren Farm (Figure 1) were provided by Centre for Ecology and Hydrology
(CEH). A Didcot neutron probe was used at these locations to measure fortnightly soil
moisture at different depths below land surface (10 cm apart down to 0.8 m, 20 cm apart
between 0.8-2.2 m, and 30 cm apart between 2.2-4.0 m) [*Hewitt et al.*, 2010].
The National River Flow Archive (NRFA) coordinates discharge measurements from the
gauging station networks across UK. These networks are operated by the Environmental
Agency (England), Natural Resources Wales, Scottish Environment Protection Agency, and
Rivers Agency (Northern Ireland). We use discharge measurement provided by NRFA to
calculate the runoff ratio over the Kennet catchment in this study.
The MOD16 product of the Moderate Resolution Imaging Spectroradiometer (MODIS) is a
part of NASA/EOS project that provides estimation of global terrestrial LE. The LE
estimation from MOD16 is based on remotely sensed land surface data [e.g., *Mu et al.*, 2007].
In this study, the 8-day and monthly LE data products from MODIS is used to evaluate the
model performance in simulating land surface energy fluxes.

**3.3. Land surface model**

In this study, we use the Joint UK Land Environment Simulator (JULES [e.g., *Best et al.*,
2011; *Clark et al.*, 2011]) version 4.2. JULES is a flexible modelling platform with a modular
structure aligned to various physical processes developed based on the Met Office Surface
Exchange Scheme (MOSES [e.g., *Cox et al.*, 1999; *Essery et al.*, 2003]). Meteorological data
including precipitation, incoming short- and longwave radiation, temperature, specific
humidity, surface pressure, and wind speed are required to drive JULES. Each grid box in
JULES can comprise nine surface types (broadleaf trees, needle leaf trees, C3 grass, C4 grass,
shrubs, inland water, bare soil, and ice) represented by respective fractional coverage. Each
surface type is represented by a tile and a separate energy balance is calculated for each tile.
Subsurface heat and water transport equations are solved based on finite difference
approximation in JULES as described in *Cox et al.* [1999]. Moisture transport in the
subsurface is described by the finite difference form of Richards' equation. The vertical soil
moisture flux is calculated using the Darcy's law. While the top boundary condition to solve
the Richards' equation is infiltration at soil surface, the bottom boundary condition in JULES
is free drainage that contributes to subsurface runoff.
Surface runoff is calculated by combining the equations of throughfall and grid box average
infiltration in JULES. In order to direct the generated runoff to a channel network, river
routing is implemented based on the discrete approximation of one-dimensional kinematic
wave equation [e.g., *Bell et al.*, 2007]. In this approach, river network is derived from the
digital elevation model (DEM) of the study area and different wave speeds are applied to
surface and subsurface runoff components and channel flows [e.g., *Bell and Moore*, 1998]. A
return flow term accounts for the transfer of water between subsurface and land surface [e.g.,
*Dadson et al.*, 2010, 2011].
**3.4. Model configurations and input data**
In this study, simulations are performed at two distinct spatial scales, namely point and
catchment. At the point scale, JULES is configured to simulate the mass and energy fluxes at
the Warren Farm site (Figure 1a). A total subsurface depth of 5 m is considered in the model
with a vertical discretization ranging from 10 cm at the land surface to 50 cm at the bottom of
the model domain.  Note that this discretization is consistent with the soil moisture
measurement depths mentioned in section 3.2. The vegetation type is implemented as C3
grass using the default parameters in JULES. Point scale simulations were performed over 2
consecutive years from 2003-2005 at an hourly time step. Except for precipitation, hourly
atmospheric forcing data to drive JULES was obtained from an automatic weather station
operated by the CEH at Warren Farm. In order to estimate hourly precipitation data to run
JULES, rain gauge measurements from the Met Office [*Met Office*, 2006] were used. Inverse
distance interpolation technique [e.g. *Garcia et al.*, 2008; *Ly et al.*, 2013] was applied on
rainfall measurements from 13 gauges closest to Warren Farm (distance varies from 25-60
km) to obtain hourly precipitation for the point scale simulations.
At the catchment scale, JULES is configured over a study area encompassing the Kennet
catchment (Figure 1a) considering a uniform lateral grid resolution of 1 km with 70 x 40 cells
in x and y dimensions, respectively. The total subsurface depth and vertical discretization are
identical to those of the point scale simulations. Spatially distributed vegetation type
information for the study area (Figure 1b) is obtained from the Land Cover Map 2007
(LCM2007) dataset [*Morton et al.*, 2011]. Simulations were performed over 5 consecutive
years from 2006-2011 at the catchment scale. Note that the simulation periods of catchment
and point scale (2003-2005) does not coincide due to the availability of soil moisture
measurements described in section 3.2. Spatially distributed meteorological data from the
Climate, Hydrology and Ecology research Support System (CHESS) was used to obtain the
atmospheric forcing to drive JULES at the catchment scale. The CHESS data includes 1 km
resolution gridded daily meteorological variables [*Robinson et al.*, 2015]. This daily data is
downscaled using a disaggregation technique described in *Williams and Clark* [2014] to
obtain hourly atmospheric forcing. The flow direction required for river routing is extracted
from the USGS HydroSHEDS digital elevation data [*Lehner et al.*, 2008].
We estimate the soil hydraulic properties based on texture (Table 2). At the point scale, loam
soil is dominant at the Warren Farm site. At the catchment scale, the Harmonized World Soil
Database (HWSD) from the Food and Agricultural Organization of UNO (FAO) is used to
obtain the texture of different soil types over Kennet (Figure 1c). The saturation-pressure
head relationship for different soil types is described using the Van Genuchten [*Van
*Genuchten*, 1980] model with parameter values (Table 2) obtained from *Schaap and Leij*
[1998].
Table 3 summarizes the hydraulic properties for chalk used in this study. These properties are
obtained based on existing literature as a first step when evaluating the uncalibrated BC
model. The BC model parameters are subsequently optimized to minimize the differences
between observed and simulated $\Delta\theta$.
In this study, we consider two different model configurations, namely *default* and *macro*
(Figure 2). The *default* configuration corresponds to the standard parameterizations of JULES
that does not represent chalk hydrology in the model. In this configuration, each soil column
in JULES is considered to be vertically homogeneous with the soil properties defined in
Table 2, which is motivated by the Met Office JULES Global Land 4.0 configuration
described in *Walters et al.* [2014]. The *macro* configuration, in contrast, explicitly represents
chalk by applying the BC model starting at 30 cm below land surface to the bottom of the
model domain (i.e. 500 cm). Therefore, the soil column in the *macro* configuration can be
divided into topsoil (0-30 cm) and chalk (30-500 cm) in *macro*. Note that except for this
inclusion of chalk, *default* and *macro* configurations are identical in terms of model set up
and input data.
The topsoil depth of 30 cm in the *macro* configuration is defined based on several augured
soil samples collected during a field campaign at Warren Farm in 2015 (Figure 2). This depth
is corroborated by additional information from the British Geological Survey (BGS) operated
borehole records (http://www.ukso.org/pmm/soil_depth_samples_points.html), which show
that topsoil depths vary from 10-40 cm over the study area. We therefore apply the *macro*
configuration assuming a spatially homogeneous topsoil depth of 30 cm for both point and
catchment scale simulations.

## 4. Results and discussion

### 4.1. Point scale simulations

At the point scale, the simulation results are evaluated using soil moisture observations at the Warren Farm site. Figure 3a compares observed and simulated soil moisture ($\theta$) from the *default* and *macro* configurations at 2 m below land surface. Note that the *macro* configuration uses the chalk hydraulic parameters collected from existing literature (Table 3). This figure shows that the *default* configuration underestimates $\theta$ throughout the simulation period, which is improved remarkably in case of *macro*. Figure 3b plots observed and simulated soil moisture variability ($\Delta\theta$) from the *default* and *macro* configurations at the Warren Farm site. In general, both configurations show discrepancies with observed $\Delta\theta$ with *macro* showing relatively better model performance.

The results show that despite the *macro* configuration improves simulated $\theta$, it shows considerable discrepancies with observed $\Delta\theta$, which is consistent throughout the whole chalk profile (results from other model layers are not shown). In order to minimize the differences between observed and modelled $\Delta\theta$ from the *macro* configuration, we optimize the BC model following the methodology described in section 2. The optimization results are summarized in Figure 4. Note that for each combination considered in the optimization, 2000 model runs were performed using randomly sampled parameters as discussed in section 2. Figure 4 presents the results from the model runs yielding the lowest RMSE.

The RMSE between observed and simulated $\Delta\theta$ for the model configurations considered in the optimization is shown in Figure 4a. This figure illustrates that the RMSE of the *default* configuration is larger than that of *macro*, indicating better model performance in reproducing $\Delta\theta$ for the latter. Therefore, it appears that the uncalibrated BC model (i.e., the *macro* configuration) is better in reproducing soil moisture variability compared to *default*.

Figure 4b, c and d presents the BC model parameter values from the model run producing the
lowest RMSE for each configuration. Concerning single BC model parameters, Figure 4a
shows that optimizing $S_0$ results in a 16% reduction of RMSE compared to the *macro*
configuration. Optimizing $K_s$ marginally improves model performance, which is observed
from a slightly lower (4%) RMSE than *macro*. Optimizing both $K_s$ and $S_0$ simultaneously
results in the largest reduction (24%) of RMSE compared to *macro*.
Additionally, Figure 4 suggests that the sensitivity of $S_0$ on the model performance in
simulating $\Delta\theta$ is substantially higher compared to $K_s$ and $f_m$, which is corroborated by the
sensitivity of the individual model parameters (Figure S2). Figure 4a also reveals the
interesting fact that the RMSE from the configuration with optimized $K_s$ and $S_0$ is identical to
that of the one with all 3 parameters optimized simultaneously (i.e., $K_s$, $S_0$ and $f_m$). Therefore,
we select the *macro* configuration with optimized $K_s$ and $S_0$ (*macro$_{opt}$* hereafter) to simulate
chalk hydrology over the study area.
Figure 5 compares $\Delta\theta$ from the *macro$_{opt}$* configuration ($\Delta\theta_{opt}$) with observed soil moisture
variability ($\Delta\theta_{obs}$). As mentioned earlier, $\Delta\theta_{default}$ and $\Delta\theta_{macro}$ show considerable discrepancies
with $\Delta\theta_{obs}$ while the *macro* configuration exhibits relatively better performance. Figure 5
illustrates that the overall agreement between observed and simulated $\Delta\theta$ improves
substantially in case of *macro$_{opt}$* compared to *default* and *macro*, which is pronounced
especially in the deeper chalk layers. Therefore, this figure indicates that the performance of
the BC model in simulating $\Delta\theta$ is further improved by optimizing the $K_s$ and $S_0$ parameters at
the Warren Farm site.
In order to assess the model performance in simulating soil moisture over the entire column,
the relative bias ($\Delta\mu$, see Appendix) of simulated $\theta$ from the *default* and *macro$_{opt}$*
configurations at Warren Farm for various depth ranges are shown in Figure 6. In the soil
layers (0-30 cm), $\theta$ from the two configurations are comparable with the *default* showing
slightly lower mean relative bias ($\Delta\mu_{mean}$) of -0.03 than *macro_{opt}* ($\Delta\mu_{mean}$ = -0.09). However,
in the chalk layers (30-500 cm), *default* simulates substantially drier conditions,
corresponding to $\Delta\mu_{mean} \leq$ -0.28. In contrast, the *macro_{opt}* configuration considerably
improves the agreement between the simulated and observed $\theta$ in the chalk layers with
$\Delta\mu_{mean} \geq$ -0.05. Therefore, the results indicate that the inclusion of the BC model in JULES
improves the performance of overall soil moisture simulation (both $\theta$ and $\Delta\theta$) at Warren
Farm especially in the chalk layers.
The drainage flux through the bottom of soil column ($d_b$) of a land surface model can be
considered as the potential recharge flux to groundwater [e.g., *Sorensen et al.*, 2014]. Figure
7 compares the daily sum of $d_b$ from the *default* and *macro_{opt}* configurations at the Warren
Farm site. The rainfall characteristics over the study period is shown in Figure 7a. In Figure
7b, the *macro_{opt}* configuration shows considerable $d_b$ during the colder months, while slow
drainage prevails throughout the rest of the year. In contrast, the *default* configuration shows
relatively high $d_b$ in summer compared to the colder months. In general, the recharge rate
through chalk unsaturated zone during the warmer periods of the year is lower than that in the
winter months [*Wellings and Bell*, 1980; *Ireson et al.*, 2009]. Therefore, the *macro_{opt}*
configuration appears to be more consistent with the recharge mechanism in chalk compared
to *default*.
In this section, the BC model was evaluated at the point scale. The results showed that in
general, the *macro* configuration performs relatively better in simulating $\theta$ and $\Delta\theta$ compared
to *default*. In order to improve the model performance even further, parameter optimization
was performed to minimize the differences between observed and simulated $\Delta\theta$ at the point
scale. In the next sections, the optimized model (*macro_{opt}*) is evaluated at the catchment scale.

**4.2. Catchment scale simulations**

In the previous section, it was observed that the *default* configuration generally

underestimates $\theta$ compared to *macro$_{opt}$*. Previous studies have demonstrated the

interconnections between shallow soil moisture and LE [e.g., *Chen and Hu*, 2004]. In order to

assess the differences between the LE from the *default* and *macro$_{opt}$* configurations at the

catchment scale, Figure 8 plots spatially averaged 8-day composites of LE from MODIS

(LE$_{MOD}$) against the LE from these configurations (LE$_{default}$ and LE$_{opt}$, respectively) over

Kennet. The agreement between simulated LE and LE$_{MOD}$ is evaluated using the coefficient

of determination ($R^2$, see Appendix) and mean bias. Comparison between LE$_{default}$ and LE$_{MOD}$

shows a coefficient of determination of $R^2_{default} = 0.78$ and a mean bias of $bias_{default} = 10.5$

Wm$^{-2}$. The agreement between simulated LE and LE$_{MOD}$ improves in case of the *macro$_{opt}$*

configuration, which is reflected by an increased coefficient of determination of $R^2_{opt} = 0.81$

and a reduced mean bias of $bias_{opt} = 3$ Wm$^{-2}$.

Figure 8 shows differences between *LE$_{default}$* and *LE$_{opt}$* especially for relatively high *LE*,

indicating discrepancies especially during the warmer months of the year. Figure 9 presents

spatially averaged time series of monthly *LE$_{MOD}$*, *LE$_{default}$* and *LE$_{opt}$*. This figure shows that

the differences between *LE$_{default}$* and *LE$_{opt}$* increases substantially in summer compared to the

colder months of the year, which is consistent with Figure 8. Consequently, the *default*

configuration underestimates *LE* in summer compared to *LE$_{MOD}$*, which is improved in case

of the *macro$_{opt}$* configuration. In contrast, the differences between *LE$_{default}$* and *LE$_{opt}$* are

negligible during the colder months of the year.

Table 4 compares observed and simulated daily average runoff from the two model

configurations over the Kennet catchment from 2006-2011. The runoff ratio (*RR*, see

Appendix), which is equal to the mean volume of flow divided by the volume of precipitation

[e.g., *Kelleher et al.*, 2015], assesses the partitioning of precipitation into runoff over the
catchment. The *default* configuration ($RR = 0.82$) shows considerably higher $RR$ compared to
observation ($RR = 0.40$), indicating overestimation of runoff by the model. Including chalk
hydrology in the model remarkably improves the agreement between observed and simulated
mean runoff over the Kennet catchment, which is assessed from a runoff ratio of $RR = 0.37$
for the *macro_opt* configuration.
In Table 4, the relative bias ($\Delta\mu$) of 1.04 between observed and simulated runoff from the
*default* configuration again indicates the overestimation by the model. In comparison,
*macro_opt* shows a relative bias ($\Delta\mu = -0.05$), indicating improvement between observed and
simulated mean runoff volume compared to *default*. The relative difference in standard
deviation ($\Delta\sigma$, see Appendix) compares the variability of observed and simulated runoff in
Table 4. This comparison shows that the *default* configuration overestimates the variability of
runoff over the Kennet catchment ($\Delta\sigma = 2.04$), which is improved in case of *macro* ($\Delta\sigma = $

350 0.70).

It was demonstrated previously that the *default* configuration predicts lower
evapotranspiration (ET) compared to *macro_opt* over the Kennet catchment due to the
differences in simulated $\theta$. In JULES, moisture from soil and canopy is depleted to meet the
ET demand. Additionally, surface runoff generation depends on canopy water storage in the
model [*Best et al.*, 2011]. Because of this connection between ET and surface runoff
generation via canopy water storage, the differences in runoff demonstrated in Table 4 can be
attributed to the disagreements between $LE_{default}$ and $LE_{macro}$ (Figure 8) due to the relatively
drier conditions simulated by *default*.
In this section, the BC model is evaluated using observed mass and energy fluxes over the
Kennet catchment. The *default* configuration showed considerably low LE over the
catchment, which was pronounced during the warmer period of the year. The agreement
between observed and simulated LE was improved in case of the *macro_{opt}* configuration
compared to *default*. It was also observed that the overall runoff prediction was improved by
*macro_{opt}* compared to *default*. Given its simplicity, our results indicate that the proposed
parameterization is suitable for use in land surface modelling applications.

## 5. Summary and Conclusions

In this study, we proposed a simple parameterization, namely the *Bulk Conductivity* (BC)
model  to simulate water flow through the matrix-fracture system of chalk in large scale land
surface modelling applications. This parameterization was implemented in the Joint UK Land
Environment Simulator (JULES) and applied to the Kennet catchment located in the southern
UK to simulate the mass and energy fluxes of the hydrological cycle for multiple years. Two
model configurations, namely *default* and *macro* were considered with the latter using the BC
model to simulate chalk hydrology.
The proposed BC model is a single continuum approach of modelling preferential flow [e.g.,
*Beven and Germann*, 2013] that involves only 2 parameters, namely macroporosity factor ($f_m$)
and relative saturation threshold ($S_0$). Initially, these parameters along with the saturated
hydraulic conductivity of the chalk matrix were estimated from existing literature. Finally,
the BC model parameters were optimized to minimize the differences between observed and
simulated soil moisture variability. Our results indicated that $S_0$ is the most influential
parameter in the model when representing water movement through a soil-chalk column,
followed by the saturated hydraulic conductivity of chalk matrix while $f_m$ showed low
sensitivity. Hence, the parameterization is further improved by optimizing both saturated
hydraulic conductivity of chalk matrix and $S_0$ to minimize the differences between observed
and simulated soil moisture variability.
The simulation results were evaluated using observed mass and energy fluxes both at point
and catchment scales. The results demonstrated that the inclusion of the BC model in JULES
improves simulated soil moisture variability at the point scale compared to a model
configuration that does not represent chalk in the subsurface (i.e., the *default* configuration).
At the catchment scale, it was illustrated that the proposed parameterization improves
simulated latent heat flux and overall runoff compared to the *default* configuration.
Note that the complexity of the BC model for simulating water flow through chalk
unsaturated zone is substantially lower compared to more commonly used models for this
purpose (e.g., dual-porosity models). Despite its simplicity, it appears that the proposed
parameterization improves mass and energy fluxes simulated by JULES over the Kennet
catchment. As mentioned previously, representing chalk hydrology in land surface models
using the dual-porosity concept is complicated mainly due to the relatively large number of
parameters involved in such approach. Therefore, the simplified parameterization proposed in
this study may be useful for large-scale land surface modelling applications over chalk-
dominated areas.

## Acknowledgements

We gratefully acknowledge the support by the "A MUlti-scale Soil moisture
Evapotranspiration Dynamics study – AMUSED" project funded by Natural Environment
Research Council (NERC) grant number NE/M003086/1. The authors would also like to
thank Ned Hewitt and Jonathan Evans from the Centre for Ecology and Hydrology (CEH) for
providing the data for the point-scale analyses at the Warren Farm. Finally, we would like to
thank Miguel Rico-Ramirez (University of Bristol) for helping preparing the precipitation
data from the rain gauge network used for the point-scale simulations, Thorsten Wagener
(University of Bristol) for his valuable suggestions on model diagnostics, and Joost Iwema
(University of Bristol) for helping with the soil samples collected during the 2015 field work
campaign.

## Appendix

**Definition of Statistical Metrics**

Coefficient of determination ($R^2$) for observation y = y1, …, yn and prediction f = f1, …, fn
is defined as
$$R^2 = 1 - \frac{SS_{res}}{SS_{tot}}$$
where, $SS_{res}$ is the residual sum of square and $SS_{tot}$ is the total sum of square. $SS_{res}$ and $SS_{tot}$
are defined as
$$SS_{res} = \sum_{i=1}^{n}(y_i - f_i)^2 \qquad \text{and}$$
$$SS_{tot} = \sum_{i=1}^{n}(y_i - \bar{y})^2 \qquad \text{with } \bar{y} \text{ being the mean of y.}$$
Runoff ratio (RR) assesses the portion of precipitation that generates runoff over the
catchment. RR is defined as
$$RR = \frac{\mu_{runoff}}{\mu_{rain}}$$
where $\mu_{runoff}$ is mean runoff and $\mu_{rain}$ is mean precipitation [e.g., *Kelleher et al.*, 2015].
Relative bias ($\Delta\mu$) between observed and simulated time series can be defined as
$$\Delta\mu = \frac{\mu_{mod} - \mu_{obs}}{\mu_{obs}}$$
where $\mu_{obs}$ and $\mu_{mod}$ are the mean of observed and simulated time series, respectively. While
the optimal value of $\Delta\mu$ is zero, negative (positive) values indicate an underestimation
(overestimation) by the model [e.g., *Gudmundsson et al.*, 2012].
Relative difference in standard deviation ($\Delta\sigma$) between observed and simulated time series
can be defined as
$\Delta\sigma = \frac{\sigma_{mod} - \sigma_{obs}}{\sigma_{obs}}$
where $\sigma_{obs}$ and $\sigma_{mod}$ are the standard deviation of observed and simulated time series,
respectively [e.g., *Gudmundsson et al.*, 2012].

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

# Tables

Table 1. Field measurements and remote sensing data.

| Data | Spatial scale | Temporal extent | Frequency | Source |
|---|---|---|---|---|
| Soil moisture | Point[a] | 2003-2005 | 15 day | N. Hewitt (CEH) |
| Latent heat flux | Global | 2006-2011 | 8 day, 1 month | MODIS |
| Discharge | Point[b] | 2006-2011 | 1 day | NRFA |

[a]Measured at Warren Farm.
[b]Locations are shown in Figure 1a.

Table 2. Hydraulic properties for different soil types (refer to Figure 1c). Saturated hydraulic conductivity ($K_s$) and porosity data are obtained from *Rawls et al.* [1982]. The Van Genuchten parameters are acquired from *Schaap and Leij* [1998].

| Texture | $K_s$ (ms$^{-1}$) | Porosity (-) | $\alpha$ (m$^{-1}$) | n (-) |
|---|---|---|---|---|
| Loam | 3.7x10$^{-6}$ | 0.463 | 3.33 | 1.56 |
| Silt loam | 2.0x10$^{-6}$ | 0.50 | 1.2 | 1.39 |
| Clay | 1.7x10$^{-7}$ | 0.475 | 2.12 | 1.2 |

Table 3. Hydraulic properties of chalk

| Properties | Unoptimized | | Optimized value |
|---|---|---|---|
| | Value | Source | |
| $K_s$ (md$^{-1}$) | 16 | Le Vine et al., 2016 | 15 |
| $S_0$ (-) | 0.8 | Observations | 0.67 |
| $f_m$ (-) | 1x10$^5$ | Price et al., 1993 | 6.1x10$^5$ |
| $\alpha$ (m$^{-1}$) | 3.0 | Le Vine et al., 2016 | - |
| n (-) | 1.4 | Le Vine et al., 2016 | - |

Table 4. Comparison between observed and simulated daily average runoff from the two configurations over the Kennet catchment.

| Metric | Observed | Simulated (*default*) | Simulated (*macro*) |
|---|---|---|---|
| *RR* | 0.40 | 0.82 | 0.37 |
| $\Delta\mu$ | - | 1.04 | -0.05 |
| $\Delta\sigma$ | - | 2.04 | 0.70 |

 **Figures**

Figure 1. Location (a), vegetation cover (b), and soil texture (c) over the study area. The red
line in (a) outlines the Kennet catchment boundary, while the river network is shown in blue.
The black triangle in (a) shows the location of the discharge gauging station at the catchment
outlet while the black square corresponds to Warren Farm location where point-scale
simulations are carried out. The shaded area in (c) represents the location of chalk in the
catchment.

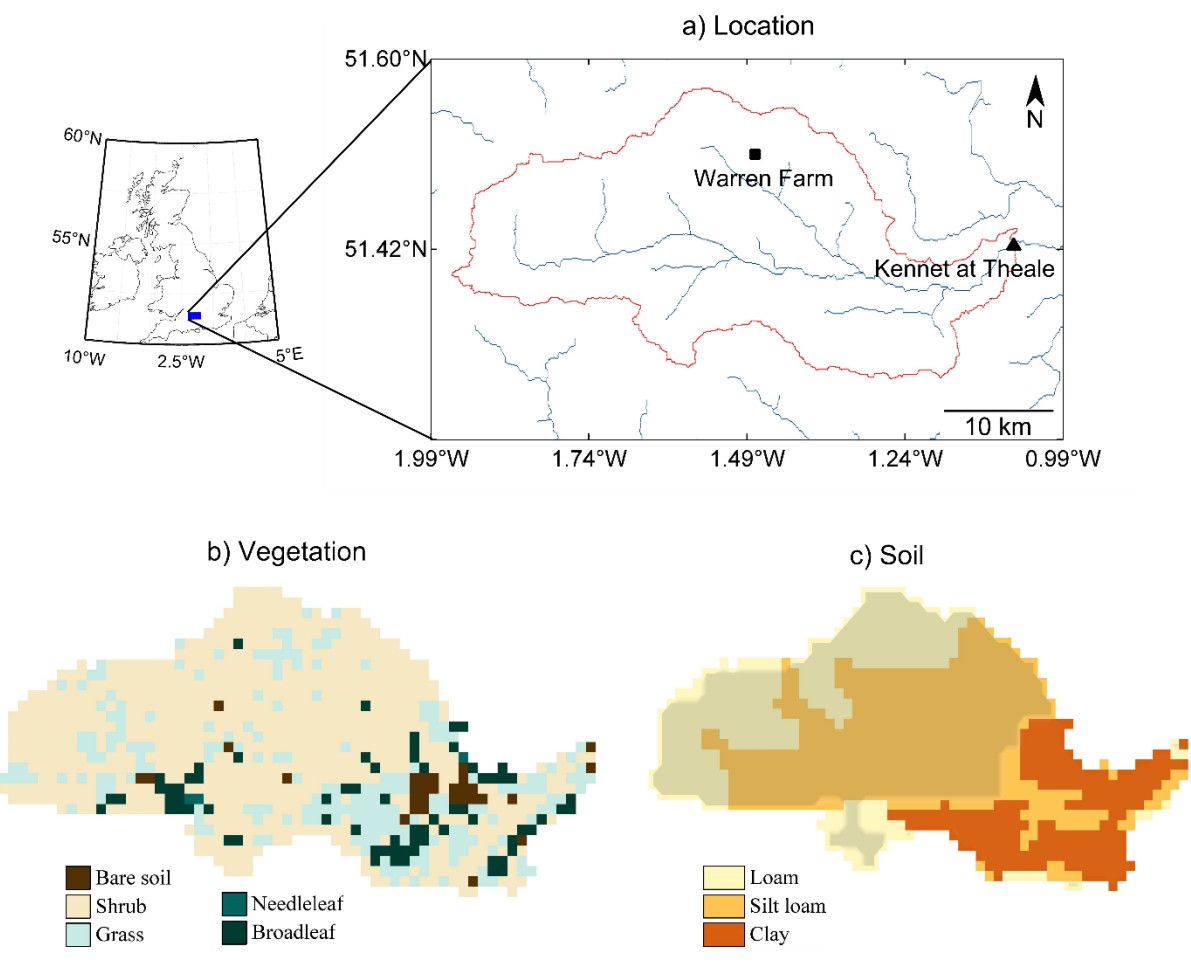





Figure 2. Example of soil profiles collected at Warren Farm during a field campaign in 2015
(a), and the two model configurations (b).













Figure 3. Comparison between observed and simulated (a) soil moisture (θ) and (b) change in soil moisture (Δθ) from the *default* and *macro* configurations at a depth of 2m below land surface. The shaded areas constructed from 2 soil moisture probes at the Warren Farm site denote the range of observed data in these plots.

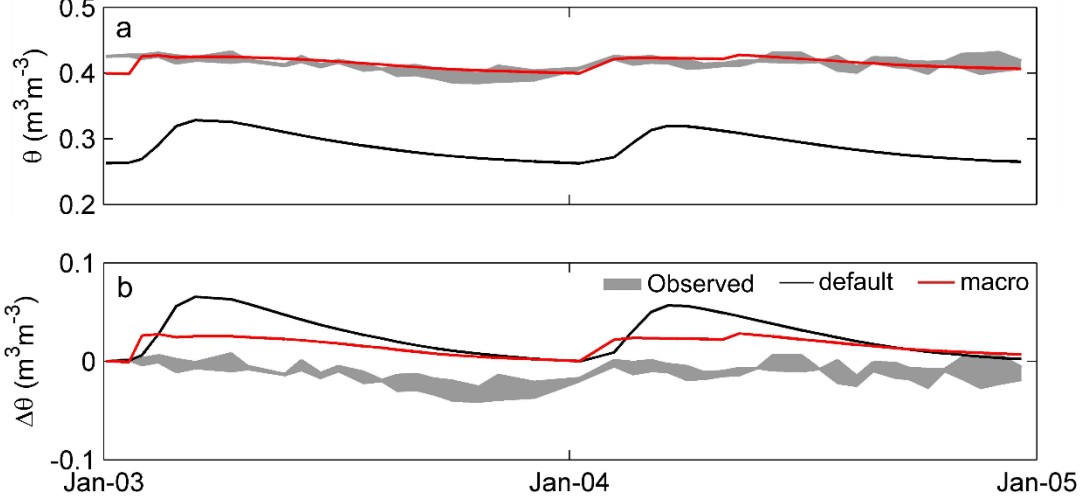

Figure 4. (a) Model performance in reproducing observed and simulated $\Delta\theta$, (b) $K_s$, (c) $S_0$ and (d) $f_m$ for various parameter combinations considered in the optimization. Note that except for the *default* and *macro*, the simulation yielding the lowest RMSE (out of 2000 model runs) is presented in this plot.

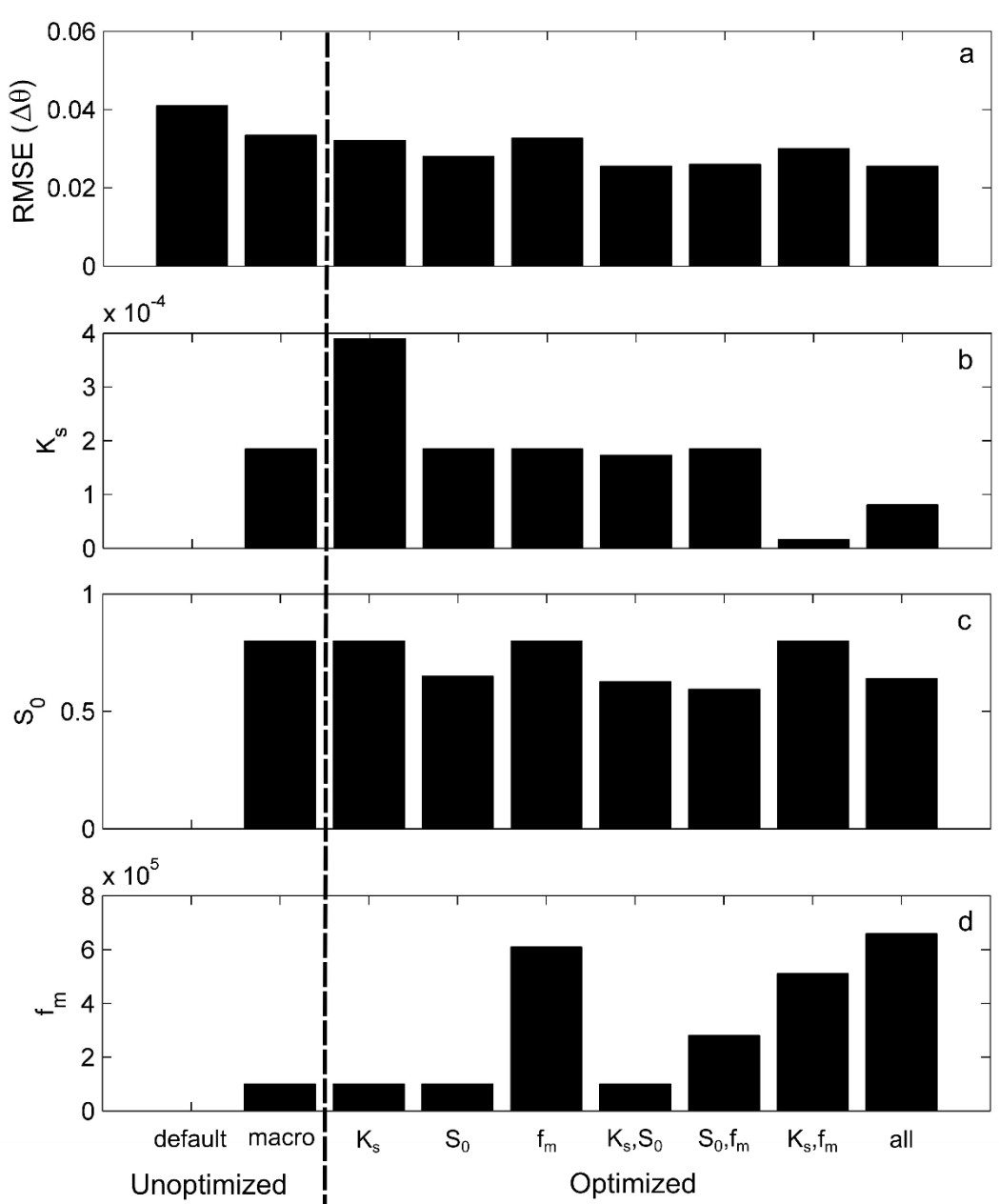

Figure 5. Comparison between observed and simulated Δθ from *default*, *macro* and *macro*<sub>*opt*</sub>
configurations at various depths below land surface. The shaded area, which is constructed
from 2 soil moisture probes at the Warren Farm site, denotes the range of Δθ.

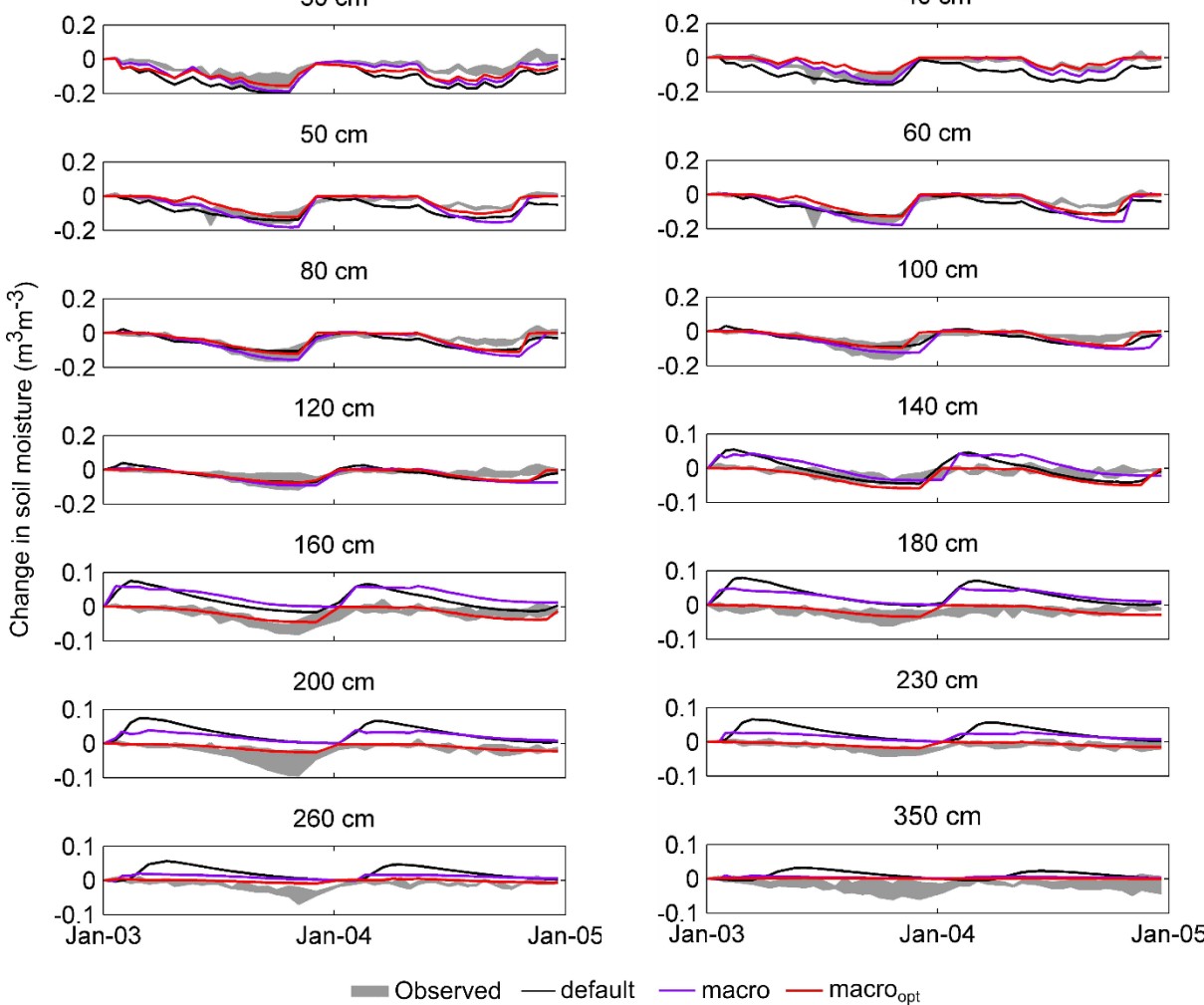






Figure 6. Box plot of relative bias (Δμ) of simulated soil moisture from *default* and *macro*
configurations at different depth ranges shown in individual intervals (e.g., 0-30 cm, 30-100
cm, and so on).

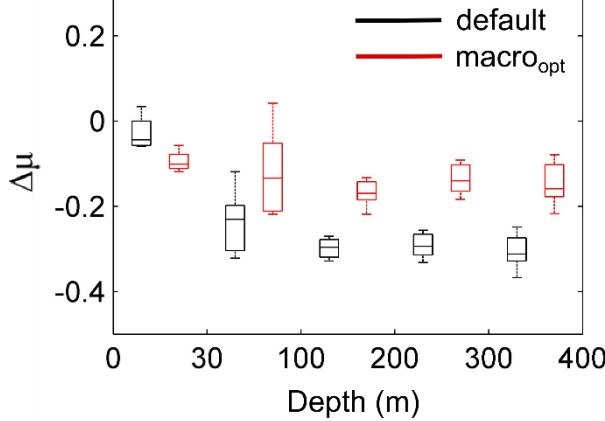













Figure 7. (a) Precipitation and (b) daily sum of drainage through the bottom of the soil
column at Warren Farm over the two simulated years (2003-2005).

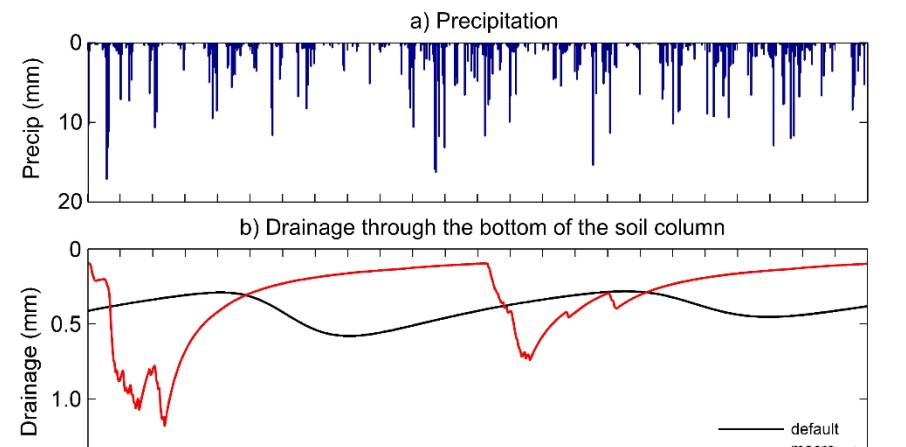













Figure 8. Catchment average 8 day composites of MODIS estimated *LE* (*LE_MOD*) against
simulated *LE* from *default* and *macro* configurations (*LE_default* and *LE_macro*, respectively) along
with the linear models fitted for *LE_default* (black line) and *LE_macro* (red line). The 1:1 line is
shown in grey, which represents the perfect fit between *LE_MOD* and simulated *LE*.

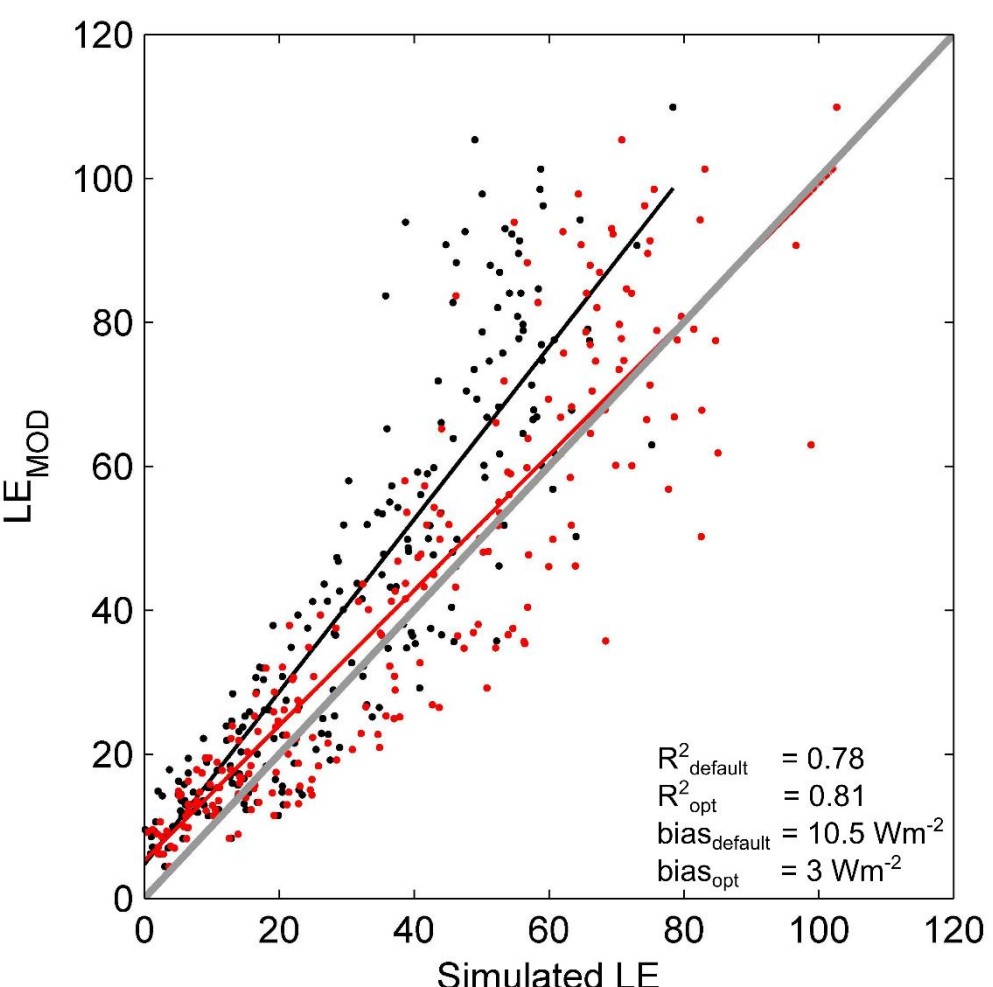





Figure 9. Spatially averaged monthly latent heat flux (*LE*) from MODIS, *default* and *macro_{opt}*
configurations over the Kennet catchment.

Monthly average latent heat flux ($Wm^{-2}$)













**Supplementary materials**
Figure S1. Saturation-pressure head relationship (May 2003 - December 2005) at Warren
Farm measured fortnightly at 40 cm below land surface. (Source: Ned Hewett, CEH, personal
communication).

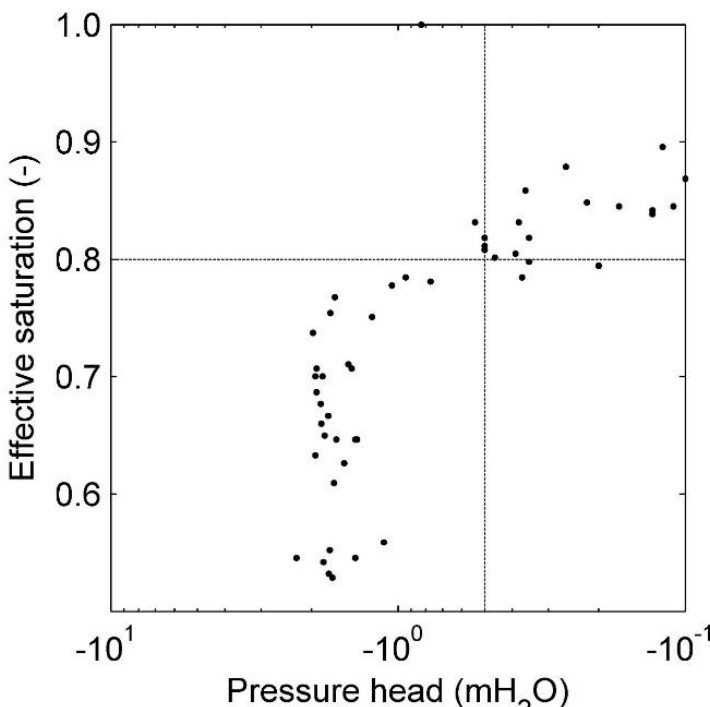









Figure S2. Sensitivity of the BC model parameters on the model performance in simulating
$\Delta\theta$. Note that the parameters are considered one-at-a-time (OAT), and the vertical axis have
different RMSE ranges.

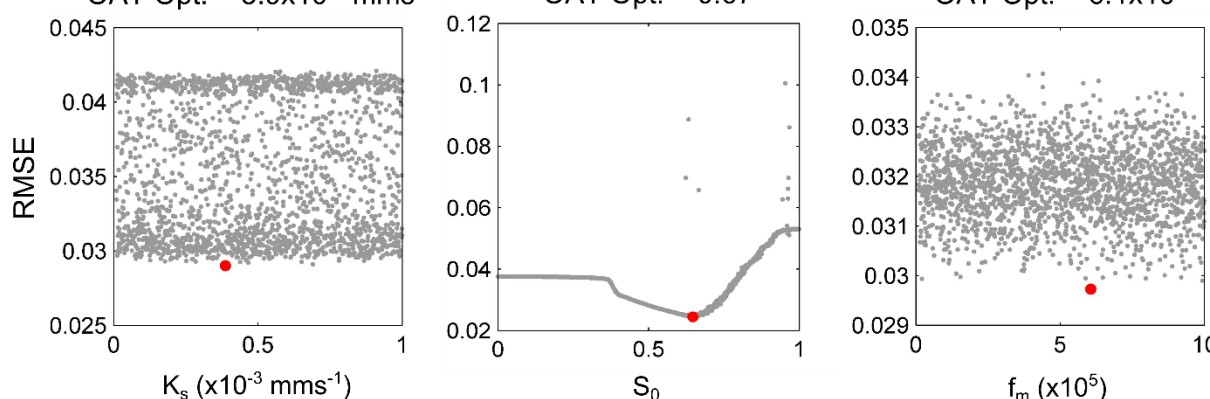


**Towards a simple representation of chalk hydrology in land**
**surface modelling**
~~The effect of chalk representation in land surface modelling~~
**Mostaquimur Rahman[1], Rafael Rosolem[1,2]**
[1]Department of Civil Engineering, University of Bristol, Bristol, UK
[2]Cabot Institute, University of Bristol, Bristol, UK
**Abstract**
Modelling and monitoring of hydrological processes in the unsaturated zone of ~~the~~ chalk,
~~which is~~ a porous medium with fractures, is important to optimize water resources assessment
and management practices in the United Kingdom (UK). However, incorporating the
processes governing water movement through chalk unsaturated zone in a numerical model is
complicated mainly due to the fractured nature of chalk that creates high-velocity preferential
flow paths in the subsurface. ~~However, efficient simulations of water movement through~~
~~chalk unsaturated zone is difficult mainly due to the fractured nature of chalk, which creates~~
~~high-velocity preferential flow paths in the subsurface.~~In general, flow through chalk
unsaturated zone is simulated using dual-porosity concept, which often involves calibration
of relatively large number of model parameters, potentially undermining applications to large
regions. Therefore, this approach may be not be suitable for large-scale land surface
modelling applications. ~~Complex hydrology in the chalk aquifers may also influence land~~
~~surface mass and energy fluxes because processes in the hydrological cycle are connected via~~
~~non-linear feedback mechanisms. In this study, it is hypothesized that explicit representation~~
~~of chalk hydrology in a land surface model influences land surface processes by affecting~~
~~water movement through the shallow subsurface. In order to substantiate this hypothesis,~~ In
this study, a simplified ~~macroporosity~~ parameterization, namely the Bulk Conductivity (BC)

model ~~is~~ is proposed for simulating hydrology in chalk unsaturated zone. This new parameterization is implemented in the Joint UK Land Environment Simulator (JULES) and~~,~~ ~~which is~~ applied to ~~on~~ a study area encompassing the Kennet catchment in the Southern UK. The simulation results are evaluated using field measurements and satellite remote sensing observations of various fluxes and states of~~in~~ the hydrological cycle (e.g., soil moisture, runoff~~,~~ and latent heat flux) at two distinct spatial scales (i.e., point and catchment). The results demonstrate that the inclusion of the BC model in JULES improves simulated land surface mass and energy fluxes over the chalk-dominated Kennet catchment. Therefore, the simple approach described in this study may be used to incorporate the flow processes through chalk unsaturated zone in large scale land surface modelling applications. ~~The results reveal the influence of representing chalk hydrology on land surface mass and energy balance components such as surface runoff and latent heat flux via subsurface processes (i.e., soil moisture dynamics) in JULES, which corroborates the proposed hypothesis.~~

**Keywords:** Chalk hydrology, macroporosity, land surface model~~ling~~, bulk conductivity model.

# 1. Introduction

Chalk can be described as a fine-grained porous medium traversed by fractures [*Price et al.*, 1993]. Previous studies showed that ~~T~~the unsaturated zone of the chalk aquifers play~~s~~ an important role on groundwater recharge ~~various important processes (e.g., recharge) of the hydrological cycle~~ in the UK [e.g., *Lee et al.*, 2006; *Ireson et al.*, 2009]. Therefore, both monitoring [e.g., *Bloomfield*, 1997; *Ireson et al.*, 2006] and modelling [e.g., *Bakopoulou, 2015; Brouyère*, 2006; *Ireson and Butler*, 2011, 2013; *Sorensen et al.*, 2014] strategies have been adapted previously to understand the governing hydrological processes in the chalk unsaturated zone.

In chalk, the matrix provides porosity and storage capacity, while the fractures greatly
enhance permeability [*Van den Daele et al.*, 2007]. Water movement through chalk matrix is
slow due to its relatively high porosity (0.3-0.4) and low permeability ($10^{-9}$-$10^{-8}$ ms$^{-1}$). A
fractured chalk system, in contrast, conducts water at a considerably higher velocity because
of relatively high permeability ($10^{-5}$-$10^{-3}$ ms$^{-1}$) and low porosity (of the order $10^{-4}$) of
fractures [*Price et al.*, 1993].
Simulating water flow through the matrix-fracture system of chalk has been the subject of
research for some time. Both conceptual [e.g., *Price et al.*, 2000; *Haria et al.*, 2003] and
physics-based [e.g., *Mathias* ~~*Mathius*~~ *et al.*, 2006; *Ireson et al.*, 2009] models have been
proposed previously to describe water flow through chalk unsaturated zone. The physics-
based models mentioned above were developed based on dual-continua approach and
required relatively large number of parameters that were calibrated via inverse modelling
using observed soil moisture and matric potential data.
~~The aforementioned studies revealed the importance of representing the matrix-fracture flow~~
~~nature in simulating subsurface hydrological processes in chalk-dominated aquifers.~~ In recent
years, representation of chalk has ~~also~~ gained attention in land surface modelling. *Gascoin et*
*al.* [2009] applied the Catchment Land Surface Model (CLSM) over the Somme River basin
in northern France. A linear reservoir was included in the TOPMODEL based runoff
formulation of CLSM to account for the contribution of chalk aquifers to river discharge. *Le*
*Vine et al.* [2016] applied the Joint UK Land Environment Simulator (JULES [*Best et al.*,
2011]) over the Kennet catchment in southern England to evaluate the hydrological
limitations of land surface models. In that study, two intersecting Brooks and Corey curves
were ~~curve was~~ proposed, which allowed a dual curve soil moisture retention representation
for the two distinct flow domains of chalk (i.e., matrix and fracture) in the model.
Considering this dual Brooks and Corey curve, a three-dimensional groundwater flow model
(ZOOMQ3D [*Jackson and Spink*, 2004]) was coupled to JULES to demonstrate the strong
influence of representing chalk hydrology and groundwater dynamics flow on simulated soil
moisture and runoff.
The above mentioned studies illustrate the importance of suggest that the representing ation
of chalk in affects the hydrological processes simulated by land surface modellings.
However, including chalk hydrology in large-scale land surface modelling using the
contemporary dual-porosity concept can be complicated because this approach generally
involves relatively large number of parameters. In this context, we propose a new
parameterization, namely the Bulk Conductivity (BC) model as a first step towards a simple
chalk representation suitable for land surface modelling. The BC model is included in JULES
and evaluated at two distinct spatial scales (i.e., point and catchment). Because the processes
of the hydrological cycle are connected via non-linear feedback mechanisms [e.g., *Kollet and*
*Maxwell*, 2008; *Rahman et al.*, 2014], the representation of water flow through the matrix-
fracture system of chalk may also influence simulated land surface energy fluxes (e.g., latent
heat flux), which has not yet been explicitly discussed. In this context, our hypothesis is that a
consistent representation of water movement through chalk in a land surface model affects
the exchange of mass and energy fluxes at the surface, which may be important to consider in
water resources assessment and management practices (e.g., flood and drought prediction
over chalk-dominated areas). In order to substantiate this hypothesis, a macroporosity
parameterization, namely the *Bulk Conductivity* (BC) model is implemented in JULES and
evaluated at two distinct spatial scales (i.e., point and catchment). At the point-scale, the BC
model is evaluated using against observed soil moisture data. The proposed model is then
applied to over the Kennet catchment in the Southern England and the fluxes and states of the
hydrological cycle are simulated for multiple years. The simulation results are evaluated
using observed latent heat flux (LE) and runoff to assess the performance of the BC model in
simulating land surface processes at the catchment scale. ~~to demonstrate the importance of~~
~~representing chalk hydrology, which supports the proposed hypothesis.~~

## 2. A model of flow through chalk unsaturated zone

In this study, the *Bulk Conductivity* (BC) model based on the work by *Zehe et al.* [2001] is
incorporated in JULES to represent the flow of water through the fractured chalk unsaturated
zone. According to this approach, if the relative saturation ($S$) exceeds a certain threshold ($S_0$)
at a soil grid, the saturated hydraulic conductivity ($K_s$) is increased to a bulk saturated
hydraulic conductivity ($K_{sb}$) as follows
$$K_{sb} = K_s + K_s f_m \frac{S-S_0}{1-S_0} \qquad \text{if } S > S_0 \qquad (1)$$
with $\quad S = \frac{\theta - \theta_r}{\theta_s - \theta_r}$
where $f_m$ is a macroporosity factor (-), $\theta$ is soil moisture ($m^3 m^{-3}$), $\theta_s$ is soil moisture at
saturation ($m^3 m^{-3}$), and $\theta_r$ is the residual soil moisture ($m^3 m^{-3}$). Note that $S$ ranges from zero
in case of completely dry soils to one for fully wet soils.
At the first step of evaluation, the $K_s$, $S_0$ and $f_m$ parameters are estimated based on existing
literature to assess the performance of the uncalibrated BC model. In this uncalibrated BC
model, $K_s$ for chalk matrix is 16 mmd$^{-1}$ according to *Le Vine et al.* [2016] for the catchment
investigated in this study (Figure 1).
Equation 1 indicates that the onset of water flow through the fracture system of chalk is
controlled by the threshold $S_0$. According to *Wellings and Bell* [1980], water flow through
fractures dominates over matrix flow in chalk when the pressure head in soil becomes higher
than -0.50 mH$_2$O. We consider a value of ~~In this study,~~ $S_0 = 0.80$ for the uncalibrated BC
model, which is based on observed soil moisture-matric potential relationship in the study
area (Figure S1).
In *Zehe et al.* [2001], $f_m$ was defined as the ratio of the saturated water flow rate in all
macropores in a model element to the corresponding value in soil matrix, which can be
determined based on the density and length of fractures at small scales. In addition, $f_m$ has
also been considered as a calibration parameter previously [e.g., *Blume*, 2008; *Zehe et al.*,
2013]. In this study, we define $f_m$ as a characteristic soil property reflecting the influence of
fractures on soil water movement [*Zehe and Blöschl*, 2004], and estimate it from the relative
difference of permeability between chalk matrix and fractured chalk system that can be of the
order $10^4$-$10^6$ according to *Price et al.* [1999]. Consequently, we consider a macroporosity
factor of $f_m = 10^5$ for the uncalibrated BC model.
In the following step, the BC model parameters are optimized to minimize the differences
between the variability of observed and simulated soil moisture. Price et al. [1999] argued
that the $K_s$ for chalk matrix is generally around 3-5 mmd$^{-1}$ (3.5-5.8x10$^{-5}$ mms$^{-1}$). In order to
optimize the BC model performance, we consider a range of $K_s = 0.8$-86 mmd$^{-1}$ ($10^{-5}$-$10^{-3}$
mms$^{-1}$) for chalk matrix in this study. As mentioned earlier, $S$ is zero for completely dry soils
and one in case of fully wet soils. Therefore, we consider a range of 0-1.0 for $S_0$ to optimize
the BC model. For $f_m$, a range of $10^4$-$10^6$ is considered, which, as discussed earlier, is
consistent with the relative difference between the permeability of fractured chalk and chalk
matrix according to *Price et al* [1999].
We use the Root Mean Squared Error (RMSE) as the objective function to optimize the BC
model parameters [e.g., *Ireson et al.*, 2009]
$$RMSE = \frac{1}{nd}\sum_{1}^{nd}\sqrt{\left(\frac{1}{nt-1}\sum_{2}^{nt}\left(\Delta\theta_{d,t}^{obs} - \Delta\theta_{d,t}^{sim}\right)^2\right)} \qquad (2)$$

where *nd* is the number of soil layers, *nt* is the number of soil moisture observations available
for a layer *d*, $\Delta\theta^{obs}$ is the observed variability of soil moisture and $\Delta\theta^{sim}$ is the simulated
variability of soil moisture. Note that we consider $\Delta\theta$ for this optimization because of its
relevance to the water flux and recharge through chalk unsaturated zone [e.g., *Ireson and*
*Butler*, 2011]. Latin hypercube technique [e.g., *McKay et al.*, 2016] is used to generate 2000
random samples for each BC model parameter within the respective range discussed above.
We perform simulations using these random samples and calculate model performance
(Equation 2) to select the optimum parameter values for the BC model (discussed in section

152 4.1).

In *Zehe et al.* [2001], $f_m$ was defined as the ratio of the saturated water flow rate in all
macropores in a model element to the corresponding value in soil matrix, which can be
determined based on density and length of fractures at small scales. In addition, $f_m$ has also
been considered as a calibration parameter previously [e.g., *Blume*, 2008; *Zehe et al.*, 2013].
In this study, we define $f_m$ as a characteristic soil property reflecting the influence of fractures
on soil water movement [*Zehe and Blöschl*, 2004], and estimate it from the relative difference
of permeability between chalk matrix and fractured chalk system that can be of the order $10^5$
according to *Price et al.* [1999]. Consequently, we consider a macroporosity factor of $f_m =$
$10^5$ in this study.

## 3. Methods

### 3.1. Study area

The study area encompasses the Kennet catchment located in the Southern England with an
area of about 1033 km$^2$ (Figure 1a). Generally, Kennet, in general, is rural in nature with
scattered settlements and has a maximum altitude of approximately 297 m (Above Ordnance
Level). The River Kennet discharges into the North Sea through London. The major Major
tributaries of this river are Lambourn, Dun, Enborne, and Foudry Brook. An average annual
rainfall of approximately 760 mm was recorded in the catchment over a 40 year period from

170    1961-1990.

Solid geology of the Kennet catchment is dominated by chalk, which is overlain by thin soil
layer. While lower chalk outcrops along the northern catchment boundary, progressively
younger rocks are found in the southern part. In general, surface runoff production is very
limited over the regions of the catchment where chalk outcrops. The flow regime shows a
distinct characteristics of slow response to groundwater held within the chalk aquifer [*Le*
*Vine et al.*, 2016]. According to *Ireson and Butler* [2013], the unsaturated zone of chalk
shows slow drainage over summer and bypass flow during wet periods in this catchment.
**3.2. Field measurements and remotely sensed data**
Table 1 summarizes the field measurements and remote sensing data used in this study. We
use *in-situ* soil moisture and runoff measurements along with remotely sensed LE ~~latent heat~~
~~flux (*LE*)~~ data to assess model performance in simulating the mass and energy balance
components of the hydrological cycle. Point scale soil moisture measurements at two
adjacent sites (~20 m apart) at the Warren Farm (Figure 1) were provided by Centre for
Ecology and Hydrology (CEH). A Didcot neutron probe was used at these locations to
measure fortnightly soil moisture at different depths below land surface (10 cm apart down to
0.8 m, 20 cm apart between 0.8-2.2 m, and 30 cm apart between 2.2-4.0 m) [*Hewitt et al.*,
2010].
The National River Flow Archive (NRFA) coordinates discharge measurements from the
gauging station networks across UK. These networks are operated by the Environmental
Agency (England), Natural Resources Wales, ~~the~~ Scottish Environment Protection Agency,
and Rivers Agency (Northern Ireland). We use discharge measurement provided by NRFA to
calculate the runoff ratio over the Kennet catchment in this study.
The MOD16 product of the Moderate Resolution Imaging Spectroradiometer (MODIS) is a
part of NASA/EOS project that provides estimation of global terrestrial LE. The LE
estimation from MOD16 is based on remotely sensed land surface data [e.g., *Mu et al.*, 2007].
In this study, the 8-day and monthly LE data products from MODIS is used to evaluate the
model's performance in simulating land surface energy fluxes.

**3.3. Land surface model**

In this study, we use the Joint UK Land Environment Simulator (JULES [e.g., *Best et al.*,
2011; *Clark et al.*, 2011]) version 4.2. JULES is a flexible modelling platform with a modular
structure aligned to various physical processes developed based on the Met Office Surface
Exchange Scheme (MOSES [e.g., *Cox et al.*, 1999; *Essery et al.*, 2003]). Meteorological data
including precipitation, incoming short- and longwave radiation, temperature, specific
humidity, surface pressure, and wind speed are required to drive JULES. Each grid box in
JULES can comprise nine surface types (broadleaf trees, needle leaf trees, C3 grass, C4 grass,
shrubs, inland water, bare soil, and ice) represented by respective fractional coverage. Each
surface type is represented by a tile and a separate energy balance is calculated for each tile.
Subsurface heat and water transport equations are solved based on finite -difference
approximation in JULES as described in *Cox et al.* [1999]. Moisture transport in the
subsurface is described by the finite difference form of Richards' equation. The vertical soil
moisture flux is calculated using the Darcy's law. While the top boundary condition to solve
the Richards' equation is infiltration at soil surface, the bottom boundary condition in JULES
is free drainage that contributes to subsurface runoff.
Surface runoff is calculated by combining the equations of throughfall and grid box average
infiltration in JULES. In order to direct the generated runoff to a channel network, river
routing is implemented based on the discrete approximation of one-dimensional kinematic
wave equation [e.g., *Bell et al.*, 2007]. In this approach, river network is derived from the
digital elevation model (DEM) of the study area and different wave speeds are applied to
surface and subsurface runoff components and channel flows [e.g., *Bell and Moore*, 1998]. A
return flow term accounts for the transfer of water between subsurface and land surface [e.g.,
*Dadson et al.*, 2010, 2011].
**3.4. Model configurations and input data**
*3.4.1. Point scale*
At the point scale, JULES is configured to simulate the mass and energy fluxes at Warren
Farm (Figure 1). A total subsurface depth of 5 m is considered in the model with a vertical
discretization ranging from 10 cm at the land surface to 50 cm at the bottom of the model
domain. Note that this discretization is consistent with the soil moisture measurement depths
mentioned in section 3.2. The vegetation type is implemented as C3 grass using the default
parameters in JULES. The soil hydraulic properties are estimated based on texture (Table 2),
which is predominantly loamy at Warren Farm. The saturation-pressure head relationship is
described using the Van Genuchten [*Van Genuchten*, 1980] model with parameter values
(Table 2) obtained from *Schaap and Leij* [1998] in the model.
Point scale simulations were performed over 2 consecutive years from 2003-2005 at an
hourly time step. Except for precipitation, hourly atmospheric forcing data to drive JULES
was obtained from an automatic weather station operated by the CEH at Warren Farm. In
order to estimate hourly precipitation data to run JULES, rain gauge measurements by the
Met Office [*Met Office*, 2006] were used. Inverse distance interpolation technique [e.g.
*Garcia et al.*, 2008; *Ly et al.*, 2013] was applied on rainfall measurements from 13 gauges
closest to Warren Farm (distance varies from 25-60 km) to obtain hourly precipitation for the
point scale simulations.
*3.4.2. Catchment scale*
At the catchment scale, JULES is configured over the study area (Figure 1) with a uniform
lateral grid resolution of 1 km with 70 x 40 cells in x and y dimensions, respectively. The
vertical discretization is identical to that of the point scale simulations described in the
previous section. Spatially distributed vegetation type information for the study area (Figure
1b) is obtained from the Land Cover Map 2007 (LCM2007) dataset [e.g., *Morton et al.*,
2011]. Harmonized World Soil Database (HWSD) from the Food and Agricultural
Organization of UNO (FAO) is used to obtain the texture of different soil types in the region
(Figure 1c). Van Genuchten model, with parameter values (Table 2) obtained from *Schaap*
*and Leij* [1998] is used to represent the saturation-pressure head relationship for different soil
types, which is identical to the point scale simulations.
Simulations were performed over 5 consecutive years from 2006-2011 at the catchment scale.
Note that the simulation periods of catchment and point scale (2003-2005) does not coincide
due to the availability of soil moisture measurements described in section 3.2. Spatially
distributed meteorological data from the Climate, Hydrology, and Ecology research Support
System (CHESS) was used to obtain the atmospheric forcing to drive JULES. The CHESS
data includes 1 km resolution gridded daily meteorological variables [*Robinson et al.*, 2015].
This daily data is downscaled using a disaggregation technique described in *Williams and*
*Clark* [2014] to obtain hourly atmospheric forcing. The flow direction required for river
routing is extracted from the USGS HydroSHEDS digital elevation data [*Lehner et al.*, 2008].
**3.5. Setup of numerical experiments**
We consider two different model configurations, namely, *default* and *macro* (Figure 2), to
explore the influence of chalk hydrology on simulated land surface processes in JULES. The
*default* configuration corresponds to the standard parameterizations of JULES that does not
represent chalk hydrology in the model. In this configuration, each soil column in JULES is
considered to be vertically homogeneous with the soil properties defined in Table 2, which is
motivated by the Met Office JULES Global Land 4.0 configuration described in *Walters et*
*al.* [2014]. The *macro* configuration, in contrast, explicitly represents chalk hydrology in the
model. The *macro* setup modifies the *default* configuration by applying chalk hydraulic
properties (Table 3) from 30 cm below land surface to the bottom of the model domain (i.e.
500 cm). The BC model is applied in the chalk layers (30-500 cm) to simulate water flow in
the *macro* configuration. Therefore, soil columns in the model can be divided into topsoil (0-
30 cm) and chalk (30-500 cm) in *macro*. Note that except for this inclusion of chalk, *default*
and *macro* configurations are identical in terms of model set up and input data.
The topsoil depth of 30 cm is defined based on several augured soil samples collected during
a field campaign at Warren Farm in 2015 (Figure 2). This depth is corroborated by additional
information from the British Geological Survey (BGS) operated borehole records
(http://www.ukso.org/pmm/soil_depth_samples_points.html), which show that topsoil depths
vary from 10-40 cm over the study area. We therefore apply the *macro* configuration
assuming a spatially homogeneous 30 cm topsoil depth for both point and catchment scale
simulations.
In this study, simulations are performed at two distinct spatial scales, namely point and
catchment. At the point scale, JULES is configured to simulate the mass and energy fluxes at
the Warren Farm site (Figure 1a). A total subsurface depth of 5 m is considered in the model
with a vertical discretization ranging from 10 cm at the land surface to 50 cm at the bottom of
the model domain.  Note that this discretization is consistent with the soil moisture
measurement depths mentioned in section 3.2. The vegetation type is implemented as C3
grass using the default parameters in JULES. Point scale simulations were performed over 2
consecutive years from 2003-2005 at an hourly time step. Except for precipitation, hourly
atmospheric forcing data to drive JULES was obtained from an automatic weather station
operated by the CEH at Warren Farm. In order to estimate hourly precipitation data to run
JULES, rain gauge measurements from the Met Office [*Met Office*, 2006] were used. Inverse
distance interpolation technique [e.g. *Garcia et al.*, 2008; *Ly et al.*, 2013] was applied on
rainfall measurements from 13 gauges closest to Warren Farm (distance varies from 25-60
km) to obtain hourly precipitation for the point scale simulations.
At the catchment scale, JULES is configured over a study area encompassing the Kennet
catchment (Figure 1a) considering a uniform lateral grid resolution of 1 km with 70 x 40 cells
in x and y dimensions, respectively. The total subsurface depth and vertical discretization are
identical to those of the point scale simulations. Spatially distributed vegetation type
information for the study area (Figure 1b) is obtained from the Land Cover Map 2007
(LCM2007) dataset [*Morton et al.*, 2011]. Simulations were performed over 5 consecutive
years from 2006-2011 at the catchment scale. Note that the simulation periods of catchment
and point scale (2003-2005) does not coincide due to the availability of soil moisture
measurements described in section 3.2. Spatially distributed meteorological data from the
Climate, Hydrology and Ecology research Support System (CHESS) was used to obtain the
atmospheric forcing to drive JULES at the catchment scale. The CHESS data includes 1 km
resolution gridded daily meteorological variables [*Robinson et al.*, 2015]. This daily data is
downscaled using a disaggregation technique described in *Williams and Clark* [2014] to
obtain hourly atmospheric forcing. The flow direction required for river routing is extracted
from the USGS HydroSHEDS digital elevation data [*Lehner et al.*, 2008].
We estimate the soil hydraulic properties based on texture (Table 2). At the point scale, loam
soil is dominant at the Warren Farm site. At the catchment scale, the Harmonized World Soil
Database (HWSD) from the Food and Agricultural Organization of UNO (FAO) is used to
obtain the texture of different soil types over Kennet (Figure 1c). The saturation-pressure
head relationship for different soil types is described using the Van Genuchten [*Van*
*Genuchten*, 1980] model with parameter values (Table 2) obtained from *Schaap and Leij*
[1998].
Table 3 summarizes the hydraulic properties for chalk used in this study. These properties are
obtained based on existing literature as a first step when evaluating the uncalibrated BC
model. The BC model parameters are subsequently optimized to minimize the differences
between observed and simulated $\Delta\theta$.
In this study, we consider two different model configurations, namely *default* and *macro*
(Figure 2). The *default* configuration corresponds to the standard parameterizations of JULES
that does not represent chalk hydrology in the model. In this configuration, each soil column
in JULES is considered to be vertically homogeneous with the soil properties defined in
Table 2, which is motivated by the Met Office JULES Global Land 4.0 configuration
described in *Walters et al.* [2014]. The *macro* configuration, in contrast, explicitly represents
chalk by applying the BC model starting at 30 cm below land surface to the bottom of the
model domain (i.e. 500 cm). Therefore, the soil column in the *macro* configuration can be
divided into topsoil (0-30 cm) and chalk (30-500 cm) in *macro*. Note that except for this
inclusion of chalk, *default* and *macro* configurations are identical in terms of model set up
and input data.
The topsoil depth of 30 cm in the *macro* configuration is defined based on several augered
soil samples collected during a field campaign at Warren Farm in 2015 (Figure 2). This depth
is corroborated by additional information from the British Geological Survey (BGS) operated
borehole records (http://www.ukso.org/pmm/soil_depth_samples_points.html), which show
that topsoil depths vary from 10-40 cm over the study area. We therefore apply the *macro*
configuration assuming a spatially homogeneous topsoil depth of 30 cm for both point and
catchment scale simulations.

## 4. Results and discussion

### 4.1. Point scale simulations

At the point scale, the simulation results are evaluated using soil moisture observations at the
Warren Farm site. Figure 3a compares observed and simulated soil moisture ($\theta$) from the
*default* and *macro* configurations at 2 m below land surface. Note that the *macro*
configuration uses the chalk hydraulic parameters collected from existing literature (Table 3).
This figure shows that the *default* configuration underestimates $\theta$ throughout the simulation
period, which is improved remarkably in case of *macro*. Figure 3b plots observed and
simulated soil moisture variability ($\Delta\theta$) from the *default* and *macro* configurations at the
Warren Farm site. In general, both configurations show discrepancies with observed $\Delta\theta$ with
*macro* showing relatively better model performance.
The results show that despite the *macro* configuration improves simulated $\theta$, it shows
considerable discrepancies with observed $\Delta\theta$, which is consistent throughout the whole chalk
profile (results from other model layers are not shown). In order to minimize the differences
between observed and modelled $\Delta\theta$ from the *macro* configuration, we optimize the BC model
following the methodology described in section 2. The optimization results are summarized
in Figure 4. Note that for each combination considered in the optimization, 2000 model runs
were performed using randomly sampled parameters as discussed in section 2. Figure 4
presents the results from the model runs yielding the lowest RMSE.
The RMSE between observed and simulated $\Delta\theta$ for the model configurations considered in
the optimization is shown in Figure 4a. This figure illustrates that the RMSE of the *default*
configuration is larger than that of *macro*, indicating better model performance in
reproducing $\Delta\theta$ for the latter. Therefore, it appears that the uncalibrated BC model (i.e., the
*macro* configuration) is better in reproducing soil moisture variability compared to ~~the~~ *default*
~~configuration.~~ Figure 4b, c and d presents the BC model parameter values from the model run
producing the lowest RMSE for each configuration. Concerning single BC model parameters,
Figure 4a shows that optimizing $S_0$ results in a 16% reduction of RMSE compared to the
*macro* configuration. Optimizing $K_s$ marginally improves model performance, which is
observed from a slightly lower (4%) RMSE than *macro*. Optimizing both $K_s$ and $S_0$
simultaneously results in the largest reduction (24%) of RMSE compared to *macro*.
Additionally, Figure 4 suggests that the sensitivity of $S_0$ on the model performance in
simulating $\Delta\theta$ is substantially higher compared to $K_s$ and $f_m$, which is corroborated by the
sensitivity of the individual model parameters (Figure S2). Figure 4a also reveals the
interesting fact that the RMSE from the configuration with optimized $K_s$ and $S_0$ is identical to
that of the one ~~the one~~ with ~~optimized~~ all 3 parameters optimized simultaneously (i.e., $K_s$, $S_0$
and $f_m$). Therefore, we select the *macro* configuration with optimized $K_s$ and $S_0$ (*macro_opt*
hereafter) to simulate chalk hydrology over the study area.
~~Figure 3 shows observed and simulated volumetric soil moisture from the *default* model~~
~~configuration at Warren Farm from 2003-2005. This figure shows that simulated soil~~
~~moisture at shallow soil layers (up to 50 cm) compares reasonably well with the observed~~
~~data. However, in the deeper layers, the model considerably underestimates soil moisture.~~
~~Figure 4 compares observed and simulated volumetric soil moisture from the *macro*~~
~~configuration at Warren Farm over the simulation period. This figure shows that especially in~~
~~the deeper soil layers, the agreement between observed and simulated soil moisture improves~~
~~remarkably relative to the *default* configuration throughout the simulation period. Notice~~
~~again that the *default* and *macro* configurations are identical in terms of model setup and~~
~~inputs except for the consideration of chalk. Therefore, the differences in soil moisture~~
~~simulations between the two model configurations can be attributed to the representation of~~
~~chalk hydrology in JULES.~~
Figure 5 compares $\Delta\theta$ from the *macro$_{opt}$* configuration ($\Delta\theta_{opt}$) with observed soil moisture
variability ($\Delta\theta_{obs}$). As mentioned earlier, $\Delta\theta_{default}$ and $\Delta\theta_{macro}$ show considerable discrepancies
with $\Delta\theta_{obs}$ while the *macro* configuration exhibits relatively better performance. Figure 5
illustrates that the overall agreement between observed and simulated $\Delta\theta$ improves
substantially in case of *macro$_{opt}$* compared to *default* and *macro*, which is pronounced
especially in the deeper chalk layers. Therefore, this figure indicates that the performance of
the BC model in simulating $\Delta\theta$ is further improved by optimizing the $K_s$ and $S_0$ parameters at
the Warren Farm site.
In order to assess the model performance in simulating soil moisture over the entire ~~chalk~~
~~profile~~column, ~~Figure 5 presents t~~Tthe relative bias ($\Delta\mu$, see Appendix) of simulated ~~soil~~
~~moisture~~$\theta$ from the *default* and *macro$_{opt}$* ~~two model~~ configurations at Warren Farm for
various depth ranges are shown in Figure 6. In the soil layers (0-30 cm), $\theta$ from the two
configurations are comparable with ~~both~~ the *default* ~~and *macro*~~ ~~configurations~~ showing
slightly lower mean relative bias ($\Delta\mu_{mean}$) of -0.03 ~~reproduces soil moisture reasonably well~~
~~with the latter showing slightly better~~ than *macro$_{opt}$* ($\Delta\mu_{mean}$ = -0.09). ~~agreement with~~
~~observations.~~ However, in the chalk layers (30-500 cm), *default* ~~fails to reproduce the soil~~
~~moisture dynamics efficiently,~~ simulat~~ing~~es substantially drier~~dry~~ conditions, corresponding
to ~~which are observed from the mean relative bias ($\Delta\mu_{mean}$) of~~ $\Delta\mu_{mean}$ $\leq$~~>~~ -0.28~~0.28 for this~~
~~configuration~~. In contrast, the *macro$_{opt}$* ~~*macro*~~ configuration considerably ~~remarkably~~
improves the agreement between the simulated and ~~with the~~ observed $\theta$ ~~soil moisture profile~~
in the chalk layers with ~~the largest calculated~~ $\Delta\mu_{mean}$ =$\geq$ -0.05~~-0.02~~. Therefore, the results
indicate that the inclusion of the BC model in JULES improves the performance of overall
soil moisture simulation (both $\theta$ and $\Delta\theta$) at Warren Farm especially in the chalk layers.
The drainage flux through the bottom of soil column ($d_b$) of a land surface model can be
considered as the potential recharge flux to groundwater [e.g., *Sorensen et al.*, 2014].
~~Therefore, the inclusion of the BC model in JULES appears to improve the performance of~~
~~overall soil moisture simulation at Warren Farm especially in the chalk layers.~~Figure ~~6~~7
compares the daily sum of ~~drainage through the bottom of the soil column ($d_b$)~~ from the
*default* and *macro_opt* configurations at the Warren Farm site. The rainfall characteristics over
the study period is shown in Figure ~~6~~7a. In Figure ~~6~~7b, the *macro_opt* configuration shows
considerable $d_b$ during the colder months, while slow drainage prevails throughout the rest of
the year. In contrast, the *default* configuration shows relatively high $d_b$ in summer compared
to the colder months. In general, the recharge rate through chalk unsaturated zone during the
warmer periods of the year is lower than that in the winter months [*Wellings and Bell*, 1980;
*Ireson et al.*, 2009]. Therefore, the *macro_opt* configuration appears to be more consistent with
the recharge mechanism in chalk compared to *default*.
In this section, the BC model was evaluated at the point scale. The results showed that in
general, the *macro* configuration performs relatively better in simulating $\theta$ and $\Delta\theta$ compared
to *default*. In order to improve the model performance even further, parameter optimization
was performed to minimize the differences between observed and simulated $\Delta\theta$ at the point
scale. In the next sections, the optimized model (*macro_opt*) is evaluated at the catchment scale.
~~In order to explore the reason of the discrepancies between simulated soil moisture from the~~
~~two model configurations, Figure 6 shows $S$ and water flux ($w_f$) profiles along with drainage~~
~~through the bottom boundary ($d_b$) of *default* and *macro* for the entire simulation period.~~
~~Figure 6b plots the contours of daily accumulated $w_f$ through chalk (30-500 cm) over daily~~
~~average $S$ for the *macro* configuration ($S_{macro}$). Figure 6c shows $S$ ($S_{default}$) and $w_f$ through the~~
~~same profile for the *default* configuration. A comparison between Figure 6b and 6c reveals~~
~~that *default* is considerably drier compared to *macro* ($S_{default} < S_{macro}$) throughout the profile,~~

which is consistent with Figure 5. Figure 6b shows notable flux through the profile following

strong precipitation events (Figure 6a), indicating fast water flow through subsurface in the

*macro* configuration (especially in winter). The *default* configuration, on the other hand,

shows relatively slower movement of water in the subsurface (Figure 6c).

According to the BC model, fracture flow in chalk is activated in a soil grid if $S$ exceeds $S_\theta$

(defined as 0.80), which is achieved predominantly during winter following strong

precipitation events because of the prevailing wet conditions. Therefore, the activation of

fracture flow explains the fast water movement patterns after strong precipitation events

observed in Figure 6b. This result is consistent with *Ireson et al.* [2009], who showed that

fracture flow through chalk dominates at Warren Farm during wet periods. Compared to the

*macro* configuration, *default* does not show fast water flow to the deeper soil layers because

the latter does not represent the matrix-fracture flow nature of chalk in JULES.

Figure 6d compares daily sum of $d_b$ from the two configurations. The *macro* configuration

generally shows lower drainage compared to *default* with an exception in March 2003.

Because of the gravity drainage lower boundary condition, water flow through the bottom of

the model domain depends on $K_s$ at the deepest soil layer in JULES. In chalk (*macro*

configuration), $K_s$ at the deepest soil layer is smaller compared to *default* (loam soil)

especially when $S_\theta < 0.8$ (Equation 1), which explains the lower drainage flux in case of the

*Chalk* configuration. The reason of higher $d_b$ in *macro* compared to *default* in March 2003 is

the strong precipitation events (Figure 6a) causing considerable fracture flow and $S > 0.8$ at

the bottom of the model domain (Figure 6b).

Figure 6 outlines the differences in simulated subsurface processes by the two model

configurations. Fracture flow in chalk is activated according to the BC approach during wet

periods that allows recharge at deeper soil layers in *macro*, which is absent in case of the

*default* configuration. Moreover, the *default* configuration generally shows higher drainage
flux through the lower boundary compared to *macro*. The combination of relatively low
recharge and high drainage through lower boundary is the reason of the drier conditions
simulated by *default*. In contrast, the *macro* configuration is characterized by fast recharge at
the deeper soil layers through fractures and slow drainage through the bottom because of
considerably lower $K_s$ compared to *default*, which is the reason of relatively higher simulated
soil moisture by this configuration that compares well with observations.
Several previous studies have discussed the influence of root zone soil moisture on land
surface mass and energy balance components [e.g., *Wetzel and Chang*, 1987; *Chen and Hu*,
2004]. Therefore, the differences in soil moisture from two configurations discussed above
may affect the land surface mass and energy fluxes in the model. In order to investigate this
effect, Figure 7 shows the difference between daily average latent heat flux (*LE*) time series
from *default* and *macro* configurations ($LE_{default}$ and $LE_{macro}$, respectively) at Warren Farm
over the simulation period. This figure shows that the *default* configuration generally
simulates lower *LE* compared to *macro* especially in the warmer months of the year.
The underestimation of *LE* in Figure 7 can be attributed to the differences in simulated soil
moisture by the two configurations (Figure 3 and 4). In winter, abundant soil moisture is
available in both *default* and *macro* to meet the relatively low evapotranspiration (ET)
demand due to the prevailing energy-limited conditions. Therefore, Figure 7 shows negligible
differences between $LE_{default}$ and $LE_{macro}$ in winter. However, in summer, the discrepancies
between soil moisture from the two model configurations result in marked differences
between $LE_{default}$ and $LE_{macro}$ because of the increased ET demand, which is consistent with
previous studies [e.g., *Rahman et al.*, 2016].
~~In this section, subsurface and land surface processes simulated by *default* and *macro*~~
~~configurations are discussed at the point scale. The simulation results show notable~~
~~differences in soil moisture and *LE* from the two configurations. Because the only difference~~
~~between *default* and *macro* configurations is the representation of the chalk hydrology, it~~
~~appears that a consistent representation of chalk in JULES affects land surface processes via~~
~~subsurface hydrodynamics supporting our hypothesis. In the next section, we test this~~
~~hypothesis regionally by evaluating the mass and energy fluxes of the hydrological cycle at~~
~~the catchment scale.~~

## 4.2. Catchment scale simulations

In the previous section, it was observed that the *default* configuration generally
underestimates $\theta$ compared to *macro*$_{opt}$. Previous studies have demonstrated the
interconnections between shallow soil moisture and LE [e.g., *Chen and Hu*, 2004]. In order to
assess the differences between the LE from the *default* and *macro*$_{opt}$ configurations at the
catchment scale, Figure ~~7~~ 8 plots spatially averaged 8-day composites of LE from MODIS
(LE$_{MOD}$) against the LE from these configurations ~~e *default* and *macro*$_{opt}$ configurations~~
(LE$_{default}$ and ~~LE$_{macro}$~~ LE$_{opt}$, respectively) over ~~the~~ Kennet ~~catchment~~. ~~In this figure, t~~The
agreement between simulated LE and LE$_{MOD}$ is evaluated using the coefficient of
determination ($R^2$, see Appendix) and mean bias. Comparison between LE$_{default}$ and LE$_{MOD}$
shows a coefficient of determination of $R^2_{default} = 0.78$ and a mean bias of $bias_{default} = 10.5$
Wm$^{-2}$. The agreement between simulated LE and LE$_{MOD}$ improves in case of the *macro*$_{opt}$
~~*macro*~~ configuration, which is reflected by an increased coefficient of determination of
$R^2_{opt}$~~*macro*~~ = 0.81~~2~~ and a reduced mean bias of $bias_{opt}$~~of~~ = 3 Wm$^{-2}$.
Figure ~~8~~7 8 shows differences between $LE_{default}$ and $LE_{opt}$ ~~$LE_{macro}$~~ especially for relatively high
$LE$, indicating discrepancies especially during the warmer months of the year. Figure 9~~a~~
presents spatially averaged time series of monthly $LE_{MOD}$, $LE_{default}$ and $LE_{opt}$ ~~$LE_{macro}$~~. This
figure shows that the ~~negligible differences in *LE* from the two configurations during the~~
~~colder months of the year, while~~ differences between $LE_{default}$ and $LE_{opt}$ ~~$LE_{macro}$~~ increases
substantially in summer compared to the colder months of the year, which is consistent with
Figure 7~~8~~. Consequently, the *default* configuration underestimates *LE* ~~especially~~ in summer
compared to $LE_{MOD}$, which is improved ~~when chalk hydrology is explicitly considered in~~
~~JULES~~ in case of the $macro_{opt}$ ~~*macro*~~ configuration. In contrast, the differences between
$LE_{default}$ and $LE_{opt}$ ~~between~~ are negligible during the colder months of the year.
~~Figure 9b plots spatially averaged time series of daily $S_{default}$ and $S_{macro}$ over the Kennet~~
~~catchment. Note that average *S* at the first 8 vertical model layer (0-100 cm below land~~
~~surface) is presented in this figure, which highlights the difference in root zone moisture~~
~~content from the two model configurations. Figure 9b shows relatively lower *S* simulated by~~
~~the *default* configuration compared to $S_{macro}$. In JULES, *LE* depends on surface conductance~~
~~to evaporation, which is controlled by the mean soil moisture in the root zone. Therefore, the~~
~~differences in $S_{default}$ and $S_{macro}$ is consistent with the underestimation of *LE* by the *macro*~~
~~configuration (Figure 9a). Note that despite the differences in *S* between the two~~
~~configurations over the entire simulation period, Figure 9a shows significant *LE* differences~~
~~only in summer. This is due to the prevailing energy limited conditions during the colder~~
~~months over the region, which was discussed in the previous section. Figure 9 suggest that~~
~~representing chalk hydrology in JULES considerably influences simulated *LE* by modifying~~
~~shallow soil moisture at the catchment scale, also supporting our hypothesis.~~
Table 4 compares observed and simulated daily average runoff from the two model
configurations over the Kennet catchment from 2006-2011. The runoff ratio (*RR*, see
Appendix), which is equal to the mean volume of flow divided by the volume of precipitation
[e.g., *Kelleher et al.*, 2015], assesses the partitioning of precipitation into runoff over the
catchment. The *default* configuration ($RR = 0.82$) shows considerably higher $RR$ compared to
observation ($RR = 0.40$), indicating overestimation of runoff by the model. Including chalk
hydrology in the model remarkably improves the agreement between observed and simulated
mean runoff over the Kennet catchment, which is assessed from a runoff ratio of $RR = 0.37$~~8~~
for the *macro~opt~* ~~*macro*~~ configuration.
In Table 4, the relative bias ($\Delta\mu$) of 1.04 between observed and simulated runoff from the
*default* configuration again indicates the overestimation by the model. In comparison,
*macro~opt~* ~~*macro*~~ shows a relative bias ($\Delta\mu = $ −0.05 ~~0.07~~), indicating improvement between
observed and simulated mean runoff volume compared to *default*. The relative difference in
standard deviation ($\Delta\sigma$, see Appendix) compares the variability ~~magnitude~~ of observed and
simulated runoff in Table 4~~3~~. This comparison shows that the *default* configuration
overestimates the variability of runoff over the Kennet catchment ($\Delta\sigma = 2.04$), which is
improved in case of *macro* ($\Delta\sigma = $ ~~0.56~~0.70).

It was demonstrated previously that the *default* configuration predicts lower
evapotranspiration (ET) compared to *macro~opt~* over the Kennet catchment due to the
differences in simulated $\theta$. In JULES, moisture from soil and canopy is depleted to meet the
~~evapotranspiration (ET)~~ demand. Additionally, surface runoff generation depends on canopy
water storage in the model [*Best et al.*, 2011]. Because of this connection between ET and
surface runoff generation via canopy water storage, the differences in runoff demonstrated in
Table 4 can be attributed to the disagreements between $LE_{default}$ and $LE_{macro}$ (~~demonstrated in~~
Figure 8) due to the relatively drier conditions simulated by *default*.
~~In JULES, moisture from soil and canopy water storage is depleted to meet the ET demand.~~
~~Additionally, surface runoff generation depends on canopy water storage in the model [*Best*~~
*et al.*, 2011]. Because of this connection between ET and surface runoff generation via
canopy water storage, the differences in runoff demonstrated in Table 4 can be attributed to
the disagreement between $LE_{default}$ and $LE_{macro}$ demonstrated in Figure 9a. Therefore, it
appears that $LE$ in JULES is affected by the inclusion of chalk hydrology, which
consequently influences surface runoff generation corroborating our hypothesis.
In this section, the BC model is evaluated using observed mass and energy fluxes over the
Kennet catchment. The *default* configuration default configurations showed considerably low
LE over the catchment, which was pronounced during the warmer period of the year. The
agreement between observed and simulated LE was improved in case of the *macro_opt*
configuration compared to *default*. It was also observed that the overall runoff prediction was
also improved by *macro_opt* compared to *default*. Given its simplicity, our results indicate that
the proposed parameterization is suitable for use in land surface modelling applications.

## 5. Summary and Conclusions

In this study, we proposed a simple parameterization we hypothesized that a consistent
representation of chalk hydrology affects land surface mass and energy balance components
via subsurface hydrodynamics simulated by a land surface model. In order to support this
hypothesis, namely the the *Bulk Conductivity* (BC) model that to simulates water flow
through the matrix-fracture system of chalk in large scale land surface modelling
applications. This parameterization was implemented in the Joint UK Land Environment
Simulator (JULES) and. This model was applied to on the Kennet catchment located in the
southern UK to simulate the mass and energy fluxes of the hydrological cycle for multiple
years. Two model configurations, namely namely *default* and *macro* were considered with
the latter representing using the BC model to simulate chalk hydrology in JULES using the
BC model.
The proposed BC model is a single continuum approach of modelling preferential flow [e.g.,
*Beven and Germann*, 2013] that involves only 2 parameters, namely macroporosity factor ($f_m$)
and relative saturation threshold ($S_0$). Initially, these parameters along with the saturated
hydraulic conductivity of the chalk matrix were estimated from existing literature. Finally,
the BC model parameters were optimized to minimize the differences between observed and
simulated soil moisture variability. Our results indicated that $S_0$ is the most influential
parameter in the model when representing water movement through a soil-chalk column,
followed by the saturated hydraulic conductivity of chalk matrix while $f_m$ showed low
sensitivity. Hence, the parameterization is further improved by optimizing both saturated
hydraulic conductivity of chalk matrix and $S_0$ to minimize the differences between observed
and simulated soil moisture variability.
The simulation results were evaluated using observed mass and energy fluxes both at point
and catchment scales. The results demonstrated that the inclusion of the BC model in JULES
improves simulated soil moisture variability at the point scale compared to a model
configuration that does not represent chalk in the subsurface (i.e., the *default* configuration).
At the catchment scale, it was illustrated that the proposed parameterization improves
simulated latent heat flux and overall runoff compared to the *default* configuration. ~~The~~
~~discrepancies between the measured and simulated fluxes and states can be improved by a~~
~~comprehensive model calibration, which is out of the scope of this study and should be the~~
~~subject of future research.~~
Note that the complexity of the BC model for simulating water flow through chalk
unsaturated zone is substantially lower compared to more commonly used models for this
purpose (e.g., dual-porosity models). Despite its simplicity, it appears that the proposed
parameterization improves mass and energy fluxes simulated by JULES over the Kennet
catchment. As mentioned previously, representing chalk hydrology in land surface models

using the dual-porosity concept is complicated mainly due to the relatively large number of parameters involved in such approach. Therefore, the simplified parameterization proposed in this study may be useful for large-scale land surface modelling applications over chalk-dominated areas.

The results showed that JULES generally underestimates root zone soil moisture without a consistent representation of chalk hydrology. Consequently, *LE* is underestimated by the model without chalk representation. The effect of chalk hydrology was also observed on runoff, which was attributed to the interconnection between *LE* and runoff generation in the model. Therefore, representing the matrix-fracture flow nature of chalk in a land surface model affects land surface processes via shallow soil moisture dynamics, which supports the proposed hypothesis.

*Habtes et al.* [2010] argued that flood flow in chalky catchments is influenced by the hydrological processes in the unsaturated zone. Implementing the BC model in JULES, this study showed that representing chalk hydrology significantly affects subsurface and land surface mass and energy fluxes. Therefore, the matrix-fracture flow nature of the aquifer may be important to consider in flood forecasting in chalk-dominated catchments.

*Leeper et al.* [2011] discussed the influence of shallow soil moisture on simulated atmospheric processes over karst landscapes because of the subsurface-land surface connection in the terrestrial system. In this study, we demonstrated that considering chalk hydrology considerably affects land surface mass and energy fluxes via subsurface hydrodynamics. This effect may be important to consider in numerical weather prediction models over the regions dominated by chalk because of the karst behaviour of chalk aquifers [e.g., *MacDonald et al.*, 1998; *Hartmann et al.*, 2014].

*Le Vine et al.* [2016] argued that the deep-groundwater system in a chalk-dominated
catchment may influence the mass and energy balance components of the hydrological cycle,
which is not considered in this study. The reason for that is JULES simulates water flow at
shallow subsurface considering free drainage lower boundary condition and does not allow
lateral movement of water between the soil columns. The effect of groundwater dynamics can
be represented in JULES by coupling a three-dimensional groundwater flow model [e.g., *Le*
*Vine et al.*, 2016; *Maxwell and Miller*, 2005], which will be addressed in future.

## Acknowledgements

We gratefully acknowledge the support by the "A MUlti-scale Soil moisture
Evapotranspiration Dynamics study – AMUSED" project funded by Natural Environment
Research Council (NERC) grant number NE/M003086/1. The authors would also like to
thank Ned Hewitt and Jonathan Evans from the Centre for Ecology and Hydrology (CEH) for
providing the data for the point-scale analyses at the Warren Farm. Finally, we would like to
thank Miguel Rico-Ramirez (University of Bristol) for helping preparing the precipitation
data from the rain gauge network used for the point-scale simulations, Thorsten Wagener
(University of Bristol) for his valuable suggestions on model diagnostics, and Joost Iwema
(University of Bristol) for helping with the soil samples collected during the 2015 field work
campaign.

**Appendix**
**Definition of Statistical Metrics**
Coefficient of determination ($R^2$) for observation $y = y1, \ldots, yn$ and prediction $f = f1, \ldots, fn$
is defined as
$R^2 = 1 - \dfrac{SS_{res}}{SS_{tot}}$
where, $SS_{res}$ is the residual sum of square and $SS_{tot}$ is the total sum of square. $SS_{res}$ and $SS_{tot}$
are defined as
$SS_{res} = \sum_{i=1}^{n}(y_i - f_i)^2$    and
$SS_{tot} = \sum_{i=1}^{n}(y_i - \bar{y})^2$    with $\bar{y}$ being the mean of y.
Runoff ratio (RR) assesses the portion of precipitation that generates runoff over the
catchment. RR is defined as
$RR = \dfrac{\mu_{runoff}}{\mu_{rain}}$
where $\mu_{runoff}$ is mean runoff and $\mu_{rain}$ is mean precipitation [e.g., *Kelleher et al.*, 2015].
Relative bias ($\Delta\mu$) between observed and simulated time series can be defined as
$\Delta\mu = \dfrac{\mu_{mod} - \mu_{obs}}{\mu_{obs}}$
where $\mu_{obs}$ and $\mu_{mod}$ are the mean of observed and simulated time series, respectively. While
the optimal value of $\Delta\mu$ is zero, negative (positive) values indicate an underestimation
(overestimation) by the model [e.g., *Gudmundsson et al.*, 2012].
Relative difference in standard deviation ($\Delta\sigma$) between observed and simulated time series
can be defined as
$\Delta\sigma = \dfrac{\sigma_{mod} - \sigma_{obs}}{\sigma_{obs}}$
where $\sigma_{obs}$ and $\sigma_{mod}$ are the standard deviation of observed and simulated time series,
respectively [e.g., *Gudmundsson et al.*, 2012].

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

**Tables**
Table 1. Field measurements and remote sensing data.

| Data | Spatial scale | Temporal extent | Frequency | Source |
|---|---|---|---|---|
| Soil moisture | Point[a] | 2003-2005 | 15 day | N. Hewitt (CEH) |
| Latent heat flux | Global | 2006-2011 | 8 day, 1 month | MODIS |
| Discharge | Point[b] | 2006-2011 | 1 day | NRFA |

[a]Measured at Warren Farm.
[b]Locations are shown in Figure 1a.

Table 2. Hydraulic properties for different soil types (refer to Figure 1c). Saturated hydraulic
conductivity ($K_s$) and porosity data are obtained from *Rawls et al.* [1982]. The Van Genuchten
parameters are acquired from *Schaap and Leij* [1998].

| Texture | $K_s$ (ms$^{-1}$) | Porosity (-) | $\alpha$ (m$^{-1}$) | n (-) |
|---------|------------------|--------------|---------------------|-------|
| Loam | 3.7x10$^{-6}$ | 0.463 | 3.33 | 1.56 |
| Silt loam | 2.0x10$^{-6}$ | 0.50 | 1.2 | 1.39 |
| Clay | 1.7x10$^{-7}$ | 0.475 | 2.12 | 1.2 |



~~Table 3. Hydraulic properties of chalk.~~

| ~~Properties~~ | ~~Value~~ | ~~Source~~ |
|------------|-------|--------|
| ~~$K_s$ (ms$^{-1}$)~~ | ~~1.85x10$^{-7}$~~ | ~~Price et al., 1993~~ |
| ~~Porosity (-)~~ | ~~0.40~~ | ~~Price et al., 1993~~ |
| ~~$\alpha$ (m$^{-1}$)~~ | ~~3.4~~ | ~~Le Vine et al., 2016~~ |
| ~~n (-)~~ | ~~1.4~~ | ~~Le Vine et al., 2016~~ |


Table 3. Hydraulic properties of chalk

| Properties | Unoptimized | | Optimized value |
|------------|-------------|--------|-----------------|
| | Value | Source | |
| $K_s$ (md$^{-1}$) | 16 | Le Vine et al., 2016 | 15 |
| $S_0$ (-) | 0.8 | Observations | 0.67 |
| $f_m$ (-) | 1x10$^5$ | Price et al., 1993 | 6.1x10$^5$ |
| $\alpha$ (m$^{-1}$) | 3.0 | Le Vine et al., 2016 | - |
| n (-) | 1.4 | Le Vine et al., 2016 | - |


Table 4. Comparison between observed and simulated daily average runoff from the two
configurations over the Kennet catchment.

| Metric | Observed | Simulated (*default*) | Simulated (*macro*) |
|--------|----------|----------------------|---------------------|
| *RR* | 0.40 | 0.82 | 0.37~~8~~ |
| $\Delta\mu$ | - | 1.04 | -0.05 ~~-0.07~~ |
| $\Delta\sigma$ | - | 2.04 | 0.70 ~~0.56~~ |




**Figures**


Figure 1. Location (a), vegetation cover (b), and soil texture (c) over the study area. The red
line in (a) outlines the Kennet catchment boundary, while the river network is shown in blue.
The black triangle in (a) shows the location of the discharge gauging station at the catchment
outlet while the black square corresponds to Warren Farm location where point-scale
simulations are carried out. The shaded area in (c) represents the location of chalk in the
catchment.

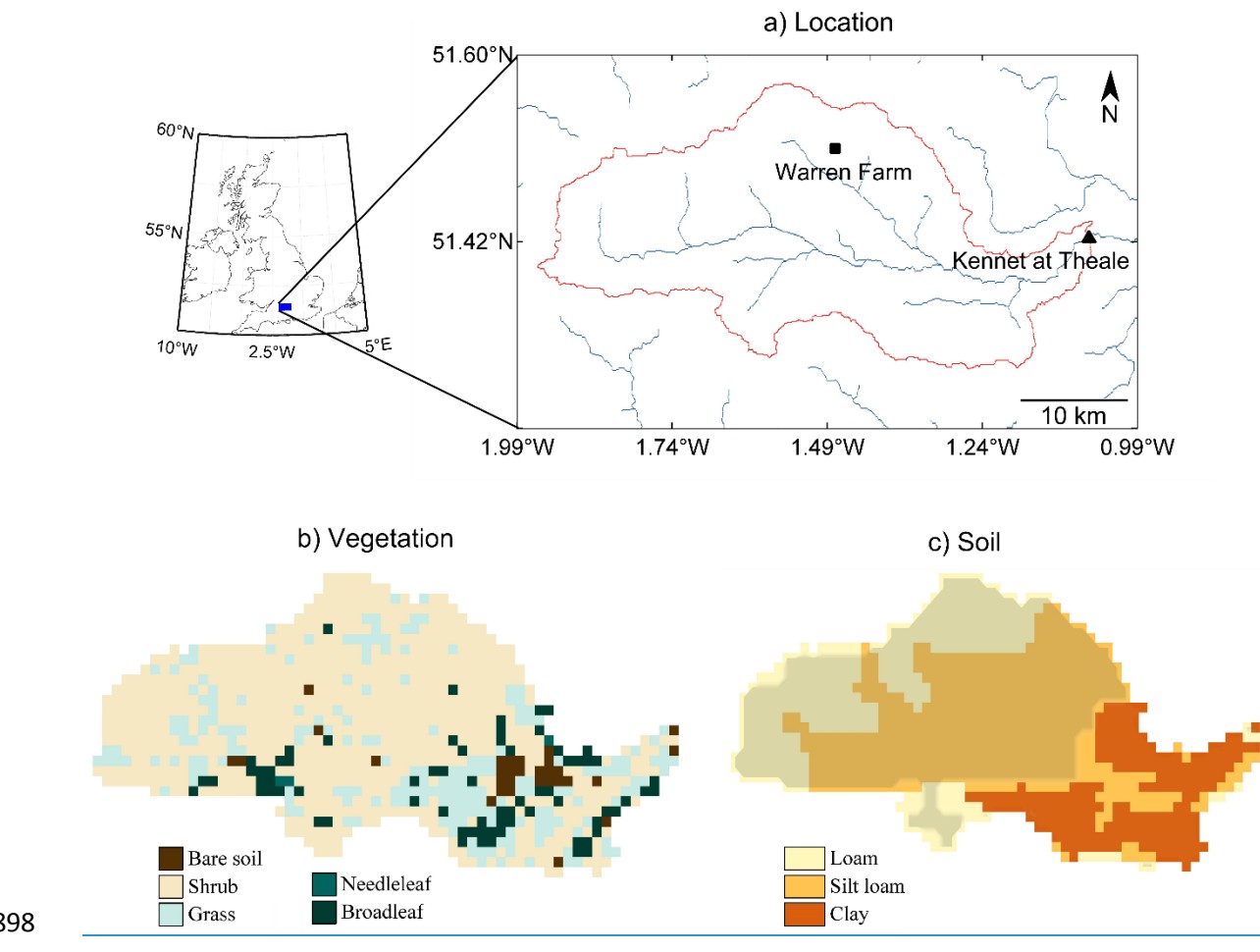


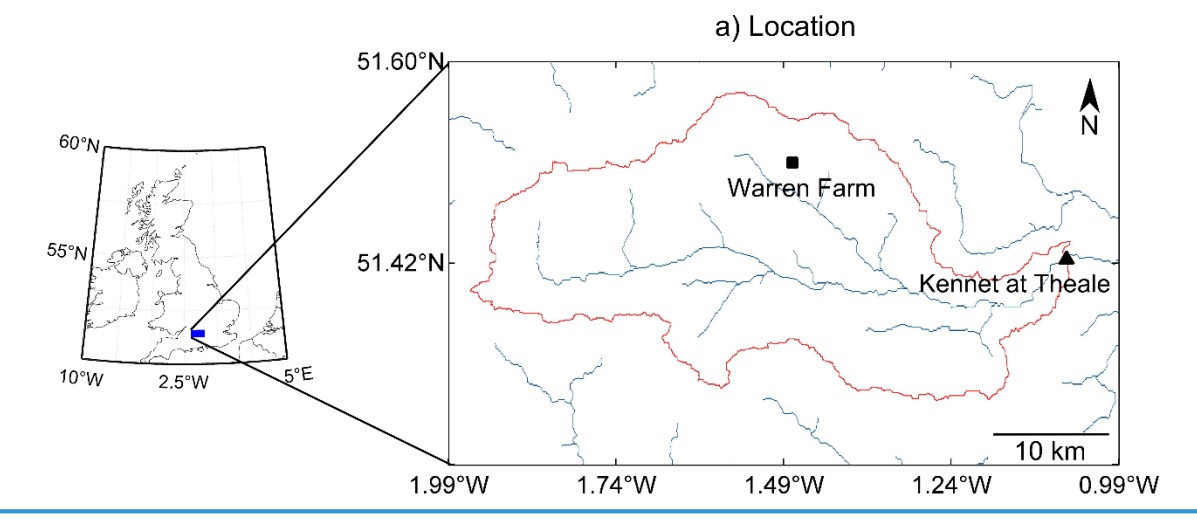

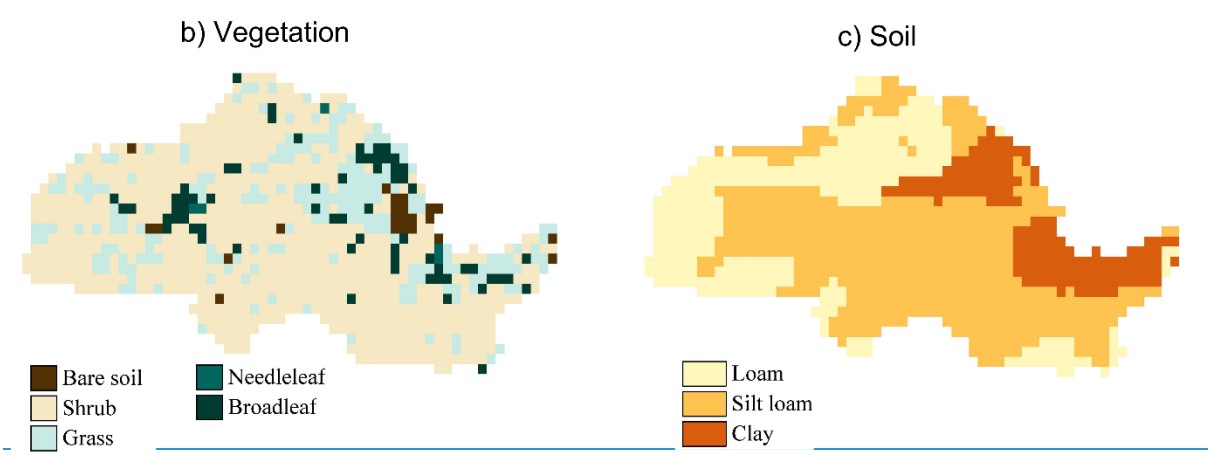

Figure 2. Example of soil profiles collected at Warren Farm during a field campaign in 2015
(a), and the two model configurations (b).

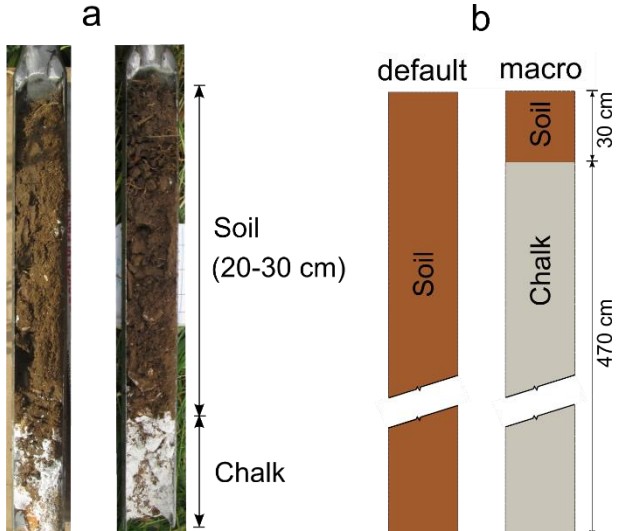













Figure 3. Comparison between observed and simulated (a) soil moisture (θ) and (b) change in
soil moisture (Δθ) from the *default* and *macro* configurations at a depth of 2m below land

surface. The shaded areas constructed from 2 soil moisture probes at the Warren Farm site

denote the range of observed data in these plots.

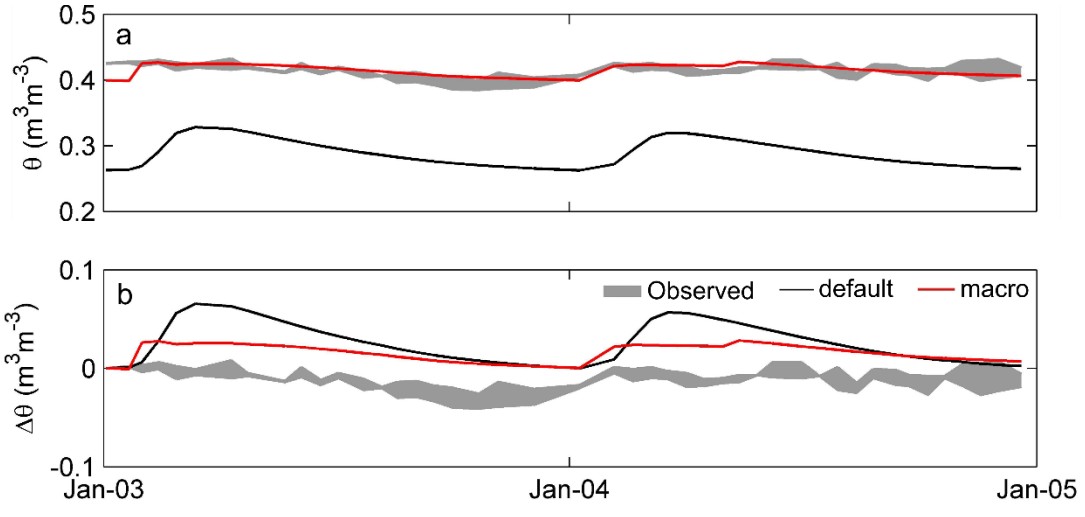












Figure 3. Observed and simulated (*default* configuration) volumetric soil moisture from
Warren Farm.

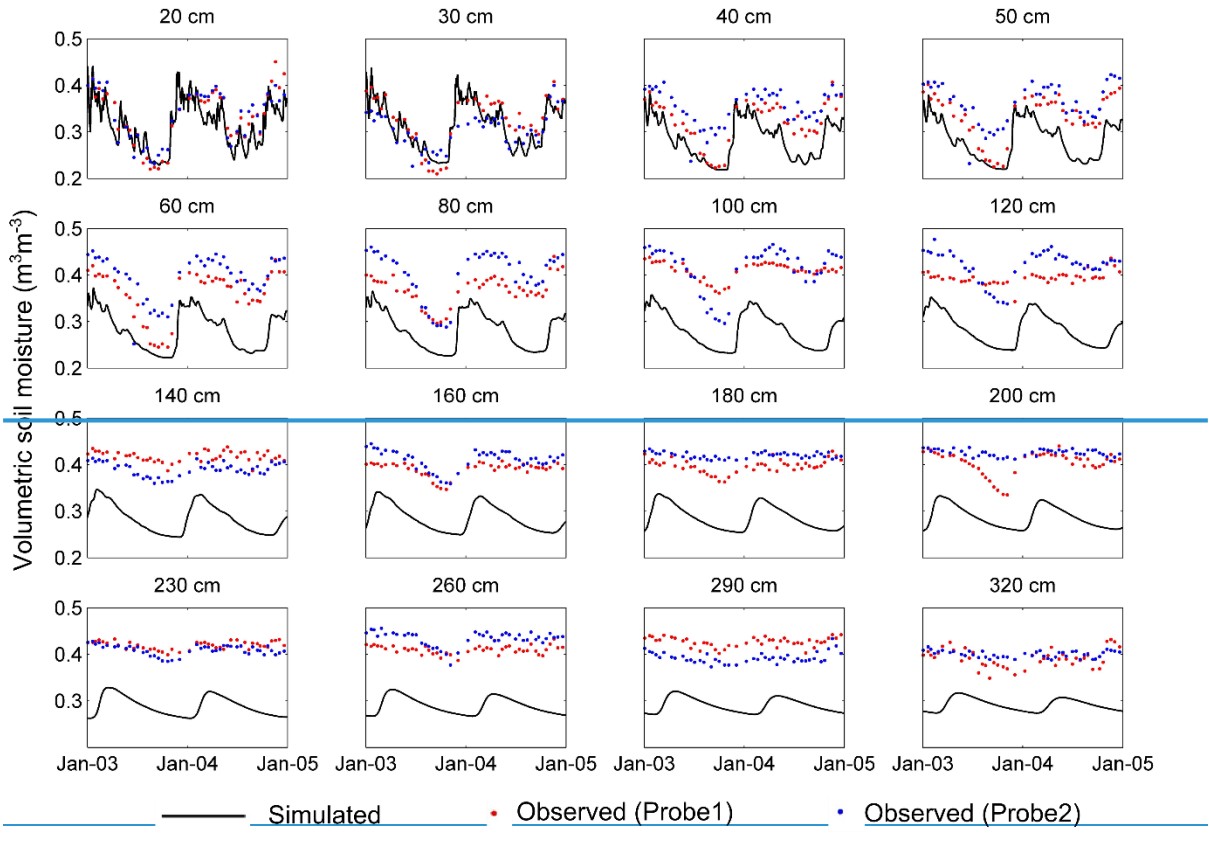

Figure 4. Observed and simulated (*macro* configuration) volumetric soil moisture from Warren Farm.

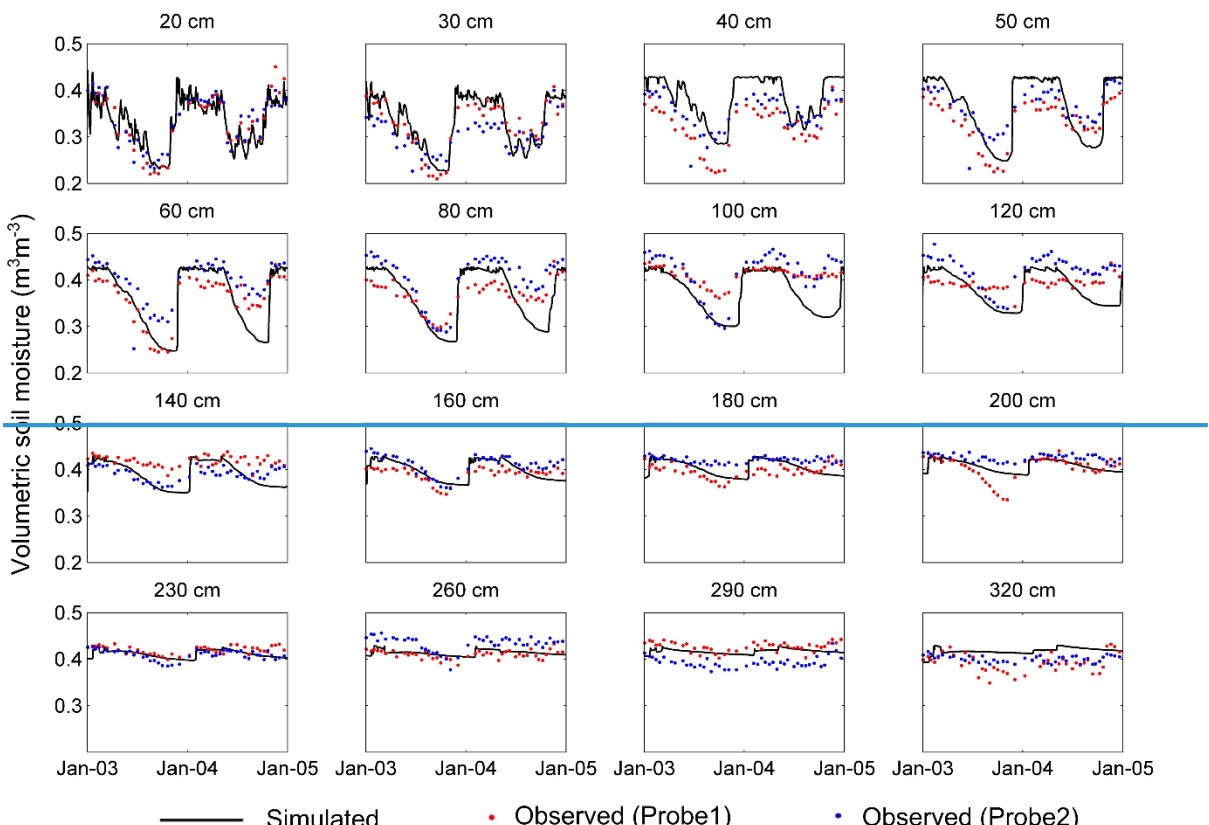

947 Figure 4. (a) Model performance in reproducing observed and simulated $\Delta\theta$, (b) $K_s$, (c) $S_0$ and

(d) $f_m$ for various parameter combinations considered in the optimization. Note that except for
the *default* and *macro*, the simulation yielding the lowest RMSE (out of 2000 model runs) is
presented in this plot.

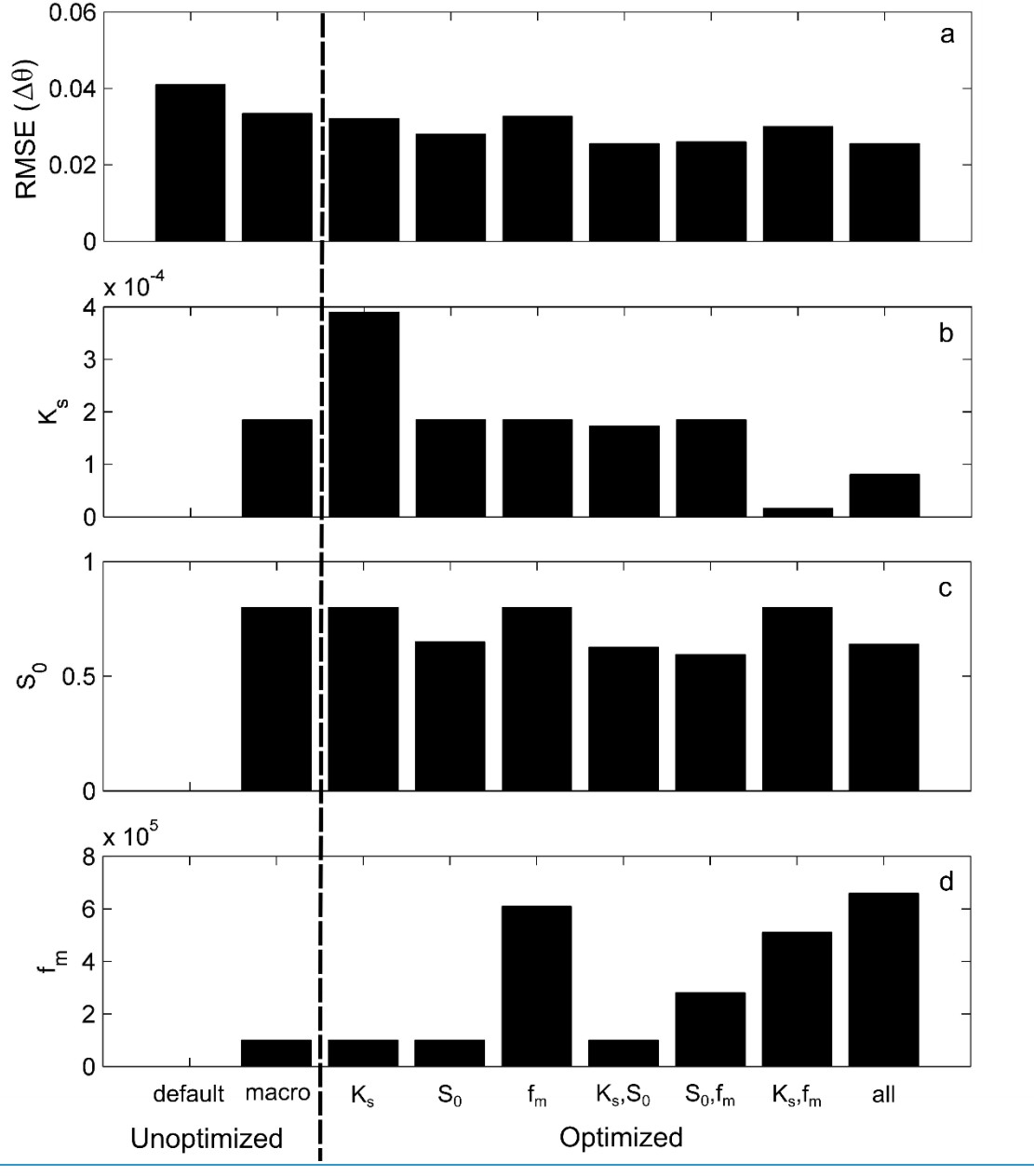








Figure 5. Comparison between observed and simulated $\Delta\theta$ from *default, macro* and *macro$_{opt}$*
configurations at various depths below land surface. The shaded area, which is constructed
from 2 soil moisture probes at the Warren Farm site, denotes the range of $\Delta\theta$.

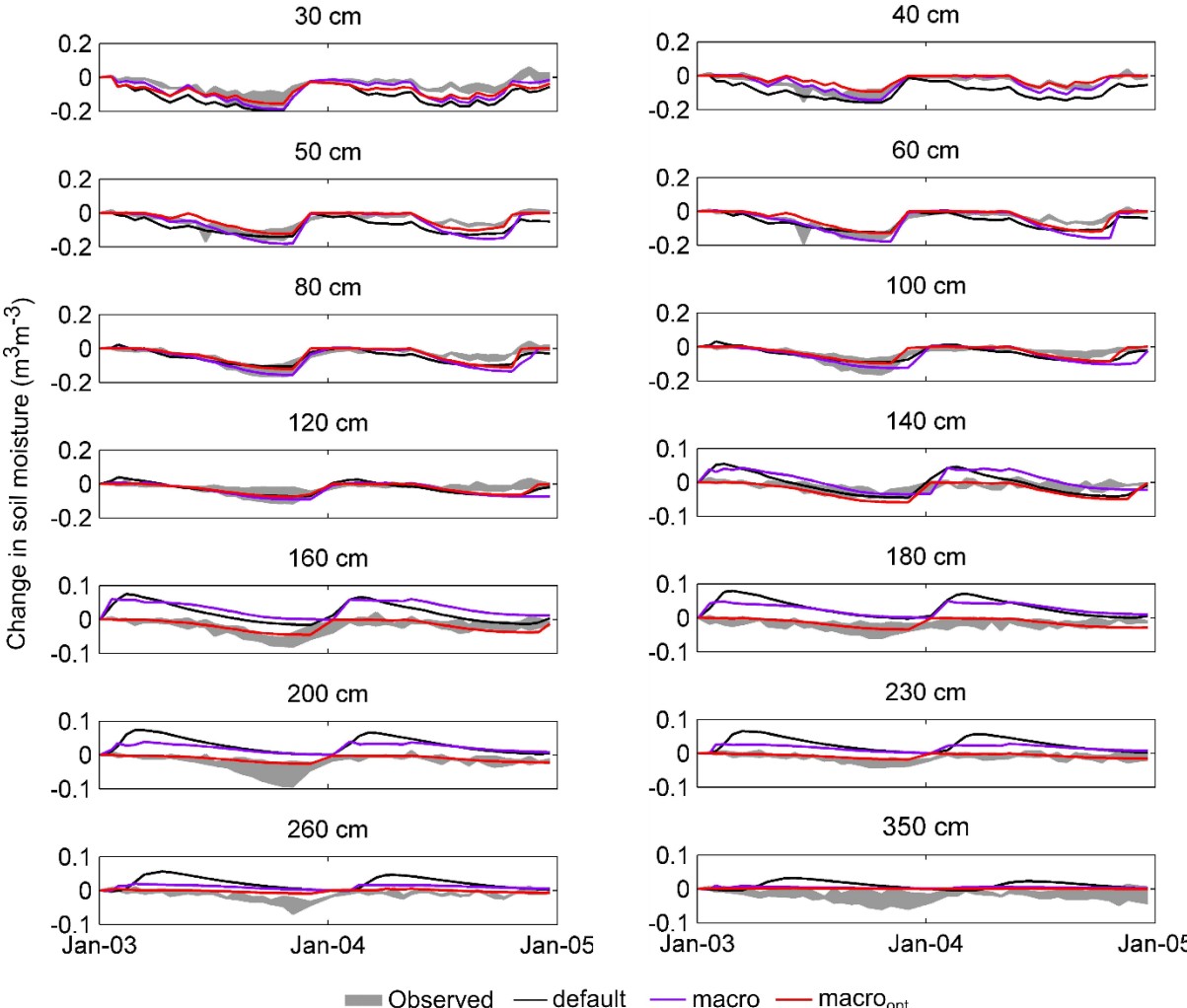







Figure 65. Box plot of relative bias ($\Delta\mu$) of simulated soil moisture from *default* and *macro*
configurations at different depth ranges shown in individual intervals (e.g., 0-30 cm, 30-100
cm, and so on).

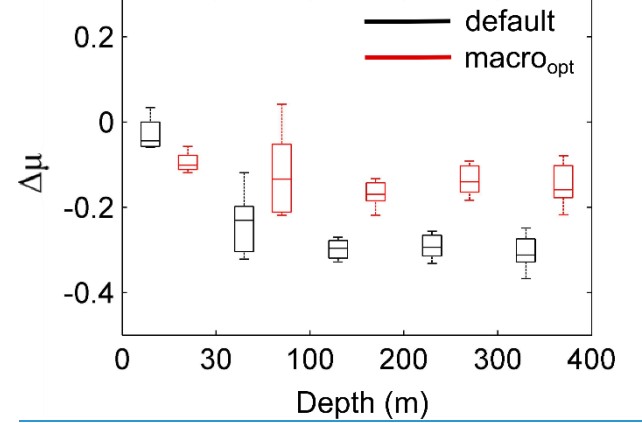


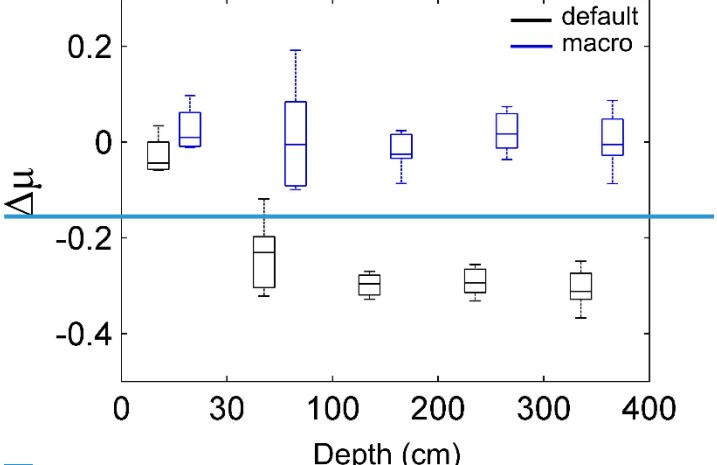













Figure 6. Precipitation (a), daily accumulated downward water flux ($w_f$, contour lines) plotted
over relative saturation ($S$, coloured shading) for *macro* (b), daily accumulated downward
water flux plotted over relative saturation for *default* (c), and daily accumulated drainage flux
through the bottom boundary simulated by the two model configurations (d) at Warren Farm
over the two simulated years (2003-2005).

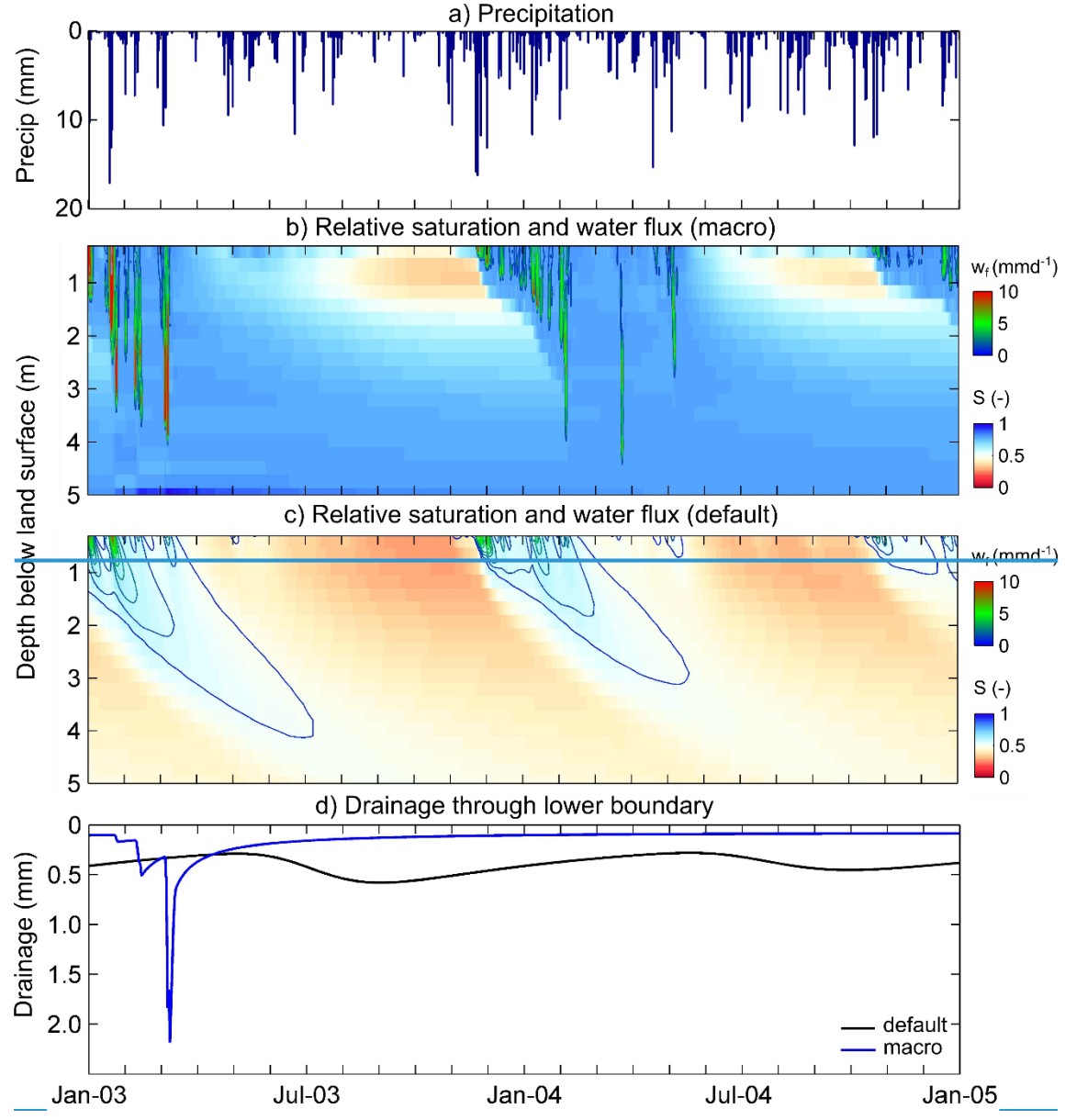


Figure 7. (a) Precipitation and (b) daily sum of drainage through the bottom of the soil

column at Warren Farm over the two simulated years (2003-2005).




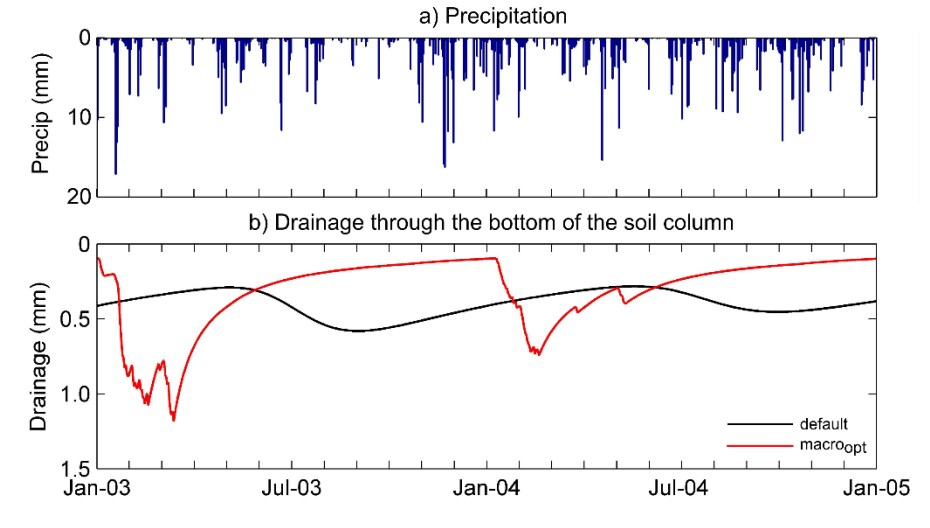

Figure 7. Differences between daily average latent heat flux time series simulated by *default* and *macro* configurations ($LE_{default}$ and $LE_{macro}$, respectively) at Warren Farm.

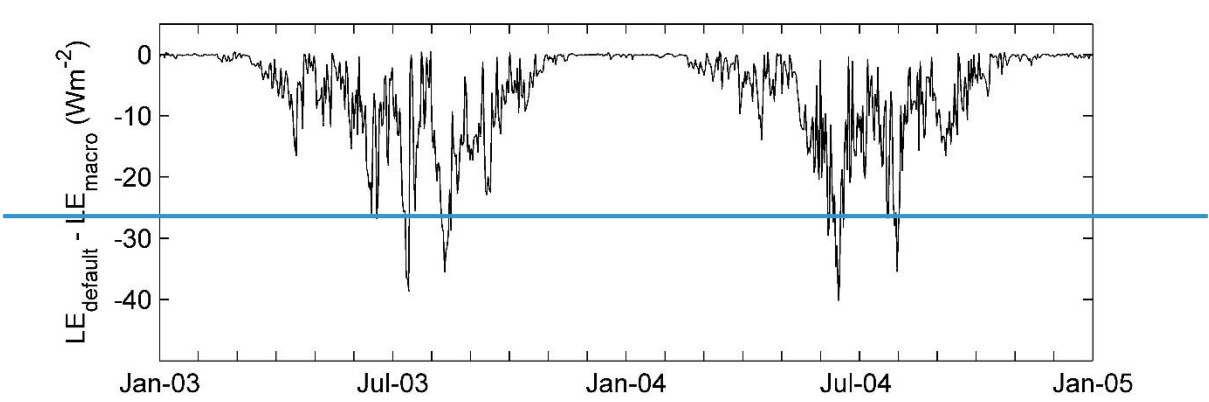














Figure 8. Catchment average 8 day composites of MODIS estimated *LE* (*LE_{MOD}*) against
simulated *LE* from *default* and *macro* configurations (*LE_{default}* and *LE_{macro}*, respectively) along
with the linear models fitted for *LE<sub>default</sub>* (black line) and *LE<sub>macro</sub>* (~~blue~~ red line). The 1:1 line
is shown in ~~grey~~red, which represents the perfect fit between *LE<sub>MOD</sub>* and simulated *LE*.

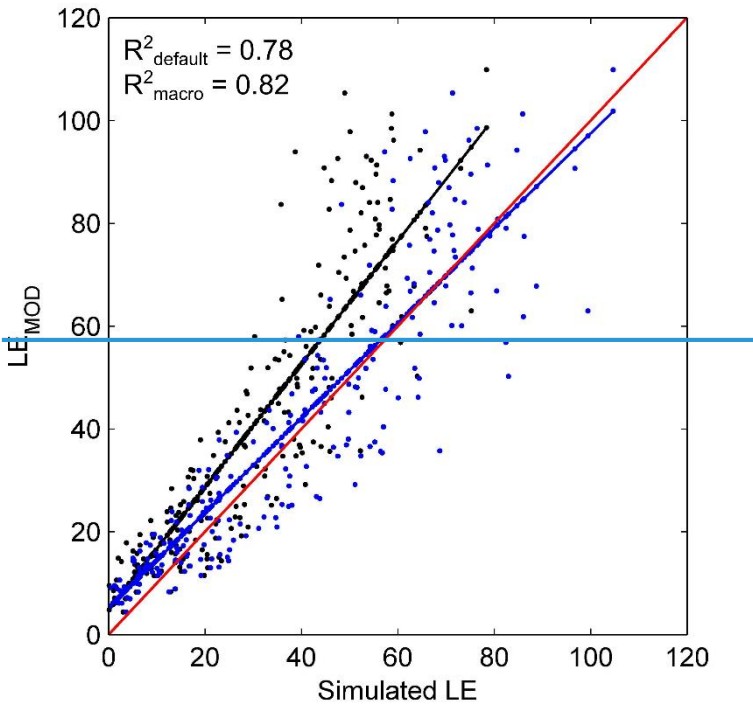



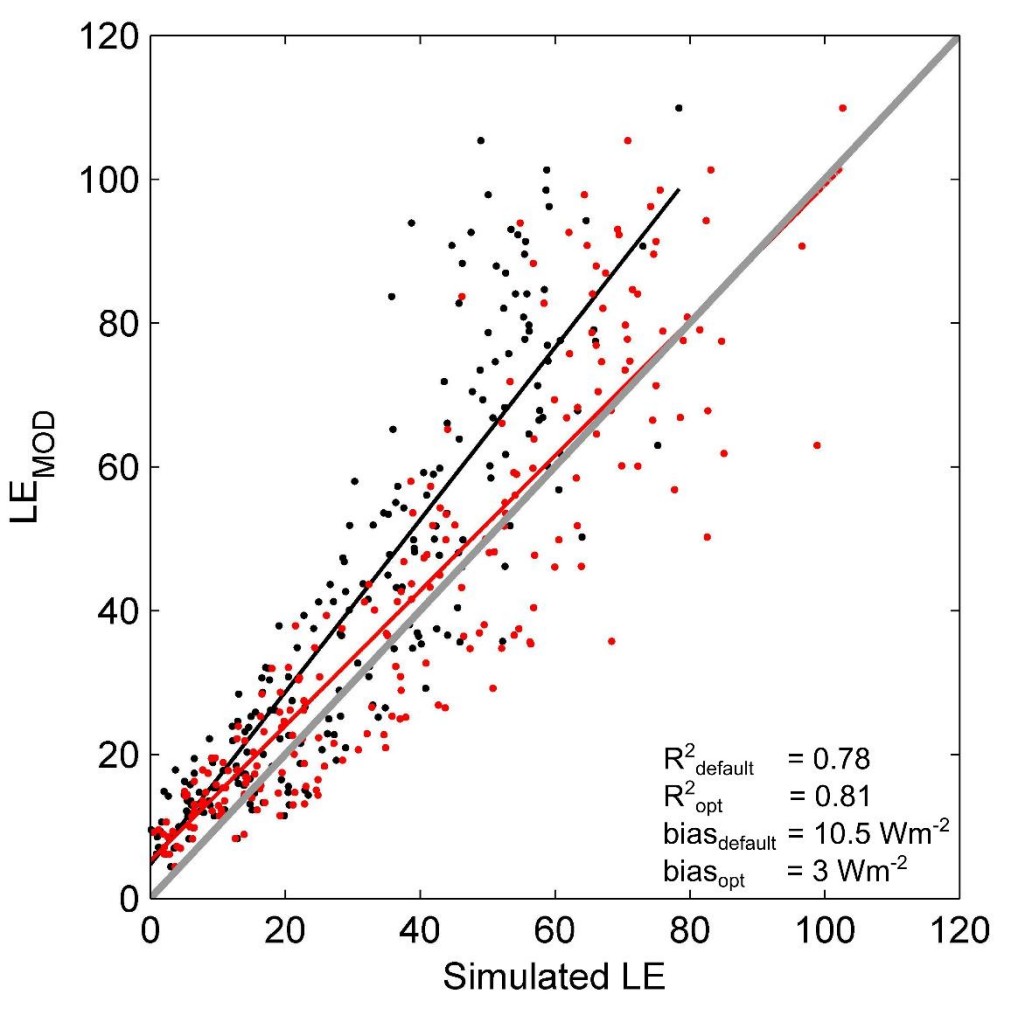






Figure 9. Spatially averaged monthly latent heat flux (*LE*) from MODIS, *default* and *macro$_{opt}$*
configurations over the Kennet catchment.

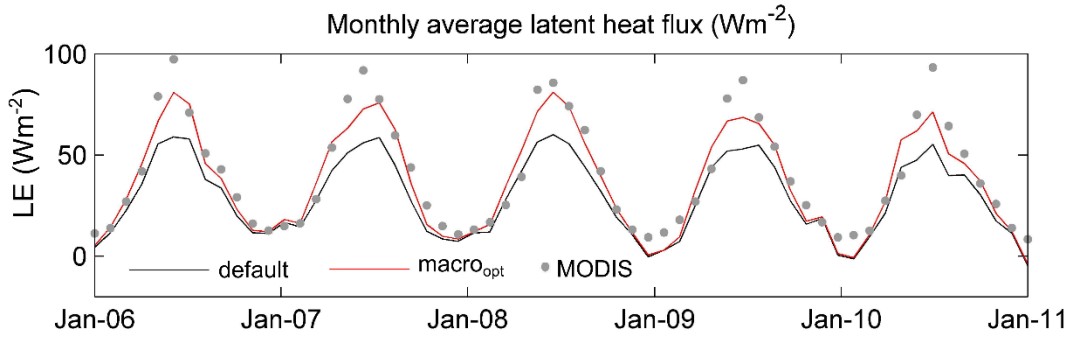


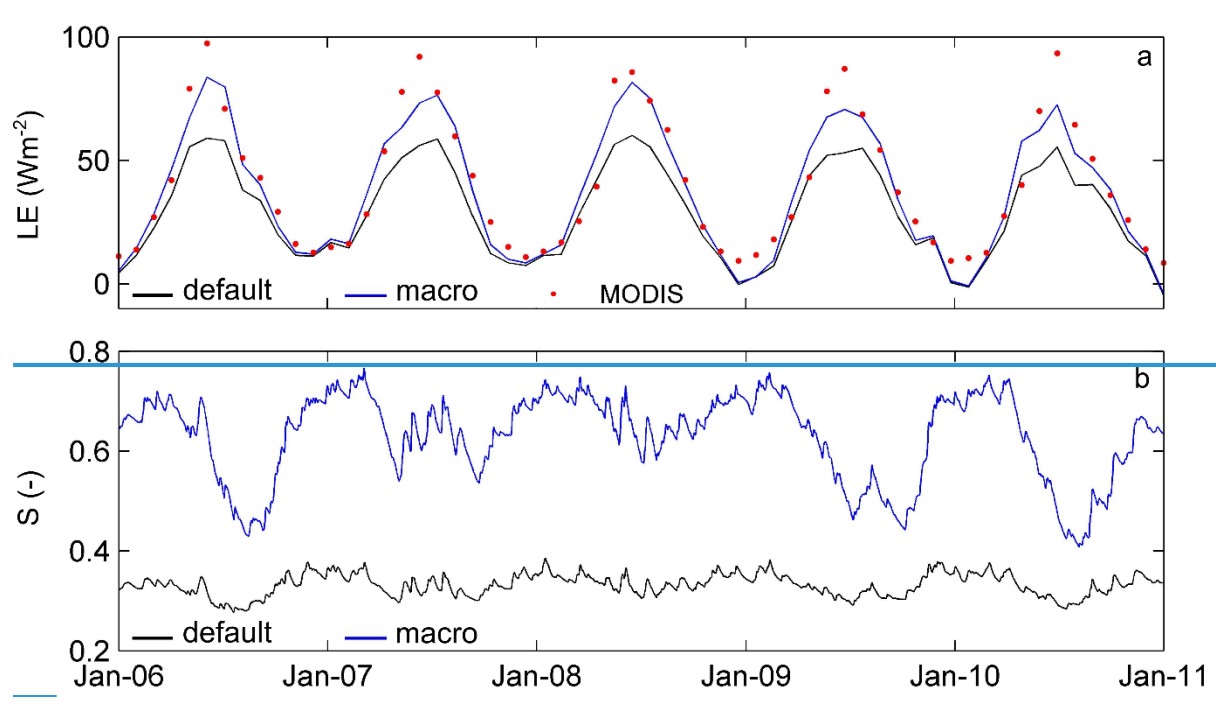













**Supplementary materials**
Figure S1. Saturation-pressure head relationship (May 2003 - December 2005) at Warren
Farm measured fortnightly at 40 cm below land surface. (Source: Ned Hewett, CEH, personal
communication).

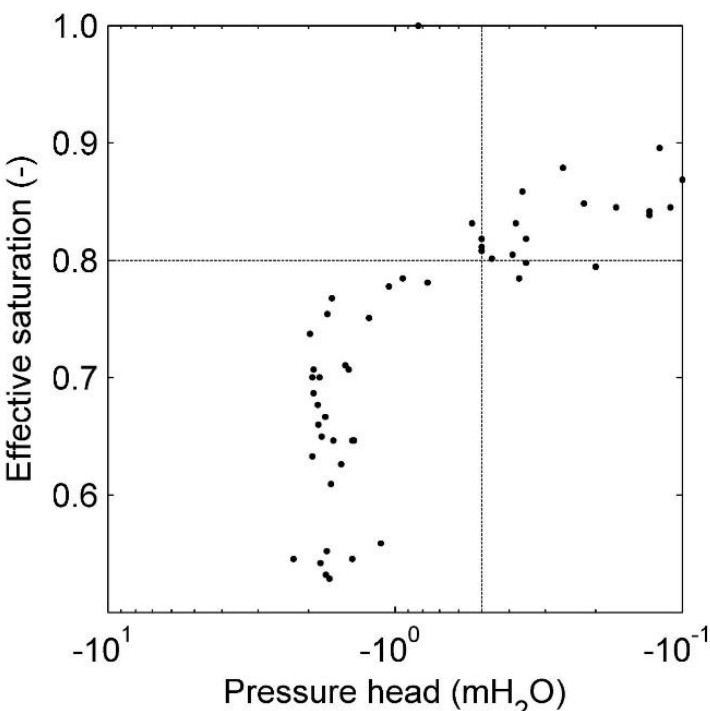









Figure S2. Sensitivity of the BC model parameters on the model performance in simulating
$\Delta\theta$. Note that the parameters are considered one-at-a-time (OAT), and the vertical axis have
different RMSE ranges.

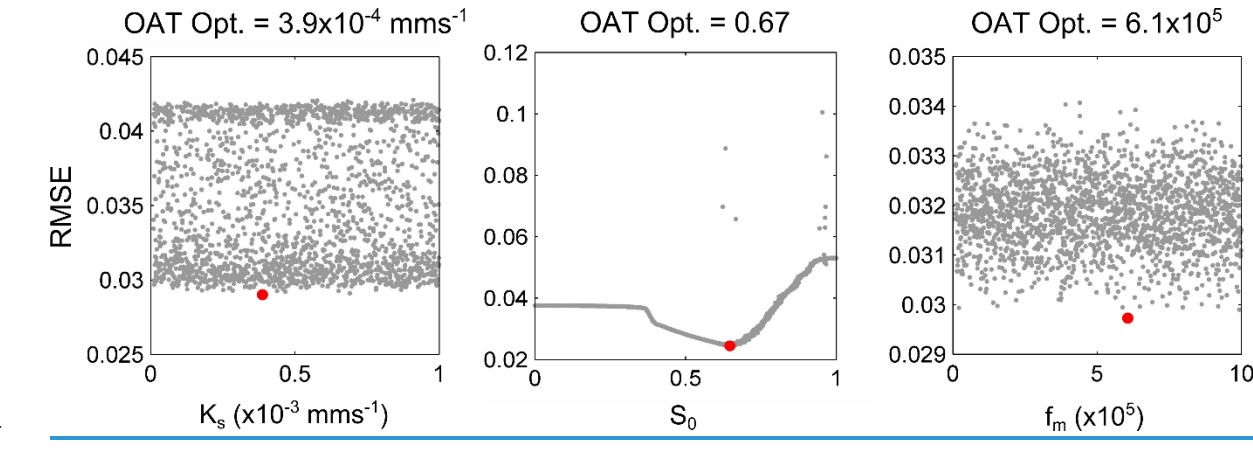