# Peer review of "Towards a simple representation of chalk hydrology in land surface modelling"

_Hydrology and Earth System Sciences, 2016_

## Referee Comment (RC1) · N. Le Vine (Referee) · 1 Jul 2016

**Review of 'The effect of chalk representation in land surface modelling' by M. Rahman and R. Rosolem, HESSD 2016**

The work evaluates an alternative representation of soil hydraulic properties to estimate soil moisture, latent heat and runoff in a chalk catchment using the land surface model JULES. The Bulk Conductivity (BC) scheme is chosen to approximate the chalk dual permeability behavior, and is compared to the default JULES soil parameterisation. The scheme is assessed to reduce JULES soil moisture dry bias, and to improve ET and runoff estimates.

One of my largest concerns regarding this work is its **novelty and contribution to knowledge**. The abstract (l. 13-15) states the work significance as "it is hypothesized that explicit representation of chalk hydrology in a land surface model influences land surface processes by affecting water movement through the shallow subsurface", and that the results corroborate the proposed hypothesis (l. 23). While being a very confusing statement (as our representation in a model does not actually affect any water movement or land surface processes), it is not clear why such a hypothesis is novel and/or requires another round of consideration, as it has been previously shown by others that explicitly accounting for the chalk behavior in the same land surface model and for the same location does have an effect on all the fluxes and states considered in the study, i.e. soil moisture, ET (latent heat), and runoff (see the cited Le Vine et al., HESS 2016, and Bakopoulou, PhD thesis 2015 found on https://spiral.imperial.ac.uk:8443/handle/10044/1/28955 ).

In this light, it seems that the main development is the use of the previously proposed by others **Bulk Conductivity model** to represent chalk hydraulic properties in the land surface model. The BC model is given by eq. (1), which shows that the BC model is activated only when soils are wet (relative saturation above 0.8), and that drier soils are governed by a more traditional van Genuchten soil hydraulic representation with parameters given in Table 3 (note that the two out of four parameters for this model are taken from Le Vine et al., 2016; and the used third parameter Ks equals Ks in the same reference). Figure 9 shows that catchment average relative saturation (S) never exceeds 0.8 (a threshold when the BC model is activated), and there is no point scale characterisation of relative saturation provided to make a judgment about what happens at the application scale of eq. (1). Does this mean that the BC scheme is never activated, and the obtained results are based on a simple re-iteration of the van Genuchten parameterisation, which was used and evaluated by others previously? If not, then the instances when the BC model is activated have to be shown, as it appears to be what distinguishes the work from the work of others. Lastly on this point, the statement in the summary section (l. 373-374) that "BC model was able to reproduce the hydrological processes in chalk without model calibration" is confusing and incorrect, as 1) it is not clear whether the BC portion of the model given in eq. (1) was ever activated, and 2) the chalk behavior for drier states – an inseparable part of the BC model to represent chalk - was governed by van Genuchten representation with parameters calibrated by others for the same application site.

Furthermore, the work compares two setups: 'default' when the standard soil parameterisation is used in JULES, and 'macro' when soil is **uniformly** represented as a

30 cm topsoil, and as chalk from 30 cm to 5 m depth. And it is stated (l. 373-374) that based on the macro model simulations "BC model was able to reproduce the hydrological processes in chalk". I find this statement surprising at the catchment scale, as **macro is an incorrect model setup** for the catchment, as approximately a third of the catchment hydrogeology is not chalk (see the BGS hydrogeology map, or Figure 1 in Le Vine et al., 2016), and thus the application of the chalk soil model uniformly throughout the catchment is erroneous.

Lastly, could the authors comment on how river **flows** were estimated at the outlet when there is no groundwater model available while the catchment is groundwater dominated? It will be very interesting to see the flow hydrographs to compliment the provided flow statistics (given in Table 4).

Regards,

Nataliya Le Vine

---

## Author Comment (AC1) · 13 Jul 2016

**Response to Review Comments by Dr Nataliya Le Vine**

We would like to thank the reviewer for carefully reviewing the manuscript and for her comments and suggestions. We address all relevant points raised by the reviewer below, which will be incorporated in a revised manuscript.

**1. Novelty and contribution to knowledge**

In our current study, we have introduced a new simple parameterization i.e., the Bulk Conductivity (BC) model to simulate the water flow through chalk unsaturated zone in a land surface model (JULES). In addition, the objective of the manuscript is to study the impact of explicit representation of soil-chalk layering features particularly on evapotranspiration (ET) in conjunction with Earth Observations (i.e., MODIS). The reviewer has outlined the similarities between the parameter values between our manuscript and Le Vine et al. (2016). This is due to the fact that our work is motivated by their published study (refer to section 5 in their article), and we therefore decided to minimize differences in other model experimental setups to ensure consistency across multiple configurations in order to better identify the actual impact of our newly proposed BC model. This is a common and expected approach in research practice. Our current study is unique because we introduce the Bulk Conductivity (BC) model (based on the work by Zehe et al., 2001) which is clearly a different approach compared to previous peer-reviewed literature on chalk. Additionally, we investigate the effect of this parameterization on JULES simulated ET in comparison to observations from MODIS.

In conclusion, we strongly believe that our study is novel because of the introduction of the new BC model and the focus on the effect of chalk unsaturated zone hydrology on land surface processes in conjunction with Earth Observations, which has not been discussed explicitly in previous peer-reviewed literature. However, we acknowledge that it is important to refer to other studies that address similar challenges in the region. Therefore we will properly cite the PhD thesis by Bakopoulou (2015) in the revised manuscript.

**2. Incorrect model setup for 'macro' configuration at catchment scale**

We would like to thank the reviewer again for pointing out this important model setup issue in terms of the spatial distribution of chalk over the Kennet catchment. We agree that a spatially uniform representation of chalk over the catchment is indeed incorrect. We have updated the spatial distribution of chalk in the model using the hydrogeology map provided by the British Geological Survey (http://www.bgs.ac.uk/products/hydrogeology/maps.html). Figures R1 and R2 below outlines the differences between the catchment scale simulation results before and after the modification (related to Figure 8 and 9 in the original manuscript, respectively). These figures will be modified in the revised manuscript.

[Figure]

Figure R1. Catchment average 8-day composites of MODIS estimated LE ($LE_{MOD}$) against simulated LE from *default* and *macro* configurations ($LE_{default}$ and $LE_{macro}$, respectively) along with the linear models fitted for $LE_{default}$ (black) and $LE_{macro}$ (blue). The 1:1 line shown in red represents the perfect fit between $LE_{MOD}$ and simulated LE. The left and right figures show the results before and after the modification of spatial distribution of chalk over Kennet, respectively.

[Figure]

Figure R2. Spatially averaged monthly latent heat flux (*LE*) from MODIS, *default*, and *macro* configurations (before and after modification of chalk spatial distribution) over the Kennet catchment.

**3. Activation of Bulk Conductivity model at point- and catchment-scale**

The reviewer has mentioned her concern about whether or not the BC model is ever activated at the catchment scale because Figure 9 in the manuscript does not show relative saturation (S) exceeding 0.8 (the threshold chosen for BC model activation). We would like to draw the attention of the reviewer to the fact that Figure 9 shows catchment average relative saturation for the top 100 cm of the profile (please refer to the caption of the figure). The BC model is applied at every grid cell of the model, which is clearly stated in the original manuscript (l. 80-84). Therefore, the catchment average root zone saturation (as shown in the original Figure 9) is not an indicator of the activation of fracture flow through chalk. The

intention of this figure is to show that soil moisture affects ET mainly in summer (l. 325-335).

As an illustrative example, Figure R3 below shows the number of instances the BC model (with the updated chalk spatial distribution) was activated at 4[th] model layer (30-40 cm below surface, the first model layer with chalk) over the entire simulation period. Moreover, note that the mechanism of the BC model at the point scale is discussed elaborately in the original manuscript (Figure 6, l. 246-283), which describes the effect of the new parameterization on water movement through chalk unsaturated zone.

[Figure]

Figure R3. Number of instances of BC model activation (i.e., relative saturation exceeding 0.8) for the 4[th] model layer (layer between 30-40 cm below surface).

**4. River discharge at outlet**

The reviewer also commented about the river discharge at the outlet of the Kennet catchment. A description of routing surface runoff to the river network in the model is provided in the manuscript (l. 153-160). We agree with the reviewer that groundwater substantially affects the hydrology over Kennet, and a groundwater representation is certainly needed to better understand river flows in the area, which is also mentioned in our manuscript (l. 397-403). However, such analysis is beyond the scope of our work, as we mainly focus on the surface water and energy partitioning as represented by the interaction between soil moisture and evapotranspiration. The analysis of overall water balance (Table 4 of the original manuscript) corroborates the fact that the overall magnitude of the hydrological fluxes in the catchment is consistent with observations.

---

## Referee Comment (RC2) · A. Ireson (Referee) · 14 Jul 2016

**Review of "The effect of chalk representation in land surface modelling"**

Rahman and Rosolem

The authors are to be congratulated for writing a paper on modelling flow in the Chalk unsaturated zone which includes only one equation - surely a record! In fact, this flippant comment underlies a more serious point, which is that the potential strength of this study lies in it's simplicity. The authors are working in a field where others have proposed various different, highly complicated models. The authors have taken a very simple model configurations and applied it to the Chalk in a manner that goes beyond

a completely naive model that might be used in routine large scale model applications (i.e. the default configuration shown here) in a manner that could be easily included within routine applications of large scale models. All of the data used to configure this model are readily available in soil databases and the Chalk literature. This is the potential contribution of this paper, in my view - it is a model for the Chalk that could be readily picked up and used by almost anyone. I think that is the holy grail which I myself and others have been searching for in the Chalk.

However, that is the potential contribution. The fundamental problem with this study is that the default model outperforms the macro model in a number of important respects. The authors failed to recognize this because they focused on which model better fits the absolute water content. This is really not an important metric of model performance - far more important is the changes in water content and the groundwater recharge signal. The default model probably outperforms the macro model in simulating the changes in water content. The authors appear not to have thought critically about the simulated potential recharge flux in relation to the water table responses - again in this regard the default model seems much better, as I detailed further below.

So, I think the authors are on the path to having a significant contribution, but not there yet. I suspect that it is the parameters they are using, specifically the matrix $K$ is too low, that are to blame for the poor model performance. Model calibration, ideally combined with a parametric sensitivity/uncertainty analysis, is essential before this work can be published. I am therefore recommending major revisions.

Another issue in this paper is that the contributions are not well described. The abstract and conclusions in particular are poorly written. It is my own suggestion that the contribution here is the simplicity of the model - the authors do not say that. Instead, a very tepid and vague hypothesis about their parameterisation having some sort of influence on the model. This must be strengthened, since there is a potentially very nice piece of work here.

HESSD

**Major comments**

The abstract is poorly written. The hypothesis is not well phrased, and uninteresting as phrased. The conclusion in the abstract is vague and doesn't make me want to read the rest of the paper. This can be improved with some more careful thinking about what the contribution of this study is, and highlighting this clearly for the reader.

The premise of the paper, while poorly described, is good - that is to take a new conceptualization of the hydraulic properties of the Chalk, test it at a point scale against local observations, and then apply this at the catchment scale.

In Section 3, it feels like the description of how the soils and Chalk were parameterized is spread out and not well organized. Perhaps you should mention this parameterization at the begining of Section 3.4 before talking about the two different scales.

In Figures 3 and 4 you show the performance of the default and macro model, respectively, in reproducing observations of soil moisture. In the text (L. 223-245), your focus is the marked improvement in the macro model at simulating the absolute values of water content in the deeper soil layers. This is also the message of Figure 5 which uses relative bias as a model performance metric. This is valid but actually I'm more interested in how well the models capture changes in soil moisture, which is a more important metric from the perspective of the water balance and recharge estimates. In this respect, it appears to me that the default model may actually be better than the macro model, and an optimal model might be somewhere between these two extremes. To highlight this point, consider Figure 6 d), which shows the potential recharge flux (or the drainage flux, if you prefer that term) from the base of the 5m model. Consider the fact that this flux will ultimately drive water table fluctuations in the Chalk, 10s of meters below ground level. As is well documented elsewhere (e.g. Wellings and Bell, 1980, who put together the classic understanding of Chalk recharge in their excellent Figure 1.) the water table follows a clear seasonal pattern. Only the default model here could result in that pattern. (Note also that water table observations at this site are available,

e.g. see Fig 12 in Ireson et al., 2009).

In Figure 7 the differences in latent heat simulated by the two models are shown. Why aren't the actual values shown and compared with observations from the local flux tower that exists at this site through the NERC LOCAR program (e.g. see Roberts, J., Rosier, P., Smith, D.M., 2005. The impact of broadleaved woodland on water resources in lowland UK: II. Evaporation estimates from sensible heat flux measurements over beech woodland and grass on chalk sites in Hampshire. Hydrology and Earth System Sciences 9 (6), 607–613.)? We can infer that the macro configuration results in more evapotranspiration, but which configuration is more realistic? Again, this goes to the heart of whether or not the macro model is an improvement, though I can't comment on this since the results are not presented clearly.

In terms of the catchment scale model application, it is notable that the macro model improves the evaporation estimates, and apparently improves runoff, although it looks rather conspicuous that no runoff plots are included. Why is this?

**Corrections to the text**

L. 8  Is it really the 'efficiency' of simulations that is the critical limitation?
L. 11  Poor grammar in this sentence, and the meaning is not completely clear. The mass and energy fluxes are influences regardless of whether the hydrology is complex or non-linear. Try to make this sentence more specific and meaningful.
L. 13  I'm not sure this hypothesis is well phrased. It would be very surprising if there was no influence of this representation on the fluxes of water. As phrased, this suggests you are just looking at sensitivity, but I think it would be better to test whether or not this representation results in some sort of improvement? That is presumably what you're actually doing anyway.
L. 17  Change "applied on" to "applied to"
L. 28  "various processes" are there others, other than recharge?
L. 36  The fracture porosity cited ($10^{-4}$) seems much lower than other published

estimates for the Chalk. I believe Price et al., 1993 actually cite a value of $10^{-2}$ (though I don't have this reference to hand, please check this).

L. 42  Change "Mathuis" to "Mathias"

L. 55  Change "curve was" to "curves were"

L. 66  What is a "consistent representation ..."? Consistent with what? This sentence restates your hypothesis, which I repeat is very unexciting and somewhat vague. You need to be much more specific here.

L. 100  It is possible, but not certain, that $f_m$ is a sensitive parameter - in which case adopting this arbitrary value without sensitivity analysis or calibration would seem dubious. I'm also not convinced that it makes sense to make the fracture conductivity functionally dependent on the matrix conductivity - is there any physical reason to not to treat these two properties as independent?

L. 167  Here you describe the hydraulic properties of the soil, but you don't described the properties of the Chalk until line 209. Would make sense to rearrange the text so that these are described in the same place inthe text.

L. 188  This is a little bit confusing - it reads like you are saying the hydraulic properties of the soil are uniform over the catchment? However, from Figure 1 and Table 2 this is not the case. Please clarify the text here.

L. 226  I think it is not particularly interesting that the model underestimates the absolute values of the observed soil moisture - I would be more interested in how well it reproduces the changes in soil moisture. It probably underestimates these, but it's not completely terrible. I'm surprised how well this simple default model with no calibration does!

L. 261  In the Ireson al al. (2009) paper referenced here (full disclosure - I am Ireson), Fig 13 shows the drainage flux at 5 m depth, which could be directly compared with Figure 5 in this paper. It can be seen in my paper there was negligible fracture flow at 5 m depth during this period, which is not consistent with the authors interpretation that "fracture flow dominates... during wet periods". So I'm afraid the result here in not consistent with my result - at least the macro model result is not - the default model

might be!

L. 311  Why use $R^2$ for this? RMSE or bias would be better - we are interested in the absolute values in this case.

L. 336  Again we have references to this weak hypothesis that there is an "influence".

L. 362  Again the word "consistent" - consistent with what?

L. 372  You say these two parameters can be estimated from the matrix without calibration, but this is an assertion, since you haven't tested these parameters in this study. In fact, my central criticism of the findings in this paper are likely due to poor choices of parameter values in your model. If you were to increase your matrix $K$, I suspect the model would improve.

L. 374  Overall, I cannot agree that the model was able to reproduce the hydrological processes in the Chalk successfully, or even to an acceptable degree. Groundwater recharge is completely wrong.

L. 376  Yes, good that you suggest calibration, but without doing this the model is not ready for publication, since it fundamentally fails to simulate convincing groundwater recharge fluxes. Especially on the 1D model, calibration really is not that hard to do, so must be done before this paper can be accepted, in my view.

L. 401  Delete this final sentence saying you will address coupling with a groundwater model in the future. That is, by definition, outside the scope of this paper, hence irrelevant.

L. 385 +  The last three paragraphs of the conclusions are very disjointed.

---

## Referee Comment (RC3) · Anonymous Referee #3 · 15 Jul 2016

**The effect of chalk representation in land surface modelling**

This paper proposes the bulk conductivity (BC) model for improving the simulation of chalk hydrology in land surface models. The bulk conductivity model appears a simple approach for simulating both matrix and fracture flow in the Chalk according to the relative saturation. This approach is implemented in JULES (macro) and the results are compared with JULES (default) runs using a typical soil parameterisation, but neither model is calibrated. This is undertaken at the point and, subsequently, catchment scale. The authors suggest that the addition of the BC model in JULES improves soil moisture, evaporation, and runoff simulation.

**Major comments**

The default soil parameterisation is based on soil texture data from the surface (a loam) down to 5 m depth (Line 164 & Table 2) despite Figure 2 showing that this soil horizon is only 20-30 cm deep, with the remaining profile being chalk. Is it then any real surprise that this uncalibrated JULES model performs worse that a JULES model modified specifically for simulating chalk hydrology? This is not a valid comparison and the conclusions drawn are not valid.

To make any comparison valid the default model runs have to be calibrated, in particular, to achieve a more appropriate soil parameterisation. Currently the default run for the catchment is simulating more than twice the observed runoff for the Kennet, which is not acceptable. Consequently, evaporation is underestimated and soil moisture storage is insufficient. The River Kennet could be used for calibration at the catchment scale and soil moisture data for the point scale.

The macro model appears to be performing reasonably well where described in the text and calibration may not be as essential. However, I would like to see a sensitivity analysis for the BC parameters, which would be very useful for anyone considering implementing this approach for chalk models in the future, particularly when there are so few parameters.

Although the macro model is performing well where explicitly described in the text, Figure 6d asks some serious questions. The macro model is simulating about 0.1 mm/d of potential recharge with the exception of the early 2003 event. This would equate to only c.35 mm of potential recharge a year on a grassland site, on outcrop chalk, in a temperate climate. This seems unrealistically low and the c.200 mm of potential recharge simulated by the default model is perhaps more realistic.

The hypothesis that is proposed and maintained throughout the paper is a minor point, in my opinion, and takes away from the headline story: a simple approach for simulating matrix/fracture flow in the Chalk unsaturated zone, which could be implemented in LSMs.

The abstract needs to be completely re-written and re-focussed on the above comment. There is currently only one sentence (lines 20-23) concerning the results and implications and this is very vague.

The last three paragraphs of the conclusions (lines 385-403) could easily all be deleted or at least only be summarised in a couple of paragraphs. Currently it dilutes the section.

**Minor comments**

Lines 28-29 – consider rewording

Line 55 – this should be plural

Section 3.1 – the Kennet is a tributary of the Thames

Throughout the paper 'the' is frequently omitted, e.g. 'River Kennet discharges', 'major tributarites of this river are Lambourn, etc

Line 122 – there are 3 years more soil moisture data available at Warren Farm, which CEH collected. These data extend into a wetter period when it would be interesting to see how the models compared.

Line 125 – suggests soil moisture observations only exist to 2.4 m depth but observations in Figure 3 suggest there are deeper data.

Liner 240 – change 'dry' to 'drier'

Section 4.2 – $r^2$ informs on the fit of the linear regression but perhaps a plot of RMSE over time would be more useful to inform on actual differences between observed and modelled.

Section 4.2 – there appears to be a lot more noise in the modelled LE during Summer?

Figures – blue is not always very distinguishable from black on the comparison plots

Figure 5 – Are the default and macro results from the same depth interval here?

---

## Editor Comment (EC1) · N. Romano (Editor) · 29 Jul 2016

Dear Authors, To further feed the discussion and in view of some criticisms received so far, I'd suggest you should provide soon some preliminary responses to the two additional referees' comments.

---

## Author Comment (AC2) · 2 Aug 2016

We would like to thank the reviewer (Dr Andrew Ireson) for his detailed comments and suggestions. Here we initially address the three main issues highlighted by the reviewer:

1. Emphasis on the simplicity of proposed Bulk Conductivity parameterization

The reviewer has mentioned that one of the potential strengths of our work is the simplicity of the Bulk Conductivity (BC) model which could aid large scale modelling applications. We thank the reviewer for this comment as this is very encouraging for us. We will make sure to better emphasize this important aspect in our revised manuscript.

2. More in-depth analysis of the BC model

We thank the reviewer for pointing out additional suggestions for analysing the BC model in depth. We will carry out sensitivity analysis and calibration of key parameters while also expanding our analysis to evaluate changes in soil moisture and potential recharge. However, we would like to emphasize that our main focus is to demonstrate the soil moisture – evapotranspiration interactions from a land surface modelling perspective, when explicitly representing the soil-chalk layering system within the catchment in JULES (as a first step of a more comprehensive analysis of hydrological processes in the region). Hence, we focus on analysing actual soil moisture magnitudes in the current manuscript.

3. Absence of river runoff plots

We believe that a detailed analysis of river flows is beyond the scope of our work as we mainly focus on the surface water and energy partitioning as represented by the interaction between soil moisture and evapotranspiration after explicitly representing soil-chalk layers in the model (as noted earlier). This is because JULES prescribes a free drainage bottom boundary condition without representing groundwater dynamics, and does not consider topography. These are common assumptions often made in traditional land surface models (such as JULES) which can substantially affect the routing of runoff to the river network. In our studied catchment, groundwater plays an important role on hydrology. Hence, we focus on the annual water balance that corroborates the fact that the overall magnitude of the hydrological fluxes simulated in the catchment is consistent with observations.

Once again, we would like to express our sincere gratitude to the reviewer for the careful and detailed review, which will be very valuable in adding to the quality of the revised manuscript.

---

## Author Comment (AC3) · 2 Aug 2016

We would like to thank the reviewer for thoroughly reviewing the manuscript. Here we initially address the three main issues highlighted by the reviewer:

1. Calibration of default JULES

The reviewer has suggested to calibrate the parameters of the default model configuration as it performs poorly. The default JULES configuration is consistent to its operational application and represents "a completely naive model that might be used in routine large scale model applications" as mentioned by Reviewer #2. The macro configuration, on the other hand, explicitly represents chalk hydrology in this model. A prior calibration of the default model would undermine the impact of proposed implementation aimed for large-scale application, in comparison to the more common

use of land surface models (i.e., without excessive calibration of individual grid cells or catchments). Hence, we believe the uncalibrated JULES configuration is the true representation of the default model used as the reference/control case in our study.

2. Sensitivity analysis of the proposed Bulk Conductivity (BC) model

We thank the reviewer for suggesting the use of sensitivity analysis in order to better understand the BC model. This is an important aspect also suggested by Reviewer #2. We will carry out sensitivity analysis of key parameters while also expanding our analysis to evaluate potential recharge in the model (also pointed out by the reviewer).

3. Simplicity of the Bulk Conductivity (BC) model

The reviewer has suggested to emphasize on the simplicity of the Bulk Conductivity (BC) model that is used in this study to simulate hydrology through chalk unsaturated zone. Note that reviewer #2 has also highlighted this point, which will be addressed in the revised manuscript. However, we would like to emphasize that our main focus is to quantify the impacts on soil moisture – evapotranspiration interactions from a land surface modelling perspective, when explicitly representing the soil-chalk layering system within the catchment in JULES. This is our first step for a more comprehensive analysis of hydrological processes in the region.

We would like to thank the reviewer again for this comments and suggestions, which will be addressed carefully in the revised manuscript.

---

## Author Response (AR1)

We would like to thank the reviewers for providing us with valuable comments and suggestions that improved the manuscript. A point-by-point reply to the comments of all the reviewers is provided below with the original comments shown in blue.

**Reviewer #1 (Dr Nataliya Le Vine)**

The work evaluates an alternative representation of soil hydraulic properties to estimate soil moisture, latent heat and runoff in a chalk catchment using the land surface model JULES. The Bulk Conductivity (BC) scheme is chosen to approximate the chalk dual permeability behavior, and is compared to the default JULES soil parameterisation. The scheme is assessed to reduce JULES soil moisture dry bias, and to improve ET and runoff estimates.

One of my largest concerns regarding this work is its novelty and contribution to knowledge. The abstract (l. 13-15) states the work significance as "it is hypothesized that explicit representation of chalk hydrology in a land surface model influences land surface processes by affecting water movement through the shallow subsurface", and that the results corroborate the proposed hypothesis (l. 23). While being a very confusing statement (as our representation in a model does not actually affect any water movement or land surface processes), it is not clear why such a hypothesis is novel and/or requires another round of consideration, as it has been previously shown by others that explicitly accounting for the chalk behavior in the same land surface model and for the same location does have an effect on all the fluxes and states considered in the study, i.e. soil moisture, ET (latent heat), and runoff (see the cited Le Vine et al., HESS 2016, and Bakopoulou, PhD thesis 2015 found on https://spiral.imperial.ac.uk:8443/handle/10044/1/28955).

- In our study, we have introduced a new simple parameterization i.e., the Bulk Conductivity (BC) model to simulate the water flow through chalk unsaturated zone. The focus of our revised manuscript (*man_rev* hereafter) is to demonstrate the suitability of the BC parameterization for large scale land surface modelling applications given the simplicity of the approach, as emphasized by the other two reviewers. Hence, the objective of *man_rev* is different from the studies mentioned by the reviewer [e.g., *Le Vine et al.*, 2016; *Bakopoulou*, 2015]. Therefore, we believe that our study is novel and contributes to the knowledge concerning chalk representation in land surface modelling. Note that the novelty and contribution of our work has also been appreciated by reviewer 2 (Dr. Andrew Ireson). Because it is important to refer to other studies that address similar challenges in the region, we have incorporated the PhD thesis by Bakopoulou (2015) in *man_rev* (L. 32).

In this light, it seems that the main development is the use of the previously proposed by others Bulk Conductivity model to represent chalk hydraulic properties in the land surface model. The BC model is given by eq. (1), which shows that the BC model is activated only when soils are wet (relative saturation above 0.8), and that drier soils are governed by a more traditional van Genuchten soil hydraulic representation with parameters given in Table 3 (note that the two out of four parameters for this model are taken from Le Vine et al., 2016; and the used third parameter Ks equals Ks in the same reference). Figure 9 shows that catchment average relative saturation (S) never exceeds 0.8 (a threshold when the BC model is activated), and there is no point scale characterisation of relative saturation provided to make a judgment about what happens at the application scale of eq. (1). Does this mean that the BC scheme is never activated, and the obtained results are based on a simple re-iteration of the van Genuchten parameterisation, which was used and evaluated by others previously? If not, then the instances when the BC model is activated have to be shown, as it appears to be what distinguishes the work from the work of others. Lastly on this point, the statement in the summary section (l. 373-374) that "BC model was able to reproduce the hydrological processes in chalk without model calibration" is confusing and incorrect, as 1) it is not clear

whether the BC portion of the model given in eq. (1) was ever activated, and 2) the chalk behavior for drier states – an inseparable part of the BC model to represent chalk - was governed by van Genuchten representation with parameters calibrated by others for the same application site.

- Figure 9 in the original manuscript (*man_org* hereafter) shows catchment average relative saturation for the top 100 cm of the profile (please refer to the caption of the figure). The BC model is applied at every grid cell of the model. Therefore, the catchment average root zone saturation (as shown in Figure 9 of *man_org*) is not an indicator of the activation of fracture flow through chalk. As an illustrative example, Figure R1 below shows the number of instances the BC model (with the updated chalk spatial distribution) was activated at 4th model layer (30-40 cm below surface, the first model layer with chalk) over the entire simulation period.

[Figure]

Figure R1. Number of instances of BC model activation for the 4th model layer (30-40 cm below surface).

In *man_rev*, we have estimated the soil hydraulic parameters of chalk in the BC model based on existing literature as a first step of model evaluation, which we believe is a common and expected approach in research practice. Subsequently, we have optimized the BC model parameters using soil moisture data from the Warren Farm site following the suggestions of Reviewer 2 and 3. Note that for the catchment scale simulations, we have used our optimized parameters in the BC model.

Furthermore, the work compares two setups: 'default' when the standard soil parameterisation is used in JULES, and 'macro' when soil is uniformly represented as a 30 cm topsoil, and as chalk from 30 cm to 5 m depth. And it is stated (l. 373-374) that based on the macro model simulations "BC model was able to reproduce the hydrological processes in chalk". I find this statement surprising at the catchment scale, as macro is an incorrect model setup for the catchment, as approximately a third of the catchment hydrogeology is not chalk (see the BGS hydrogeology map, or Figure 1 in Le Vine et al., 2016), and thus the application of the chalk soil model uniformly throughout the catchment is erroneous.

- We would like to thank the reviewer for pointing out this important issue. We agree that a spatially uniform representation of chalk over the Kennet catchment is indeed incorrect.

Therefore, we have updated the spatial distribution of chalk in *man_rev* (Figure 1c) using the hydrogeology map provided by the British Geological Survey (http://www.bgs.ac.uk/products/hydrogeology/maps.html).

Lastly, could the authors comment on how river flows were estimated at the outlet when there is no groundwater model available while the catchment is groundwater dominated? It will be very interesting to see the flow hydrographs to compliment the provided flow statistics (given in Table 4).

- A description of surface and subsurface runoff routing to the river network in JULES is provided in *man_rev* (L. 175-181). Figure R2 below shows a comparison between observed and simulated discharge from the $macro_{opt}$ configuration (please refer to *man_rev*) at the Theale gauging station (please refer to Figure 1a in *man_rev*). Note that Figure R2 compares 10-day average runoff to minimize the effect of routing in the model. Considering the lack of groundwater, the observed and simulated runoff at Theale shows quite reasonable agreement.

We agree that a groundwater representation is needed for an efficient estimate of river discharge at gauging stations in the area. However, such implementation is beyond the scope of this study. In this study, we aim to demonstrate that the newly proposed BC model improves the overall mass and energy balance components compared to the *default* configuration that represents a "completely naïve model" (as Dr. Andrew Ireson has pointed out in his review). We did not include the comparison between observed and simulated river discharge at gauging stations in *man_rev*. Nevertheless, the seasonal aspect of discharge is captured reasonably well in our proposed model (Figure R2). The analysis of overall water balance (Table 4 of *man_rev*) corroborates the fact that the overall magnitude of the hydrological fluxes in the catchment is consistent with observations.

[Figure]

Figure R2. Comparison between observed and simulated discharge from the $macro_{opt}$ configuration at the Theale gauging station.

**Reviewer #2 (Dr Andrew Ireson)**

The authors are to be congratulated for writing a paper on modelling flow in the Chalk unsaturated zone which includes only one equation - surely a record! In fact, this flippant comment underlies a more serious point, which is that the potential strength of this study lies in it's simplicity. The authors are working in a field where others have proposed various different, highly complicated models. The authors have taken a very simple model configurations and applied it to the Chalk in a manner that goes beyond a completely naive model that might be used in routine large scale model applications (i.e. the default configuration shown here) in a manner that could be easily included within routine applications of large scale models. All of the data used to configure this model are readily available in soil databases and the Chalk literature. This is the potential contribution of this paper, in my view - it is a model for the Chalk that could be readily picked up and used by

almost anyone. I think that is the holy grail which I myself and others have been searching for in the Chalk.

- We would like to thank the reviewer for his encouraging comments.

However, that is the potential contribution. The fundamental problem with this study is that the default model outperforms the macro model in a number of important respects. The authors failed to recognize this because they focused on which model better fits the absolute water content. This is really not an important metric of model performance - far more important is the changes in water content and the groundwater recharge signal. The default model probably outperforms the macro model in simulating the changes in water content. The authors appear not to have thought critically about the simulated potential recharge flux in relation to the water table responses - again in this regard the default model seems much better, as I detailed further below.

- We agree with the reviewer that the change in soil moisture ($\Delta\theta$) is a very important metric that is connected to groundwater recharge. In *man_rev*, we have analysed the $\Delta\theta$ as suggested by the reviewer. The results (Figure 4 and 5 in *man_rev*) show that the *macro* configuration shows relatively better performance in simulating $\Delta\theta$ compared to *default*. However, both configurations show considerable discrepancies with observed $\Delta\theta$ in general. Therefore, we have optimized the model parameters to minimize the differences between observed and simulated $\Delta\theta$ in *man_rev* as suggested by the reviewer.

So, I think the authors are on the path to having a significant contribution, but not there yet. I suspect that it is the parameters they are using, specifically the matrix K is too low, that are to blame for the poor model performance. Model calibration, ideally combined with a parametric sensitivity/uncertainty analysis, is essential before this work can be published. I am therefore recommending major revisions.

- This is a very valuable suggestion and we thank the reviewer for pointing that out. In *man_rev*, we have optimized the BC model parameters (including the matrix $K_s$ of chalk) following the suggestion of the reviewer. The sensitivity of the BC model parameters on simulated $\Delta\theta$ is illustrated in Figure S2 and the optimization results are shown Figure 4 in *man_rev*. Based on the sensitivity analysis results, we improved the BC parameterization further by optimizing both $K_s$ of chalk matrix and $S_0$ to minimize the differences between observed and simulated $\Delta\theta$ as suggested by the reviewer.

Another issue in this paper is that the contributions are not well described. The abstract and conclusions in particular are poorly written. It is my own suggestion that the contribution here is the simplicity of the model - the authors do not say that. Instead, a very tepid and vague hypothesis about their parameterisation having some sort of influence on the model. This must be strengthened, since there is a potentially very nice piece of work here.

- We have re-written the abstract and conclusion sections in *man_rev*. We have focused on the simplicity of the BC model in the new version following the suggestions of the reviewer. We have also changed the title of the manuscript to "Towards a simple representation of chalk hydrology in land surface modelling" in order to emphasize the simplicity of our proposed approach.

**Major comments**

The abstract is poorly written. The hypothesis is not well phrased, and uninteresting as phrased. The conclusion in the abstract is vague and doesn't make me want to read the rest

of the paper. This can be improved with some more careful thinking about what the contribution of this study is, and highlighting this clearly for the reader.

- The abstract has been re-written in *man_rev*. Please, refer to comment above.

The premise of the paper, while poorly described, is good - that is to take a new conceptualization of the hydraulic properties of the Chalk, test it at a point scale against local observations, and then apply this at the catchment scale.

In Section 3, it feels like the description of how the soils and Chalk were parameterized is spread out and not well organized. Perhaps you should mention this parameterization at the begining of Section 3.4 before talking about the two different scales.

- We have re-written section 3.4 in the *man_rev* following the suggestions of the reviewer. In *man_rev*, the hydraulic properties of soil and chalk are discussed consecutively (L. 212-222, Table 2 and 3).

In Figures 3 and 4 you show the performance of the default and macro model, respectively, in reproducing observations of soil moisture. In the text (L. 223-245), your focus is the marked improvement in the macro model at simulating the absolute values of water content in the deeper soil layers. This is also the message of Figure 5 which uses relative bias as a model performance metric. This is valid but actually I'm more interested in how well the models capture changes in soil moisture, which is a more important metric from the perspective of the water balance and recharge estimates.

In this respect, it appears to me that the default model may actually be better than the macro model, and an optimal model might be somewhere between these two extremes.
To highlight this point, consider Figure 6 d), which shows the potential recharge flux (or the drainage flux, if you prefer that term) from the base of the 5m model. Consider the fact that this flux will ultimately drive water table fluctuations in the Chalk, 10s of meters below ground level. As is well documented elsewhere (e.g. Wellings and Bell, 1980, who put together the classic understanding of Chalk recharge in their excellent Figure 1.) the water table follows a clear seasonal pattern. Only the default model here could result in that pattern. (Note also that water table observations at this site are available, e.g. see Fig 12 in Ireson et al., 2009).

- We thank the reviewer for his valuable points. We have compared the performances of the *default* and *macro* configurations in simulating $\Delta\theta$ in *man_rev* (Figure 4 and 5). The results show that the *macro* configuration performs relatively better in this respect compared to *default*. Despite this relative improvement in model performance, we found considerable discrepancies between observed and simulated $\Delta\theta$ for both configurations. Therefore, we optimize the BC model parameters to minimize the differences between observed and simulated $\Delta\theta$. The results of the optimization show that *macro$_{opt}$* with optimized $K_s$ and $S_o$ parameters results in the best model performance in simulating $\Delta\theta$ *(Figure 4 of man_rev).*

Drainage through the bottom of the soil column ($d_b$) at Warren Farm is shown in Figure 7 in *man_rev*. This seasonal pattern of drainage is more consistent with the recharge pattern in chalk [*Wellings and Bell*, 1980; *Ireson et al.*, 2009] compared to the default configuration that shows higher drainage in summer. In conclusion, the *macro$_{opt}$* configuration performs better compared to *default* in simulating $\theta$, $\Delta\theta$ and the seasonal pattern of drainage through the bottom of the soil column at the Warren Farm site. Note that JULES considers a free-drainage lower boundary condition and does not represent groundwater dynamics. Moreover, as discussed in *Ireson et al.* [2009] the variation in the water table elevation may not be the result of changes in the recharge flux over time at Warren Farm. Therefore, we did not incorporate groundwater table depth data in this study in relation to $d_b$.

In Figure 7 the differences in latent heat simulated by the two models are shown. Why aren't the actual values shown and compared with observations from the local flux tower that exists at this site through the NERC LOCAR program (e.g. see Roberts, J., Rosier, P., Smith, D.M., 2005. The impact of broadleaved woodland on water resources in lowland UK: II. Evaporation estimates from sensible heat flux measurements over beech woodland and grass on chalk sites in Hampshire. Hydrology and Earth System Sciences 9 (6), 607–613.)? We can infer that the macro configuration results in more evapotranspiration, but which configuration is more realistic? Again, this goes to the heart of whether or not the macro model is an improvement, though I can't comment on this since the results are not presented clearly.

- We thank the reviewer for his comments. We have made significant effort to acquire the flux data from Warren/Sheepdrove Farm (see below) even prior to the original version of the manuscript. However, despite our efforts, we could not access the actual data. We were also unsuccessful in requesting the dataset to CEH Wallingford directly.

List of visited databases:
1. LOCAR main database as listed on NERC website: http://www.nwl.ac.uk/locar/main.htm
2. Centre for Environmental Data Analysis: http://www.ceda.ac.uk/
3. NERC Earth Observation Data Centre: http://neodc.nerc.ac.uk/
4. Environmental Information Data Centre: http://eidc.ceh.ac.uk/
5. Fluxnet: https://fluxnet.ornl.gov/site/784
6. European Fluxes Database Cluster: http://gaia.agraria.unitus.it/home/log-in/
7. Fluxmet: http://fluxmet.ceh.ac.uk/page/login.aspx

In terms of the catchment scale model application, it is notable that the macro model improves the evaporation estimates, and apparently improves runoff, although it looks rather conspicuous that no runoff plots are included. Why is this?

- Figure R3 below shows a comparison between observed and simulated discharge from the $macro_{opt}$ configuration (please refer to $man\_rev$) at the Theale gauging station (please refer to Figure 1a in $man\_rev$). Note that Figure R2 compares 10-day average runoff to minimize the effect of routing in the model. Considering the lack of groundwater, the observed and simulated runoff at Theale shows quite reasonable agreement.

[Figure]

Figure R3. Comparison between observed and simulated discharge from the $macro_{opt}$ configuration at the Theale gauging station.

A groundwater representation is likely needed for further improve the estimates of river discharge at gauging stations in the area. However, such implementation is beyond the scope of this study. In this study, we aim to demonstrate that the newly proposed BC model improves the overall mass and energy balance components compared to the *default* configuration that represents a "completely naïve model" (as the reviewer pointed out). We did not include the comparison between observed and simulated river discharge at gauging

stations in *man_rev*. Nevertheless, the seasonal aspect of discharge is captured reasonably well in our proposed model (Figure R2). The analysis of overall water balance (Table 4 of *man_rev*) corroborates the fact that the overall magnitude of the hydrological fluxes in the catchment is consistent with observations.

**Corrections to the text**

L. 8 Is it really the 'efficiency' of simulations that is the critical limitation?

- We thank the reviewer for making this point. In order to clarify this point, we have re-written the sentence as (L. 9-12 in *man_rev*)

"However, incorporating the processes governing water movement through chalk unsaturated zone in a numerical model is complicated mainly due to the fractured nature of chalk that creates high-velocity preferential flow paths in the subsurface."

L. 11 Poor grammar in this sentence, and the meaning is not completely clear. The mass and energy fluxes are influences regardless of whether the hydrology is complex or non-linear. Try to make this sentence more specific and meaningful.

- This sentence is removed from *man_rev*.

L. 13 I'm not sure this hypothesis is well phrased. It would be very surprising if there was no influence of this representation on the fluxes of water. As phrased, this suggests you are just looking at sensitivity, but I think it would be better to test whether or not this representation results in some sort of improvement? That is presumably what you're actually doing anyway.

- We have re-written the abstract following the comments of the reviewer in *man_rev*.

L. 17 Change "applied on" to "applied to"

- Corrected in *man_rev* (L. 18).

L. 28 "various processes" are there others, other than recharge?

- Re-written in *man_rev* as (L. 30-31):

"Previous studies showed that the unsaturated zone of the chalk aquifers plays an important role on groundwater recharge in the UK [e.g., *Lee et al.*, 2006; *Ireson et al.*, 2009]."

L. 36 The fracture porosity cited (10-4) seems much lower than other published estimates for the Chalk. I believe Price et al., 1993 actually cite a value of 10-2 (though I don't have this reference to hand, please check this).

- We have checked the values in *Price et al.* (1993) once again to confirm fracture porosity to be $10^{-4}$ as cited.

L. 42 Change "Mathuis" to "Mathias"

- Corrected in *man_rev* (L. 44).

L. 55 Change "curve was" to "curves were"
- Corrected in *man_rev* (L. 56).

L. 66 What is a "consistent representation …"? Consistent with what? This sentence restates your hypothesis, which I repeat is very unexciting and somewhat vague. You need to be much more specific here.
- This sentence is removed from *man_rev*.

L. 100 It is possible, but not certain, that fm is a sensitive parameter - in which case adopting this arbitrary value without sensitivity analysis or calibration would seem dubious. I'm also not convinced that it makes sense to make the fracture conductivity functionally dependent on the matrix conductivity - is there any physical reason to not to treat these two properties as independent?

- The results in Figure 4 and S2 in *man_rev* shows that $S_0$ is the most influential parameter in the model when simulating $\Delta\theta$, followed by the $K_s$ of chalk matrix while $f_m$ showed low sensitivity. Therefore, we selected $K_s$ and $S_0$ parameters for the optimization to minimize the differences between observed and simulated $\Delta\theta$.

The BC model is based on the work by *Zehe et al.* [2011], who proposed a linear increment of matrix conductivity for a fractured system. That the proposed parameterization is based on a single continuum approach that treats preferential flow considering modified conductivity close to saturation [*Beven and Germann*, 2013], which is observed from Equation 1 of *man_rev*. Additionally, note that the $f_m$ parameter that controls the increment of conductivity near saturation is based on the physical properties of chalk (i.e., the relative difference in permeability of fractured chalk and chalk matrix).

L. 167 Here you describe the hydraulic properties of the soil, but you don't described the properties of the Chalk until line 209. Would make sense to rearrange the text so that these are described in the same place in the text.

- In *man_rev*, the hydraulic properties of soil and chalk are consecutively discussed (L. 212-222).

L. 188 This is a little bit confusing - it reads like you are saying the hydraulic properties of the soil are uniform over the catchment? However, from Figure 1 and Table 2 this is not the case. Please clarify the text here.

- The soil hydraulic properties at the catchment scale are described in *man_rev* as (L. 213-218):

"At the catchment scale, the Harmonized World Soil Database (HWSD) from the Food and Agricultural Organization of UNO (FAO) is used to obtain the texture of different soil types over Kennet (Figure 1c). The saturation-pressure head relationship for different soil types is described using the Van Genuchten [*Van Genuchten*, 1980] model with parameter values (Table 2) obtained from *Schaap and Leij* [1998]."

L. 226 I think it is not particularly interesting that the model underestimates the absolute values of the observed soil moisture - I would be more interested in how well it reproduces the changes in soil moisture. It probably underestimates these, but it's not completely terrible. I'm surprised how well this simple default model with no calibration does!

- As discussed earlier, we have taken all steps to incorporate analysis of $\Delta\theta$ in the manuscript as proposed by the reviewer. We have also demonstrated that the *macro* configuration shows relatively better model performance in simulating $\Delta\theta$ compared to *default* (Figure 4 and 5 in *man_rev*).

L. 261 In the Ireson al al. (2009) paper referenced here (full disclosure - I am Ireson), Fig 13 shows the drainage flux at 5 m depth, which could be directly compared with Figure 5 in this paper. It can be seen in my paper there was negligible fracture flow at 5 m depth during this period, which is not consistent with the authors interpretation that "fracture flow dominates... during wet periods". So I'm afraid the result here in not consistent with my result - at least the macro model result is not - the default model might be!

- As mentioned by the reviewer, drainage flux at 5 m depth in Figure 13 of *Ireson et al.* [2009] may be compared to the drainage through bottom boundary ($d_b$, Figure 7 in *man_rev*). With regards to our results, the *macro* configuration (Figure 6 in *man_org*) shows very low drainage and no seasonal variability (also pointed out by Reviewer #3). With the optimized parameters, the *macro_{opt}* configuration shows significantly higher drainage (228 mm) compared to *macro* (112 mm). Additionally, the *macro_{opt}* configuration shows higher $d_b$ during the colder months of the year compared to summer which is more consistent with the recharge pattern in chalk compared to the *default* configuration that shows higher drainage in summer [*Wellings and Bell*, 1980; *Ireson et al.*, 2009].

L. 311 Why use R2 for this? RMSE or bias would be better - we are interested in the absolute values in this case.

- We have included bias in *man_rev* (Figure 8).

L. 336 Again we have references to this weak hypothesis that there is an "influence".

- We have removed this sentence from *man_rev*.

L. 362 Again the word "consistent" - consistent with what?

- We have removed this sentence from *man_rev*.

L. 372 You say these two parameters can be estimated from the matrix without calibration, but this is an assertion, since you haven't tested these parameters in this study. In fact, my central criticism of the findings in this paper are likely due to poor choices of parameter values in your model. If you were to increase your matrix K, I suspect the model would improve.

- In *man_rev*, we have carried out sensitivity analysis and parameter estimation (i.e., optimization) as recommended by the reviewer, which has been already discussed above.

L. 374 Overall, I cannot agree that the model was able to reproduce the hydrological processes in the Chalk successfully, or even to an acceptable degree. Groundwater recharge is completely wrong.

- We thank the reviewer for his valuable comment. We agree that simulating meaningful recharge is important because it drives the groundwater table dynamics. Therefore, in *man_rev* we have optimized the BC model parameters (i.e., the *macro_{opt}* configuration) to minimize the differences between observed and simulated $\Delta\theta$ (connected to recharge) following the suggestions of the reviewer. Figure 7 in *man_rev* demonstrates that the parameter optimization results in a more consistent seasonal variability of $d_b$ compared to *default* when the recharge patterns through chalk unsaturated zone is concerned.

L. 376 Yes, good that you suggest calibration, but without doing this the model is not ready for publication, since it fundamentally fails to simulate convincing groundwater recharge

fluxes. Especially on the 1D model, calibration really is not that hard to do, so must be done before this paper can be accepted, in my view.

- As discussed earlier, parameter optimization for the BC model is performed in *man_rev* as suggested by the reviewer.

L. 401 Delete this final sentence saying you will address coupling with a groundwater model in the future. That is, by definition, outside the scope of this paper, hence irrelevant.

- This sentence is deleted from *man_rev*.

L. 385 + The last three paragraphs of the conclusions are very disjointed.

- The conclusion is completely re-written in *man_rev*. The last three paragraphs of *man_org* does not appear in *man_rev*.

**Reviewer #3**

This paper proposes the bulk conductivity (BC) model for improving the simulation of chalk hydrology in land surface models. The bulk conductivity model appears a simple approach for simulating both matrix and fracture flow in the Chalk according to the relative saturation. This approach is implemented in JULES (macro) and the results are compared with JULES (default) runs using a typical soil parameterisation, but neither model is calibrated. This is undertaken at the point and, subsequently, catchment scale. The authors suggest that the addition of the BC model in JULES improves soil moisture, evaporation, and runoff simulation.

**Major comments**

The default soil parameterisation is based on soil texture data from the surface (a loam) down to 5 m depth (Line 164 & Table 2) despite Figure 2 showing that this soil horizon is only 20-30 cm deep, with the remaining profile being chalk. Is it then any real surprise that this uncalibrated JULES model performs worse that a JULES model modified specifically for simulating chalk hydrology? This is not a valid comparison and the conclusions drawn are not valid.

To make any comparison valid the default model runs have to be calibrated, in particular, to achieve a more appropriate soil parameterisation. Currently the default run for the catchment is simulating more than twice the observed runoff for the Kennet, which is not acceptable. Consequently, evaporation is underestimated and soil moisture storage is insufficient. The River Kennet could be used for calibration at the catchment scale and soil moisture data for the point scale.

- We thank the reviewer for his/her valuable comments. We emphasize that in land surface model development, a common approach is to test/compare the proposed improvement against the current state of the model, possibly related to its operational configuration. We refer for instance to the work from reviewer #1 (Dr Le Vine) in which the standard JULES model is used as the baseline model to investigate a series of subsequent improvements (please, refer to Table 3 in Le Vine et al., 2016). In addition, Reviewer #2 (Dr Andrew Ireson) has highlighted that the default configuration represents a "completely naïve model" setup that might be used in large scale applications over chalk-dominated areas.

As also recommended by reviewer #2, in *man_rev*, we have used the soil moisture data from the Warren Farm site to optimize the BC model parameters as suggested by the reviewer. The results suggest that this optimization improves simulated $\theta$ and $\Delta\theta$ compared to the default configuration (Figure 4, 5 and 6 in *man_rev*).

The macro model appears to be performing reasonably well where described in the text and calibration may not be as essential. However, I would like to see a sensitivity analysis for the BC parameters, which would be very useful for anyone considering implementing this approach for chalk models in the future, particularly when there are so few parameters.

- This is a very important point raised by the reviewer. Sensitivity analysis of the BC model parameters is shown in Figures 4 and S2 of *man_rev*.

Although the macro model is performing well where explicitly described in the text, Figure 6d asks some serious questions. The macro model is simulating about 0.1 mm/d of potential recharge with the exception of the early 2003 event. This would equate to only c.35 mm of potential recharge a year on a grassland site, on outcrop chalk, in a temperate climate. This seems unrealistically low and the c.200 mm of potential recharge simulated by the default model is perhaps more realistic.

- As recommended during the revision of this manuscript, the BC model parameters are optimized to minimize the differences between observed and simulated soil moisture variability in *man_rev* to improve the potential recharge in the JULES (following the review by Dr Andrew Ireson). The results (Figure 7 in *man_rev*) show that the drainage from the *macro$_{opt}$* (228 mm) is substantially higher than that of the *macro* configuration (112 mm) shown in Figure 6 of the original manuscript (*man_org* hereafter).

The hypothesis that is proposed and maintained throughout the paper is a minor point, in my opinion, and takes away from the headline story: a simple approach for simulating matrix/fracture flow in the Chalk unsaturated zone, which could be implemented in LSMs. The abstract needs to be completely re-written and re-focussed on the above comment. There is currently only one sentence (lines 20-23) concerning the results and implications and this is very vague.

- We thank the reviewer for making this recommendation (in agreement with reviewer #2). We have re-written the abstract and conclusion sections in *man_rev* focusing on the suitability of the BC model for land surface modelling applications given the simplicity of the proposed approach, as proposed by both reviewers.

The last three paragraphs of the conclusions (lines 385-403) could easily all be deleted or at least only be summarised in a couple of paragraphs. Currently it dilutes the section.

- The last three paragraphs of *man_org* is deleted following the suggestion of the reviewer.

**Minor comments**

Lines 28-29 – consider rewording

- Modified in *man_rev* as (L. 30-31):

"Previous studies showed that the unsaturated zone of the chalk aquifers plays an important role on groundwater recharge in the UK [e.g., *Lee et al.*, 2006; *Ireson et al.*, 2009]."
Line 55 – this should be plural

- Modified in *man_rev* (L. 55-58).

Section 3.1 – the Kennet is a tributary of the Thames
Throughout the paper 'the' is frequently omitted, e.g. 'River Kennet discharges', 'major tributarites of this river are Lambourn, etc

- We have improved the use of definite article in *man_rev*.

Line 122 – there are 3 years more soil moisture data available at Warren Farm, which CEH collected. These data extend into a wetter period when it would be interesting to see how the models compared.

- We thank the reviewer for this suggestion. We have made significant efforts to acquire additional in situ data from available databases and CEH. Despite our efforts, we were only able to obtain the soil moisture data used in the original manuscript. Please refer to a similar comment by reviewer #2 for additional detail on page 6.

Line 125 – suggests soil moisture observations only exist to 2.4 m depth but observations in Figure 3 suggest there are deeper data.

- We apologize for the misunderstanding. Please refer to L. 145-147 in *man_rev*:
"A Didcot neutron probe was used at these locations to measure fortnightly soil moisture at different depths below land surface (10 cm apart down to 0.8 m, 20 cm apart between 0.8-2.2 m, and 30 cm apart between 2.2-4.0 m) [*Hewitt et al.*, 2010]."

Liner 240 – change 'dry' to 'drier'

- Modified in *man_rev* (L. 291).

Section 4.2 – r2 informs on the fit of the linear regression but perhaps a plot of RMSE over time would be more useful to inform on actual differences between observed and modelled.

- We have used bias to assess the *actual* differences between LE from the two model configurations in *man_rev* (Figure 8, L. 320-325).

Section 4.2 – there appears to be a lot more noise in the modelled LE during Summer?

- The highest temporal resolution of available MODIS data is 8-day. In comparison, the simulated 8-day and monthly averages are calculated using hourly data (model time step). This may be the reason of differences in high-frequency variability (i.e., noise) mentioned by the reviewer.

Figures – blue is not always very distinguishable from black on the comparison plots

- We have changed the colour scheme in the plots in *man_rev*.

Figure 5 – Are the default and macro results from the same depth interval here?

- The results are binned at the same depth interval for the two model configurations (Figure 6 in *man_rev*).

**References**

Bakopoulou, C. (2015), Critical assessment of structure and parameterization of JULES land surface model at different spatial scales in a UK Chalk catchment, PhD thesis, Imperial College London, UK, available at: https://spiral.imperial.ac.uk:8443/handle/10044/1/28955.

Beven, K., and P. Germann (2013), Macropores and water flow in soils revisited, Water Resour. Res., 49, 3071–3092.

Ireson, A. M., S. A. Mathias, H. S. Wheater, A. P. Butler and J. Finch (2009), A model for flow in the chalk unsaturated zone incorporating progressive weathering, J. Hydrol., 365, 244-260.

Le Vine, N., A . Butler, N. McIntyre, and C. Jackson (2016), Diagnosing hydrological limitations of a land surface model: application of JULES to a deep-groundwater chalk basin, Hydrol. Earth Syst. Sc., 20, 143-159.

Price, A., R. A. Downing, and W.M. Edmunds (1993), The chalk as an aquifer. In: Downing, R. A., M. Price, and G. P. Jones *The hydrogeology of the chalk of north-west Europe*. Oxford: Claredon Press, 35-58.

Wellings, S. R., and J. P. Bell (1980), Movement of water and nitrate in the unsaturated zone of upper chalk near Winchester, Hants., England, J. Hydrol, 48, 119-136.

---

## Referee Report (RR1)

**Towards a simple representation of chalk hydrology in land surface modelling**
Mostaquimur Rahman, Rafael Rosolem

The paper describes the addition of the bulk conductivity (BC) model into the JULES land-surface model. The authors have implemented this very simple model in attempts to improve the simulation of chalk hydrology. The BC model is calibrated and shown to improve the simulated mass and energy fluxes over the Kennet catchment. The authors conclude by commenting on the potential suitability of this simple model for large scale land-surface modelling.

It is clear that the authors have taken the time to properly implement the comments and suggestions from the first pass of reviews which have greatly improved the manuscript. The rewording of the title and abstract highlight the strength of the simplicity of the approach, and the calibration of the BC model parameters makes for a much cleaner and more convincing study.

I have found the paper clunky to read in places. Some restructuring could help improve the readability of the manuscript.

Personally I feel that the calibration step needs to be clearer. The calibration step is not mentioned in the Abstract or Introduction of the paper. The word evaluation is used in places where 'calibration' would be more relevant. The BC model is first calibrated using soil moisture data at a point scale. The calibrated model performance is assessed and compared to the performance of the default JULES parameterization. Finally it is evaluated against independent data at a catchment scale.

**Minor Comments**

Introduction: It is worth mentioning in the introduction what default JULES does, this becomes clearer later but a sentence here to the effect: 'JULES has nothing in place for chalk' would be useful.

Pg 4 line 65: *'…relatively large number of parameters'* could be worth stating how many to contrast with the three parameter model described in the paper.

Pg 3 line 69: '*At the point-scale the BC model is **evaluated** using observed soil moisture data.*' maybe use calibrated instead of evaluated.

Pg 4 line 87: the units for $K_s$ are inconsistent with the units found in table 3.

Pg 5 line 105: it is not clear how Price et al.'s values of 3-5 mmd$^{-1}$ for $K_s$ imply a range of 0.8-86 mmd$^{-1}$ for calibration.

Pg 5 lines 112-123: Please consider moving the calibration description into the methodology.

Pg 8 line 189 or Pg 10 line 223: Consider discussing the choice of using the default JULES parameters in the configuration. It is not too surprising that the calibrated parameterization of BC model would outperform an uncalibrated JULES model. The authors address this point in the first pass of reviews but I feel some of the text would benefit from being in the text. Especially since it highlights the BC model's application to a completely 'naive' model setup.

FIg 1b): The 'Bare soil', 'Needleleaf' and 'Broadleaf' colours are very similar and hard to distinguish.

Fig 4 (panels (b) to (c)): It would help to shade the bars of the unchanged values differently so that the changes in parameter values becomes more obvious. These panels also don't get discussed much in the main text.

Pg 12 line 271-276: The sensitivity of $S_0$ is particularly interesting and in turn the model doesn't seem sensitive to $f_m$. I feel the sensitivity of the parameters could be further discussed. Why is it advantageous to use the macro configuration with the two optimized parameters vs the three optimized parameters?

Table 3: there seems to be a lack of consistency between Figure 4 and Table 3. In the text (line 276) '... *we select the macro configuration with optimized* $K_s$ *and* $S_0$...' which should mean $f_m$ remains unchanged. However in Table 3, $f_m$'s optimized value differs for its unoptimized value. The value for $f_m$ in this table might be from optimising over this parameter alone, but then the other parameters have optimized values which differ from their single optimisation value (e.g. $K_s$). What optimized values are shown in this table? Surely the values used in macro$_{opt}$ should be represented here.

A further comment on Fig.4 and Table 3: what are the units of $K_s$? In figure they are of order $x10^{-4}$, in the table units ($md^{-1}$) and earlier in the text ($mmd^{-1}$).

Table 3: Add ranges over which the parameters were allowed to vary in calibration stage.

Pg 13 line 196: It is worth commenting that it is still an underestimate.

Pg 16 line 375: Two parameters vs the three referenced in the rest of the paper… is $K_s$ no longer considered a parameter in the conclusion?

Fig 6: Consider adding vertical lines between the bins to highlight the fact the boxplots are of the same depth. This is not immediately clear and I've noticed that reviewer 3 also mentioned the slight confusion caused by this figure.

---

## Author Response (AR2)

We would like to thank the reviewers for the valuable comments and suggestions that improved the manuscript. A point-by-point reply to the comments of both reviewers is provided below with the original comments shown in blue.

**Reviewer #1 (anonymous)**

The paper describes the addition of the bulk conductivity (BC) model into the JULES Land-surface model. The authors have implemented this very simple model in attempts to improve the simulation of chalk hydrology. The BC model is calibrated and shown to improve the simulated mass and energy fluxes over the Kennet catchment. The authors conclude by commenting on the potential suitability of this simple model for large scale land-surface modelling.

It is clear that the authors have taken the time to properly implement the comments and suggestions from the first pass of reviews which have greatly improved the manuscript. The rewording of the title and abstract highlight the strength of the simplicity of the approach, and the calibration of the BC model parameters makes for a much cleaner and more convincing study.

- We would like to thank the reviewer for the encouraging comments.

I have found the paper clunky to read in places. Some restructuring could help improve the readability of the manuscript.

- We have re-structured the manuscript as suggested by the reviewer in the "Minor Comments" section below to improve readability.

Personally I feel that the calibration step needs to be clearer. The calibration step is not mentioned in the Abstract or Introduction of the paper. The word evaluation is used in places where 'calibration' would be more relevant. The BC model is first calibrated using soil moisture data at a point scale. The calibrated model performance is assessed and compared to the performance of the default JULES parameterization. Finally it is evaluated against independent data at a catchment scale.

- We thank the reviewer for this valuable comment. We have incorporated the calibration step in the introduction of the revised manuscript (L. 75-82 in manuscript_revised.pdf):

"At the point-scale, the proposed parameterization is calibrated using observed soil moisture profile data. This is achieved by randomly sampling the parameter space and extensively running the model in order to minimize the differences between observed and simulated soil moisture variability at different depths. Finally, the proposed model is applied to the Kennet catchment in the Southern England and the fluxes and states of the hydrological cycle are simulated for multiple years. The simulation results are evaluated using observed latent heat flux (LE) and runoff data to assess the performance of the BC model in simulating land surface processes at the catchment scale."

**Minor Comments**

Introduction: It is worth mentioning in the introduction what default JULES does, this becomes clearer later but a sentence here to the effect: 'JULES has nothing in place for chalk' would be useful.

- We have incorporated this comment in the introduction of the revised manuscript (L. 71-74 in manuscript_revised.pdf):

"In order to test the proposed parameterization, the BC model is included in JULES (version 4.2), which, by default (i.e., uniform soil column representation using general soil database as typically applied in land surface models), does not represent any chalk feature."

Pg 4 line 65: '...relatively large number of parameters' could be worth stating how many to contrast with the three parameter model described in the paper.

- We have mentioned the number of parameters used in relevant previous studies in the revised manuscript (L. 48-52 in manuscript_revised.pdf):

"The physics-based models mentioned above were developed based on dual-continua approach and required relatively large numbers of parameters (i.e., on the order of 20-30 parameters) that were calibrated via inverse modelling using observed soil moisture and matric potential data [e.g., *Ireson et al.*, 2009; *Mathias et al.*, 2006]."

Pg 3 line 69: 'At the point-scale the BC model is evaluated using observed soil moisture data.' maybe use calibrated instead of evaluated.

- Updated in the revised manuscript (L. 76 in manuscript_revised.pdf):

"At the point-scale, the proposed parameterization is calibrated using observed soil moisture profile data."

Pg 4 line 87: the units for Ks are inconsistent with the units found in table 3.

- We have used mmd$^{-1}$ as the unit of $K_s$ consistently in the revised manuscript (e.g., L. 97, L. 248-250, Table 2 and Table 3 in manuscript_revised.pdf)

Pg 5 line 105: it is not clear how Price et al.'s values of 3-5 mmd-1 for Ks imply a range of 0.8-86 mmd-1 for calibration.

- We thank the reviewer for pointing out this important issue. Note that we have chosen a different range of $K_s$ is the revised manuscript following the suggestions of the other reviewer (Dr. Andrew Ireson). While *Ireson et al.* [2009] suggested a range of 0.2-2.0 mmd$^{-1}$, *Price et al.* [1993] argued that $K_s$ is around 3-5 mmd$^{-1}$ for most chalk soils. Therefore, we consider a range of 0.2-5.0 mmd$^{-1}$ in the revised manuscript for $K_s$ calibration. This updated range of $K_s$ has been discussed in the revised manuscript (L. 248-250 in manuscript_revised.pdf):

"*Ireson et al.* [2009] suggested a range of 0.2-2.0 mmd$^{-1}$ for $K_s$. On the other hand, *Price et al.* [1993] argued that in general, $K_s$ is around 3-5 mmd$^{-1}$ for most chalk soils. Therefore, we consider a range of 0.2-5.0 mmd$^{-1}$ in optimizing $K_s$."

Pg 5 lines 112-123: Please consider moving the calibration description into the methodology.

- We have moved the description of calibration strategy to the "Methods" section (section 3.5) in the revised manuscript (L. 236-260 in manuscript_revised.pdf).

Pg 8 line 189 or Pg 10 line 223: Consider discussing the choice of using the default JULES parameters in the configuration. It is not too surprising that the calibrated parameterization of BC model would outperform an uncalibrated JULES model. The authors address this point in the first pass of reviews but I feel some of the text would benefit from being in the text. Especially since it highlights the BC model's application to a completely 'naive' model setup.

- We have explained the choice of the parameters in the *default* configuration and the fact that it represents a naïve model configuration deprived of model calibration and chalk representation in the revised manuscript (L. 217-220 in manuscript_revised.pdf):

"In this configuration, each soil column in JULES is considered to be vertically homogeneous with the soil properties defined in Table 2, which is motivated by the Met Office JULES Global Land 4.0 configuration described in *Walters et al.* [2014]."

(L. 231-234 in manuscript_revised.pdf):

"It should also be emphasized that *default* represents a "naïve" configuration deprived of model calibration. Moreover, this configuration does not represent chalk, which, according to previous studies [e.g., *Le Vine et al.*, 2016], substantially affects the hydrology of the study area considered here."

FIg 1b): The 'Bare soil', 'Needleleaf' and 'Broadleaf' colours are very similar and hard to distinguish.

- Figure 1b is updated in the revised manuscript of make the colours distinguishable.

Fig 4 (panels (b) to (c)): It would help to shade the bars of the unchanged values differently so that the changes in parameter values becomes more obvious. These panels also don't get discussed much in the main text.

- We have shaded the bars representing uncalibrated and calibrated parameter values using blue and red colours, respectively in Figure 4 of the revised manuscript. We have also enhanced the discussion on Figure 4b, c and d in the revised manuscript (L. 300-313 in manuscript_revised.pdf).

Pg 12 line 271-276: The sensitivity of S0 is particularly interesting and in turn the model doesn't seem sensitive to fm . I feel the sensitivity of the parameters could be further discussed. Why is it advantageous to use the macro configuration with the two optimized parameters vs the three optimized parameters?

- We have enhanced the discussion on the sensitivity of parameters on model performance in the revised manuscript (L. 282-299 in manuscript_revised.pdf). We have also discussed the benefit of reduced model complexity (i.e., reduced number of parameters for calibration) and our choice of parameters for calibration (i.e., $K_s$ and $S_o$) in the revised manuscript (L. 293-299 in manuscript_revised.pdf):

"Arguably, the BC model can be implemented in other chalk regions by constraining only $S_o$ parameter. Such result could potentially be advantageous for transferability to other regions in the UK in order to assess chalk hydrology at large-scale. However since this is the first time the BC model is introduced, we decide to take a conservative approach and select the *macro* configuration with optimized $K_s$ and $S_o$ (*macro_{opt}* hereafter) to simulate chalk hydrology over the study area that ensures best overall model performance."

(L. 408-415 in manuscript_revised.pdf):

"Our results indicated that $S_o$ is by far the most influential parameter in the model when representing water movement through a soil-chalk column. This highlights the simplicity of the proposed BC model for large-scale studies and potential ease in transferability. In comparison, $K_s$ and $f_m$ showed secondary (low) sensitivity on the model performance. Since this study introduces the BC model, we decided however to take a conservative approach. We optimized $K_s$ and $S_o$ simultaneously for our catchment scale simulations since this combination resulted in the best overall model performance."

Table 3: there seems to be a lack of consistency between Figure 4 and Table 3. In the text (line 276) '. .. we select the macro configuration with optimized Ks and S 0 ... ' which should mean f m remains unchanged. However in Table 3, fm 's optimized value differs for its unoptimized value. The value for f m in this table might be from optimising over this parameter alone, but then the other parameters have optimized values which differ from their single optimisation value (e.g. Ks ). What optimized values are shown in this table? Surely the values used in macro opt should be represented here.

- We thank the reviewer for raising this important issue. We agree that the values presented in Table 3 of the previous manuscript were not consistent with our choice of parameters for calibration (i.e., $K_s$ and $S_o$). We have updated Table 3 in the revised manuscript (P. 27 in manuscript_revised.pdf). Note that in updated Table 3, calibrated value for $K_s$ (0.31 mmd$^{-1}$) and $S_o$ (0.46) are presented, while $f_m$ (10$^5$) remains uncalibrated.

A further comment on Fig.4 and Table 3: what are the units of Ks ? In figure they are of order x10 4, in the table units (md 1) and earlier in the text (mmd 1).

- We have updated Figure 4 and added units of $K_s$ (mmd$^{-1}$) in the revised manuscript (P. 31 of manuscript_revised.pdf). The unit of $K_s$ in Table 3 of the revised manuscript is also updated to mmd$^{-1}$ (P. 27 in manuscript_revised.pdf).

Table 3: Add ranges over which the parameters were allowed to vary in calibration stage.

- Ranges are added in Table 3 of the revised manuscript (P. 27 in manuscript_revised.pdf).

Pg 13 line 196: It is worth commenting that it is still an underestimate.

- Note that this sentence along with the associated figure (i.e., Figure 6 in the previous manuscript) is removed from the revised manuscript. In manuscript_revised.pdf, we focus on the simplicity of the BC parameterization and the calibration of the model parameters to reduce the differences between observed and simulated soil moisture variability based on the comments by Dr. Andrew Ireson during the previous rounds of review. The results show that the calibrated model improves simulated key hydrological processes over the Kennet catchment compared to the *default* configuration. Therefore Figure 6 (previous manuscript) is no longer necessary because all the relevant information (i.e., the comparison between observed and simulated soil moisture variability) is shown in Figure 5 of manuscript_revised.pdf.

Pg 16 line 375: Two parameters vs the three referenced in the rest of the paper... is K s no longer considered a parameter in the conclusion?

- We would like to thank the reviewer for pointing out this issue. This discussion on the model parameters is clarified in the revised manuscript (L. 403-415 in manuscript_revised.pdf):

"The proposed BC model is a single continuum approach of modelling preferential flow [e.g., *Beven and Germann*, 2013] that involves only 3 parameters, namely the saturated hydraulic conductivity of chalk matrix ($K_s$), macroporosity factor ($f_m$) and relative saturation threshold ($S_o$). Initially, these parameters were estimated from existing literature to assess the performance of the uncalibrated BC model. Finally, the BC model parameters were optimized to minimize the differences between observed and simulated soil moisture variability. Our results indicated that $S_o$ is by far the most influential parameter in the model when representing water movement through a soil-chalk column. This highlights the simplicity of the proposed BC model for large-scale studies and potential ease in transferability. In comparison, $K_s$ and $f_m$ showed secondary (low) sensitivity on the model performance. Since this study introduces the BC model, we decided however to take a conservative approach. We optimized $K_s$ and $S_o$ simultaneously for our catchment scale simulations since this combination resulted in the best overall model performance."

Fig 6: Consider adding vertical lines between the bins to highlight the fact the boxplots are of the same depth. This is not immediately clear and I've noticed that reviewer 3 also mentioned the slight confusion caused by this figure.

- As mentioned above, this figure is removed from the revised manuscript.

**Reviewer #2 (Dr. Andrew Ireson)**

The paper is much improved, and I think that this is a useful contribution. However, there is a lingering problem that really should be addressed before the paper can be published. The authors have misinterpretted the hydraulic conductivity in the LeVine Paper (cited in the manuscript), which was a bulk hydraulic conductivity and not a matrix conductivity. Moreover, in the optimisation of K, the authors have not sampled K on a log-scale (unless they have and have neglected to say this), meaning that the optimized K is biased towards higher values. As a result, I believe the authors still have a matrix K which is unrealistically large compared with other published estimates. In my opinion, this is important and should be corrected before this paper can be published, which unfortunately means re-running the models.

- We would like to thank the reviewer for his valuable suggestions. In the revised manuscript, we have selected a different range for $K_s$ calibration based on relevant previous studies. While *Ireson et al.* [2009] suggested a range of 0.2-2.0 mmd$^{-1}$, *Price et al.* [1993] argued that $K_s$ is around 3-5 mmd$^{-1}$ for most chalk soils. Therefore, we consider a range of 0.2-5.0 mmd$^{-1}$ in the revised manuscript for $K_s$ calibration. This updated range of $K_s$ has been discussed in the revised manuscript (L. 248-250 in manuscript_revised.pdf):

"*Ireson et al.* [2009] suggested a range of 0.2-2.0 mmd$^{-1}$ for $K_s$. On the other hand, *Price et al.* [1993] argued that in general, $K_s$ is around 3-5 mmd$^{-1}$ for most chalk soils. Therefore, we consider a range of 0.2-5.0 mmd$^{-1}$ in optimizing $K_s$."

Another significant error, though easily fixed, is that the revised manuscript still does not include a plot of runoff for the catchment scale application. Such a plot was included in the response to reviewers (Figure R2 in the response), so should be incorporated into the manuscript. I think that the performance in this figure was reasonable. They should also show the default model performance.

- We thank the reviewer for highlighting this positive result from our study. As suggested, we have incorporated the comparison between observed and simulated runoff (for both *default* and *macro$_{opt}$*) in the revised manuscript (Figure 8b, L. 360-368 in manuscript_revised.pdf).

If this can be quickly addressed, this will be a nice paper.

I have also identified some minor errors to be corrected, below:

- Corrected in the revised manuscript (L. 50 in manuscript_revised.pdf):

L 80 - to be rigorous you should also show what "K_sb" is when S <= S_0.

- Updated in the revised manuscript (L. 90 in manuscript_revised.pdf)

L 87 - I believe the authors have misinterpretted the LeVine paper, which has a bulk saturated hydraulic conductivity of 16 mm/d, and not a matrix hydraulic conductivity. This number is at least an order of magnitude too high (see e.g. Ireson et al 2009, Price et al 1993, Price et al 2000, Brouyere 2006, all cited in the manuscript).

- As discussed earlier, we have selected a different range for $K_s$ calibration (0.2-5.0 mmd$^{-1}$) based on relevant previous studies as suggested by the reviewer (L. 248-250 in manuscript_revised.pdf):

"*Ireson et al.* [2009] suggested a range of 0.2-2.0 mmd$^{-1}$ for $K_s$. On the other hand, *Price et al.* [1993] argued that in general, $K_s$ is around 3-5 mmd$^{-1}$ for most chalk soils. Therefore, we consider a range of 0.2-5.0 mmd$^{-1}$ in optimizing $K_s$.".

Table 2 & 3 should probably use the same units for hydraulic conductivity - m/d is good.

- Updated in the revised manuscript (P. 27 in manuscript_revised.pdf).

- Updated in the revised manuscript (Figure 1 in manuscript_revised.pdf).

Figure 4 - Need to label the units of K - and what are these? Not mm/d or m/s as far as I can tell - maybe mm/s? Better to use m/d.

- Unit (mmd$^{-1}$) is added in the revised manuscript (Figure 4 in manuscript_revised.pdf).

L 106 - the range in Ks considered is good, though I would expect the true value to be right at the bottom of this range. Did you sample Ks in log-space? I assume not - which is a problem as it will bias your sampling to the higher values. This is important - you need to sample in log-space. I think your default and optimized Ks are still too high (i.e. values in Figure 4). Other than this, the optimization approach is good.

- We have sampled $K_s$ in log space in the revised manuscript (L. 256-257 in manuscript_revised.pdf):

[revised manuscript text omitted]